# AMiD: Knowledge Distillation for LLMs with $\alpha$-mixture Assistant Distribution

**Donghyeok Shin**[1], **Yeongmin Kim**[1], **Suhyeon Jo**[1], **Byeonghu Na**[1], **Il-Chul Moon**[1,2]

[1]Korea Advanced Institute of Science and Technology (KAIST), [2]summary.ai
{tlsehdgur0,alsdudrla10,suhyeonjo,byeonghu.na,icmoon}@kaist.ac.kr

## ABSTRACT

Autoregressive large language models (LLMs) have achieved remarkable improvement across many tasks but incur high computational and memory costs. Knowledge distillation (KD) mitigates this issue by transferring knowledge from a large teacher to a smaller student through distributional alignment. Previous studies have proposed various discrepancy metrics, but the capacity gap and training instability caused by near-zero probabilities, stemming from the high-dimensional output of LLMs, remain fundamental limitations. To overcome these challenges, several approaches implicitly or explicitly incorporating assistant distribution have recently been proposed. However, the past proposals of assistant distributions have been a fragmented approach without a systematic investigation of the interpolation path and the divergence. This paper proposes $\alpha$-mixture assistant distribution, a novel generalized family of assistant distributions, and $\alpha$-mixture distillation, coined AMiD, a unified framework for KD using the assistant distribution. The $\alpha$-mixture assistant distribution provides a continuous extension of the assistant distribution by introducing a new distribution design variable $\alpha$, which has been fixed in all previous approaches. Furthermore, AMiD generalizes the family of divergences used with the assistant distributions based on optimality, which has also been restricted in previous works. Through extensive experiments, we demonstrate that AMiD offers superior performance and training stability by leveraging a broader and theoretically grounded assistant distribution space. We release the code at https://github.com/aailab-kaist/AMiD.

## 1 INTRODUCTION

Autoregressive large language models (LLMs) have recently achieved remarkable advances, delivering outstanding performance across a wide spectrum of tasks and application domains (Achiam et al., 2023; Touvron et al., 2023; Team et al., 2024). However, their massive parameter scales impose prohibitive computational and memory costs, which hinder their deployment in practical applications. Accordingly, an essential objective for practical deployment is to compress these high-capacity models by reducing the parameter count while preserving their strong performance.

Knowledge distillation (KD) (Hinton et al., 2015) is a widely adopted compression technique that transfers knowledge from a large teacher model to a smaller student model by aligning their token-level predictive distributions. The selection of a discrepancy metric is an important research topic in KD for LLMs. Several prior studies have proposed either (1) the use of various forms of divergence, including the capability of regulating the quality-diversity trade-off (Wang et al., 2025), or (2) employing a combination of these divergences (Agarwal et al., 2024; Wu et al., 2025) as the discrepancy metric. However, these approaches do not fundamentally resolve the large capacity gap between the high-capacity teacher and smaller student models, and the optimization instability due to near-zero probabilities, which is prevalent in the high-dimensional probability space of LLMs.

A practical remedy is to introduce an *assistant distribution* that interpolates teacher and student distributions to stabilize optimization and bridge this capacity gap. Recently, several methodologies have been proposed that either (1) utilize the discrepancy metric that inherently includes a specific form of assistant distribution (Agarwal et al., 2024; Ko et al., 2024; 2025) or (2) explicitly model the assistant distribution (Shing et al., 2025). However, these approaches have generally been treated as independent recipes in different papers without a systematic study, which hinders the development of general and effective methodologies.

In this paper, we propose a generalized framework that integrates the fragmentarily employed assistant distribution and divergence. First, we interpret the existing assistant distributions from the information theory view, revealing that the existing methodology can be expressed as an $m$-mixture, which mixes two probability distributions via arithmetic mean, and an $e$-mixture, which mixes them via geometric mean. Next, we present a new assistant distribution family, coined $\alpha$-mixture assistant distribution, by extending the mean concept via the generalized $f_\alpha$ mean. The $\alpha$-mixture assistant distribution introduces a new design variable $\alpha$ for the assistant distribution, which adjusts the geometry of the interpolation path. Here, $\alpha$ is an independent parameter distinct from the well-utilized parameter $\lambda$, which controls the portion of interpolation. The $\alpha$-mixture assistant distribution not only includes the existing assistant distributions as a special case ($\alpha = \pm 1$) but also provides several new assistant distribution that were not investigated in KD for LLMs area.

Under the concept of $\alpha$-mixture assistant distribution, we investigate several properties of the $\alpha$-mixture assistant distribution, which are meaningful in KD for LLMs, such as the analysis with $\alpha$-divergence, controllable support via $\alpha$, and continuity with respect to $\alpha$. Next, we propose a new KD framework for LLMs, coined as $\underline{\alpha}$-$\underline{\text{mixture}}$ $\underline{\text{distillation}}$ (AMiD), which generalizes the optimization schemes of prior research by unifying both the assistant distribution and the divergence. AMiD aims to align the $\alpha$-mixture assistant distribution and either the teacher or student. We theoretically prove the optimality of AMiD, which enables us to achieve the primary goal of KD (teacher = student) even when employing arbitrary divergence, $\alpha$, and $\lambda$, under the perfect optimization assumption. Furthermore, through gradient analysis when employing $f$-divergence, we theoretically demonstrate that $\alpha$ adjusts the mode-covering and mode-seeking properties of the student distribution, with both toy experiments and real-world experiment results supporting this finding. Across various evaluation scenarios, our proposed framework AMiD consistently demonstrates superior performance compared to methodologies that do not utilize the assistant distribution and those employing limited assistant distribution.

## 2 PRELIMINARY

### 2.1 KNOWLEDGE DISTILLATION FOR LARGE LANGUAGE MODELS

We denote the input prompt and output token sequences as $x$ and $y$, respectively, where $y \coloneqq (y_1, y_2, \ldots, y_L) \in \mathcal{V}^L$ is a token sequence of length $L$, with each token drawn from the vocabulary set $\mathcal{V}$. Given the input $x$, an autoregressive large language model (LLM) outputs a next-token distribution $p(y_l|x, y_{<l})$, conditioned on both the prompt $x$ and the previously generated tokens $y_{<l} \coloneqq (y_1, y_2, \ldots, y_{l-1})$. We assume access to two LLMs: a large fixed teacher model $p(y_l|x, y_{<l})$, and a smaller student model $q_\theta(y_l|x, y_{<l})$ parameterized by $\theta$. The goal of knowledge distillation (KD) for LLMs is to transfer the knowledge of the teacher into the student. Concretely, KD for LLMs is typically formulated as aligning the next-token distributions of the teacher and student:

$$\min_\theta \mathbb{E}_{(x,y)\sim\mathcal{D}} \left[ \sum_{l=1}^{L} D(p(y_l|x, y_{<l}), q_\theta(y_l|x, y_{<l})) \right] \tag{1}$$

where $D$ denotes the divergence and the dataset $\mathcal{D}$ is composed of the predefined dataset (Hinton et al., 2015), or various strategies using the student-generated outputs (SGOs): on-policy (Lin et al., 2020), a mixed approach (Agarwal et al., 2024; Gu et al., 2024; Xu et al., 2025), and an adaptive off-policy (Ko et al., 2024). For notational brevity, we omit the explicit dependence on $x$ and $y$ whenever it is clear from context, writing $p \coloneqq p(y_l|x, y_{<l})$ and $q_\theta \coloneqq q_\theta(y_l|x, y_{<l})$.

The choice of divergence $D$ plays a pivotal role in KD for LLMs. The widely used Kullback–Leibler (KL) divergence $D_{\text{KL}}(p\|q_\theta) \coloneqq \sum_k p(k) \log \frac{p(k)}{q_\theta(k)}$ in KD (Hinton et al., 2015; Kim & Rush, 2016) emphasizes mode-covering, often assigning mass to less informative regions. To mitigate this effect, the reverse KL divergence $D_{\text{RKL}}(p\|q_\theta) \coloneqq D_{\text{KL}}(q_\theta\|p)$ is employed for its mode-seeking properties (Gu et al., 2024), which possesses mode-seeking properties, but either choice entails a trade-off between quality and diversity. Recent studies address this by (1) combining divergences, e.g., GKD (Agarwal et al., 2024) with the generalized Jensen–Shannon divergence $D_{\text{GJS}}(p|q_\theta) \coloneqq \lambda D_{\text{KL}}(p\|\lambda p + (1-\lambda)q_\theta) + (1-\lambda)D_{\text{KL}}(q_\theta\|\lambda p + (1-\lambda)q_\theta)$, and (2) extending classical divergences to enable explicit control, as in ABKD (Wang et al., 2025), which adopts the $\alpha$-$\beta$-divergence $D_{\text{AB}}$ (Cichocki et al., 2011) as a generic framework. Concurrently, CSD (Kim et al., 2026) adapts a concrete score (Meng et al., 2022) to avoid the constraints of probability matching.

Meanwhile, several methodologies have recently been proposed to improve the optimization stability of KD for LLMs. Ko et al. (2024) leverages the skew KL divergence $D_{\text{SKL}}(p\|q_\theta) \coloneqq D_{\text{KL}}(p\|\lambda p + (1-\lambda)q_\theta)$ and the skew reverse KL divergence $D_{\text{SRKL}}(p\|q_\theta) \coloneqq D_{\text{KL}}(q_\theta\|\lambda p + (1-\lambda)q_\theta)$. TAID (Shing et al., 2025) introduces an adaptive intermediate distribution that gradually shifts from the student's initial distribution to the teacher distribution, i.e., $D_{\text{TAID}}(p\|q_\theta) \coloneqq D_{\text{KL}}(r_t\|q_\theta)$ where $r_t \coloneqq \text{softmax}((1-\lambda_t) \cdot \text{logit}(q'_\theta) + \lambda_t \cdot \text{logit}(p))$ with time-dependent interpolation parameter $\lambda_t$, detached student logits $\text{logit}(q'_\theta)$, and teacher logits $\text{logit}(p)$.

## 2.2 $m$-MIXTURE AND $e$-MIXTURE

Mixture models are a standard tool for integrating information from multiple distributions. Information geometry (Amari, 2016; Nielsen, 2020; Eguchi & Komori, 2022) provides a dualistic structure on the manifold of probability distributions, characterized by two affine connections: the mixture connection and the exponential connection. These connections induce two natural ways of interpolating between distributions, commonly referred to as the $m$-*mixture* and the $e$-*mixture*.

Given two probability distributions $p$ and $q$ defined on the same measureable space, the $m$-mixture is defined as a convex combination of $p$ and $q$:

$$p^{(m)}(x) \coloneqq (1-t)p(x) + tq(x), \qquad t \in [0,1] \tag{2}$$

In contrast, the $e$-mixture is defined multiplicatively:

$$p^{(e)}(x) \coloneqq \frac{p(x)^t q(x)^{1-t}}{Z(t)}, \qquad Z(t) \coloneqq \int p(x)^t q(x)^{1-t} dx \tag{3}$$

The $m$-mixture forms a straight line in probability space, while the $e$-mixture forms one in log-probability space. Some studies leverage $m$- and $e$-mixtures, for example, to construct paths for annealed importance sampling (Grosse et al., 2013; Masrani et al., 2021).

## 2.3 GENERALIZED $f$-MEAN

Generalized $f$-mean (Kolmogorov & Castelnuovo, 1930) is a generalized framework of the mean by using a monotonically increasing differentiable function $f : \mathbb{R} \to \mathbb{R}$. Given a set of weights $\{w_i \in \mathbb{R}^+ \mid \sum_i w_i = 1\}$ and the set of corresponding input elements $\{u_i \in \mathbb{R}\}$, the generalized $f$-mean is defined as:

$$m_f(\{w_i\}, \{u_i\}) \coloneqq f^{-1}\left(\sum_i w_i f(u_i)\right) \tag{4}$$

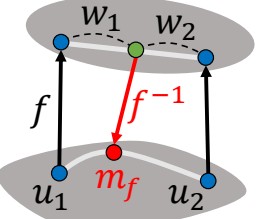

Figure 1: Illustration of generalized $f$-mean.

The $m_f$ applies a nonlinear transformation to the inputs, combines them with weights in the transformed domain, and maps the result back to the original domain. The well-known means, such as the arithmetic mean and geometric mean, have *homogeneity*, which stands for a scale-free property $m_f(\{w_i\}, \{c \cdot u_i\}) = c \cdot m_f(\{w_i\}, \{u_i\})$ for $c > 0$. The generalized $f$-mean is homogeneous only when $f$ belongs to the unique class of functions (Hardy, 1952; Amari, 2007):

$$f(u) \coloneqq f_\alpha(u) = \begin{cases} u^{\frac{1-\alpha}{2}}, & \alpha \neq 1 \\ \log u, & \alpha = 1 \end{cases}, \qquad u \in \mathbb{R}^+ \tag{5}$$

This family includes various notable examples, such as the weighted arithmetic mean for $\alpha = -1$, the weighted geometric mean for $\alpha = 1$, the weighted harmonic mean for $\alpha = 3$, and $\min\{u_i\}, \max\{u_i\}$ for $\alpha \to \infty$ and $\alpha \to -\infty$, respectively.

## 3 METHODOLOGY

This section introduces a new KD framework for LLMs, coined $\alpha$-mixture distillation (AMiD), which generalizes both the assistant distribution and the associated optimization scheme. Section 3.1 reveals the connection among the existing assistant distributions and highlights the need for a systematic study. Section 3.2 proposes the $\alpha$-mixture assistant distribution, which provides a unified and generalized assistant distribution family via $\alpha$-mixture distribution. Finally, Section 3.3 extends the assistant-based KD objective into a generic divergence framework.

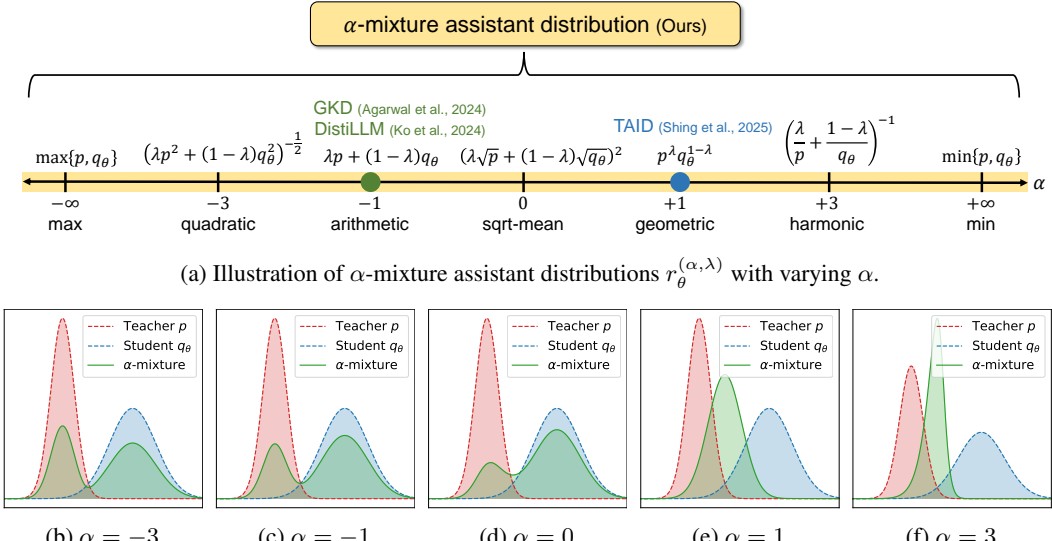

(a) Illustration of $\alpha$-mixture assistant distributions $r_\theta^{(\alpha,\lambda)}$ with varying $\alpha$.

(b) $\alpha = -3$     (c) $\alpha = -1$     (d) $\alpha = 0$     (e) $\alpha = 1$     (f) $\alpha = 3$

Figure 2: Visualization of the $\alpha$-mixture assistant distribution family. (a) The $\alpha$-mixture assistant distribution provides a generalized framework for assistant distributions, with prior studies (Agarwal et al., 2024; Ko et al., 2024; Shing et al., 2025) recoverable as special cases. (b-f) Illustration of the $\alpha$-mixture assistant distribution where $p = \mathcal{N}(0, 0.5^2)$, $q_\theta = \mathcal{N}(3, 1^2)$, and $\lambda = 0.3$. For $\alpha < 1$, the support of the $\alpha$-mixture assistant distribution corresponds to the union of the supports of $p$ and $q_\theta$, whereas for $\alpha \geq 1$, it corresponds to their intersection.

## 3.1 MOTIVATION

Our primary motivation stems from the observation that recent studies inherently include the composition of the teacher distribution $p$ and the student distribution $q_\theta$, which we will refer as *assistant distribution* $r_\theta$ in this paper. For example, several studies (Agarwal et al., 2024; Ko et al., 2024) utilize the divergences that include $r_\theta := \lambda p + (1 - \lambda)q_\theta$ with $\lambda \in [0, 1]$, which is an weighted arithmetic mean, also known as $m$-*mixture*. Moreover, we have newly discovered that the assistant distribution of TAID (Shing et al., 2025) is $e$-*mixture*, also known as a weighted geometric mean.

**Proposition 3.1.** *The assistant distribution of TAID (Shing et al., 2025) is e-mixture of $p$ and $q_\theta$:*[1]

$$r_\theta := softmax((1 - \lambda) \cdot logit(q_\theta) + \lambda \cdot logit(p)) \propto p^\lambda q_\theta^{1-\lambda} \tag{6}$$

Please refer to Appendix A.2 for proof. Using the assistant distribution provides several advantages in KD for LLMs. First, the assistant distribution facilitates more effective knowledge transfer between the teacher and the student. In KD, a significant capacity gap often arises due to differences in model size (Mirzadeh et al., 2020), and this issue becomes particularly pronounced in LLMs (Zhang et al., 2023; Sun et al., 2025) due to the high-dimensional nature. This gap makes it difficult for the student to faithfully capture the knowledge encoded in the teacher (Mirzadeh et al., 2020; Shing et al., 2025). By introducing the assistant distribution that serves as a bridge between the teacher and student, the information transfer might be more efficient (Shing et al., 2025). Second, the assistant distribution improves training stability. Due to the high-dimensional nature of LLMs, most of probabilites in $p$ and $q_\theta$ are inevitably close to zero. These near-zero probabilities might cause instability in both the loss and the gradient computation when divergences involving density ratios (e.g., KL divergence) are used (Ko et al., 2024). A suitably constructed assistant distribution yields more stable density-ratio estimates, thereby enhancing the robustness of optimization (Ko et al., 2024).

Despite these advantages, no systematic study has examined (1) the distinction between $m$- and $e$-mixture assistant distributions, (2) alternative candidates, (3) their compatibility with diverse divergences, and (4) their implications for KD in LLMs, supported by theoretical and empirical analyses. This gap hinders the development of general and effective methodologies, so the recent studies often fall into sub-optimal performances by relying on an isolated design of assistant distribution. In this paper, we alleviate this gap by unifying the existing assistant distributions into a generalized design principle of assistant distribution.

---

[1]We omit the time index $t$ and detached notation for the sake of uniformity.

## 3.2 $\alpha$-MIXTURE ASSISTANT DISTRIBUTION

As discussed in Section 3.1, the existing assistant distributions can be formulated as the mixture of the teacher distribution and the student distribution via the mean function. To integrate the fragmentarily employed assistant distributions, we employ the generalized $f_\alpha$-mean (Amari, 2016) and introduce a new assistant distribution family, coined $\alpha$-*mixture assistant distribution* as follows:

**Definition 1** ($\alpha$-*mixture assistant distribution*). Let $\alpha \in \mathbb{R}$ and $\lambda \in [0, 1]$. For distributions $p$ and $q_\theta$ defined either on a discrete support $\mathcal{X}$ indexed by $k$ or on a continuous domain $\mathcal{X}$ with variable $x$, define the unnormalized $\alpha$-mixture assistant distribution as:

$$\tilde{r}_\theta^{(\alpha,\lambda)}(z) = \begin{cases} \left( \lambda\, p(z)^{\frac{1-\alpha}{2}} + (1-\lambda)\, q_\theta(z)^{\frac{1-\alpha}{2}} \right)^{\frac{2}{1-\alpha}}, & \text{if } \alpha \neq 1, \\ p(z)^\lambda\, q_\theta(z)^{1-\lambda}, & \text{if } \alpha = 1, \end{cases} \tag{7}$$

where $z = k$ in the discrete case and $z = x$ in the continuous case.

Consequently, the (normalized) $\alpha$-mixture assistant distribution is defined as:

$$r_\theta^{(\alpha,\lambda)}(z) = \frac{\tilde{r}_\theta^{(\alpha,\lambda)}(z)}{Z_r}, \qquad Z_r := \sum_k \tilde{r}_\theta^{(\alpha,\lambda)}(k) \ \text{ or } \ \int_{\mathcal{X}} \tilde{r}_\theta^{(\alpha,\lambda)}(x)\, dx \tag{8}$$

The $\alpha$-mixture assistant distribution $r_\theta^{(\alpha,\lambda)}$ contains two tunable parameters: $\alpha$ and $\lambda$. The $\lambda$ determines the portion of the interpolation between teacher $p$ and student model $q_\theta$, which has been fine-tuned in previous works (Agarwal et al., 2024; Ko et al., 2024; Shing et al., 2025). The other parameter $\alpha$ is a new axis of distribution design variable, which was only employed as a specialized case ($\alpha = \pm 1$), controls the geometry of the interpolation path, as depicted in Figure 3a. Since the form of the generalized $f_\alpha$-mean is solely governed by $\alpha$, once $\alpha$ is fixed, $\lambda$ only serves to control the portion between $p$ and $q_\theta$ along that determined path. In addition, Theorem 3.2 provides an helpful information geometric perspective of the $\alpha$-mixture (assistant) distribution:

**Theorem 3.2.** *(Amari, 2007) Given a fixed $\alpha$ and $\lambda$, the $r^{(\alpha,\lambda)}$ defined as Eq. (8) is unique minizer of a weighted sum of Amari's $\alpha$-dviergences $D_\alpha$ (see Appendix A.1 for the definition):*

$$r^{(\alpha,\lambda)} = \arg\min_r \lambda \cdot D_\alpha(p\|r) + (1-\lambda) \cdot D_\alpha(q\|r) \tag{9}$$

Theorem 3.2 indicates that $r_\theta^{(\alpha,\lambda)}$ is the internal division distribution of $p$ and $q_\theta$ in terms of $\alpha$-divergence, which bridges the generalization of the mean concept and the geodesic in information geometry. Due to the generalized $f_\alpha$-mean, the existing assistant distributions are recoverable as special instances of $r_\theta^{(\alpha,\lambda)}$: $r_\theta^{(-1,\lambda)}$ is $m$-mixture (Agarwal et al., 2024; Ko et al., 2024) that is minimizer of a weighted sum of $D_{\text{KL}}$, and $r_\theta^{(1,\lambda)}$ is $e$-mixture (Shing et al., 2025) that is minimizer of a weighted sum of $D_{\text{RKL}}$. Furthermore, $r_\theta^{(\alpha,\lambda)}$ provides several new assistant distributions that were not previously used in KD literature, as depicted in Figure 2a.

Moreover, the support of $\alpha$-mixture assistant distribution determined by the range of $\alpha$: $\text{supp}(r_\theta^{(\alpha,\lambda)}) = \text{supp}(p) \cup \text{supp}(q_\theta)$ when $\alpha < 1$, and $\text{supp}(r_\theta^{(\alpha,\lambda)}) = \text{supp}(p) \cap \text{supp}(q)$ when $\alpha \geq 1$. This property demonstrates the necessity of determining the range of $\alpha$ based on the characteristics at the intersection of $p$ and $q_\theta$. For instance, if $p$ and $q_\theta$ overlap significantly, setting $\alpha \geq 1$ can strengthen the matching within the intersection region. Conversely, if they overlap minimally, setting $\alpha < 1$ indicates that matching occurs across a broader range. Although in KD for LLMs, $p$ and $q_\theta$ typically share the same support defined by the vocabulary set, this property remains useful because many probabilities are very small close to zero due to the high dimensionality. Figures 2b-2f shows the different behaviors of $r_\theta^{(\alpha,\lambda)}$ among the various $\alpha$ values.

Lastly, we also demonstrate that $r_\theta^{(\alpha,\lambda)}$ is a continuous function with respect to $\alpha$ in Proposition 3.3, even though the $r_\theta^{(\alpha,\lambda)}$ is a piecewise-defined function. This property enables the design of a curriculum-based adaptive $\alpha$ scheduling, paralleling prior work (Shing et al., 2025; Ko et al., 2025) that investigated adaptive strategies for $\lambda$. Please refer to Appendix A.3 for proof.

**Proposition 3.3.** *(Continuity) Assume that $p$ and $q_\theta$ are not both zero. Then, $r_\theta^{(\alpha,\lambda)}$ is continuous function w.r.t $\alpha$ under the fixed $\lambda \in [0, 1]$.*

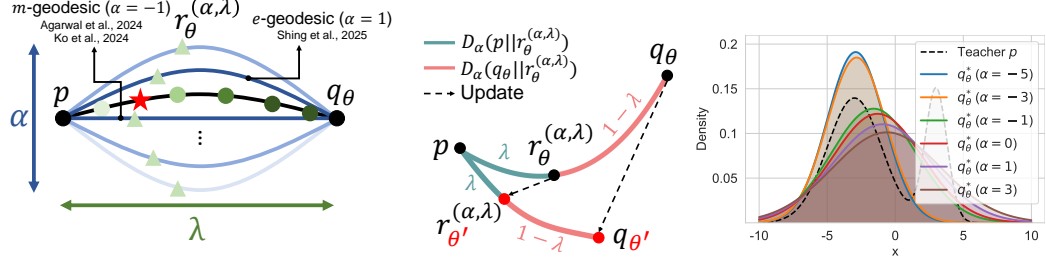

(a) Role of $\alpha$ and $\lambda$ in the distribution space. (b) Optimization dynamics. (c) Role of $\alpha$ for $q_\theta^*$.

Figure 3: Visualization of the characteristics of the $\alpha$-mixture distillation, AMiD. (a) $\alpha$ determines the geometry of interpolation while $\lambda$ controls the portion between $p$ and $q_\theta$. In general, the curvature of the path increases as $\alpha$ moves farther away from $-1$ (the straight line). (b) $r_\theta^{(\alpha,\lambda)}$ can be interpreted as the internal division point in terms of $\alpha$-divergence. Due to the uniqueness, the updated parameter $\theta'$ via optimization of AMiD, directly affects the student distribution. (c) Toy experiment with two-modal $p$ and uni-modal $q_\theta$. $\alpha$ controls the property of the optimized student distribution $q_\theta^*$ between the mode-covering and mode-seeking, even though we minimize the fixed divergence $D_{\mathrm{KL}}(p\|r_\theta^{(\alpha,\lambda)})$. In particular, when $\alpha \le 1$, increasing $\alpha$ encourages $q_\theta^*$ to exhibit more mode-covering behavior, whereas using smaller $\alpha$ strengthens mode-seeking behavior.

## 3.3 AMiD: Knowledge distillation with $\alpha$-mixture assistant distribution

In this section, we present a token-level KD for LLMs with $\alpha$-mixture assistant distribution, coined as $\underline{\alpha}$-mixture distillation (AMiD), which aims to align $r_\theta^{(\alpha,\lambda)}$ and either $p$ or $q_\theta$. Specifically, the optimization of AMiD is defined as follows, similar to Eq. (1):

$$\min_\theta \mathbb{E}_{(x,y)\sim\mathcal{D}}\left[\sum_{l=1}^{L} D(p, r_\theta^{(\alpha,\lambda)})\right] \quad \text{or} \quad \min_\theta \mathbb{E}_{(x,y)\sim\mathcal{D}}\left[\sum_{l=1}^{L} D(q_\theta, r_\theta^{(\alpha,\lambda)})\right] \quad (10)$$

We highlight that AMiD allows the use of arbitrary divergence $D$ and any dataset $\mathcal{D}$ (see Section 2.1) since $r_\theta^{(\alpha,\lambda)}$ is a valid distribution. Furthermore, AMiD generalizes the optimization schemes of prior research by extending both the assistant distribution and the divergence. For example, DistiLLM (Ko et al., 2024) corresponds to $D_{\mathrm{KL}}(p\|r_\theta^{(-1,\lambda)})$ and $D_{\mathrm{KL}}(q_\theta\|r_\theta^{(-1,\lambda)})$; and TAID (Shing et al., 2025) corresponds to $D_{KL}(r_\theta^{(1,\lambda)}\|q_\theta)$. Next, we aim to characterize the optimality of AMiD.

**Theorem 3.4.** *(Optimality) Let $D$ be any proper divergence and $\alpha \in \mathbb{R}$.*

- *If $\lambda \in [0, 1)$ and $\exists \theta$ s.t. $D(p, r_\theta^{(\alpha,\lambda)}) = 0$, then $D(p, r_\theta^{(\alpha,\lambda)}) = 0$ if and only if $p = q_\theta$.*

- *If $\lambda \in (0, 1]$ and $\exists \theta$ s.t. $D(q_\theta, r_\theta^{(\alpha,\lambda)}) = 0$, then $D(q_\theta, r_\theta^{(\alpha,\lambda)}) = 0$ if and only if $p = q_\theta$.*

Please refer to Appendix A.4 for the proof. Theorem 3.4 demonstrates that even if we minimize the divergence between $p$ (or $q_\theta$) and $r_\theta^{(\alpha,\lambda)}$, the primary goal of KD is guaranteed i.e., $p = q_\theta$. It is intuitive because the interpolation point needs to coincide with one of the endpoints when it coincides with the other (see Figure 3b). Therefore, leveraging the benefits of the assistant distribution, we establish optimality. Although Theorem 3.4 establishes theoretical optimality for any choice of $D$, $\alpha \in \mathbb{R}$, and $\lambda \in (0, 1)$, the effectiveness of AMiD might depend on selecting an appropriate $\alpha$ value due to the imperfect practical optimization. Please refer to Appendix B.3 for the theoretical insights based $\alpha$ tuning guidelines, Appendix C.4 for overlap-based adaptive $\alpha$ scheduling.

Now, we provide the gradient analysis to investigate the specific role of $\alpha$. In particular, we consider $f$-divergence, which is widely used in many areas, including KD for LLMs.

**Proposition 3.5.** *(Gradient analysis) The gradient of $f$-divergence $D_f(p\|r_\theta^{(\alpha,\lambda)})$ be expressed as:*

$$\nabla_\theta D_f\left(p\|r_\theta^{(\alpha,\lambda)}\right) = \mathbb{E}_{r_\theta^{(\alpha,\lambda)}}\left[w \cdot \left\{\psi_f\left(\frac{p}{r_\theta^{(\alpha,\lambda)}}\right) - \mathbb{E}_{r_\theta^{(\alpha,\lambda)}}\left[\psi_f\left(\frac{p}{r_\theta^{(\alpha,\lambda)}}\right)\right]\right\} \cdot \nabla_\theta \log q_\theta\right] \quad (11)$$

*where $w := \dfrac{(1-\lambda)q_\theta^{\frac{1-\alpha}{2}}}{\lambda p^{\frac{1-\alpha}{2}} + (1-\lambda)q_\theta^{\frac{1-\alpha}{2}}}$ and $\psi_f(v) := f(v) - vf'(v)$.*

Table 1: ROUGE-L scores (↑) on five task-agnostic instruction-following datasets. **Bold** and Underline mean the best and second-best performance of each column, except the teacher, respectively. All results are based on our own re-implementation. We use $D_{AB}$ and $\lambda = 0.1$ for AMiD. We conduct the evaluation with five random seeds. More results of baselines are in Appendix C.1.

| Model | Val. (↑) | Dolly Eval (↑) | Self Inst (↑) | Vicuna (↑) | Super NI (↑) | UnNI (↑) | Avg. (↑) |
|---|---|---|---|---|---|---|---|
| GPT-2 XL (Teacher) | − | 27.14 ±0.15 | 14.55 ±0.82 | 16.12 ±0.31 | 27.21 ±0.25 | 31.41 ±0.06 | 23.29 |
| ***GPT-2 XL (1.5B) → GPT-2 (0.1B)*** | | | | | | | |
| GKD | 27.06 | 24.58 ±0.13 | 11.78 ±0.44 | 14.60 ±0.37 | 22.84 ±0.12 | 25.04 ±0.09 | 19.77 |
| TAID | 28.37 | 25.74 ±0.27 | 12.91 ±0.31 | 17.09 ±0.18 | 23.66 ±0.31 | 26.82 ±0.05 | 21.24 |
| DistiLLM (SKL) | 27.88 | 25.50 ±0.28 | 12.35 ±0.39 | 16.10 ±0.22 | 23.87 ±0.39 | 26.16 ±0.06 | 20.80 |
| DistiLLM (SRKL) | 28.21 | 25.74 ±0.20 | 12.13 ±0.23 | 16.34 ±0.15 | 25.40 ±0.10 | 26.91 ±0.12 | 21.30 |
| ABKD | 28.61 | 25.49 ±0.24 | 12.52 ±0.52 | **17.36** ±0.55 | 26.07 ±0.14 | 27.36 ±0.10 | 21.76 |
| **AMiD (Ours)** | **29.24** | **26.44** ±0.12 | **13.74** ±0.49 | 16.76 ±0.24 | **29.71** ±0.08 | **30.35** ±0.09 | **23.40** |
| ***GPT-2 XL (1.5B) → GPT-2 Medium (0.3B)*** | | | | | | | |
| GKD | 27.90 | 25.06 ±0.55 | 12.36 ±0.42 | 15.71 ±0.58 | 23.83 ±0.26 | 27.14 ±0.09 | 20.82 |
| TAID | 29.45 | 27.01 ±0.27 | 14.53 ±0.47 | 17.58 ±0.20 | 25.14 ±0.15 | 29.79 ±0.14 | 22.81 |
| DistiLLM (SKL) | 29.65 | 26.87 ±0.13 | 14.11 ±0.29 | 16.85 ±0.54 | 25.59 ±0.22 | 28.84 ±0.03 | 22.45 |
| DistiLLM (SRKL) | 29.72 | 26.50 ±0.20 | 13.79 ±0.71 | 17.14 ±0.52 | 26.25 ±0.11 | 29.31 ±0.16 | 22.60 |
| ABKD | 29.64 | 26.93 ±0.17 | 13.69 ±0.32 | 17.45 ±0.27 | 28.15 ±0.18 | 30.94 ±0.06 | 23.43 |
| **AMiD (Ours)** | **30.83** | **27.34** ±0.18 | **15.26** ±0.46 | **17.69** ±0.27 | **29.04** ±0.20 | **33.15** ±0.13 | **24.50** |
| ***GPT-2 XL (1.5B) → GPT-2 Large (0.8B)*** | | | | | | | |
| GKD | 29.36 | 26.38 ±0.24 | 14.44 ±0.66 | 17.02 ±0.46 | 26.64 ±0.16 | 30.99 ±0.13 | 23.09 |
| TAID | 29.83 | 26.85 ±0.32 | 15.07 ±0.31 | 17.02 ±0.48 | 26.71 ±0.23 | 31.09 ±0.17 | 23.35 |
| DistiLLM (SKL) | 29.69 | 26.12 ±0.27 | 15.69 ±0.75 | 16.91 ±0.43 | 27.23 ±0.18 | 30.73 ±0.12 | 23.34 |
| DistiLLM (SRKL) | 30.59 | 27.09 ±0.40 | 14.61 ±0.66 | 16.39 ±0.27 | 28.44 ±0.45 | 31.04 ±0.06 | 23.51 |
| ABKD | 30.49 | 27.67 ±0.34 | 15.46 ±0.81 | **17.43** ±0.25 | 30.74 ±0.22 | 33.11 ±0.15 | 24.88 |
| **AMiD (Ours)** | **31.10** | **27.86** ±0.29 | **16.46** ±0.41 | 16.62 ±0.50 | **32.64** ±0.26 | **35.64** ±0.07 | **25.84** |

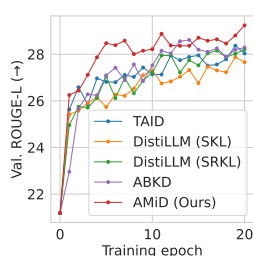

Figure 4: ROUGE-L curve on Dolly dataset.

Table 2: Experimental results on the task-specific distillation. "Trans." and "Summ." indicate translation and summarization task, respectively. We use $D_{KL}$ and $\lambda = 0.1$. $q_\theta$ indicates the no-assistant baseline. $\alpha = -1$ and $\alpha = 1$ correspond to the assistant distribution employed in DistiLLM and TAID, respectively.

| Model | SFTed Gemma-7B-It → Gemma-2B-It | | | SFTed Qwen2-7B-It to Qwen2-0.5B-It | | |
|---|---|---|---|---|---|---|
| | Trans. COMET (↑) | Summ. ROUGE-L (↑) | GSM8K. Acc (↑) | Trans. COMET (↑) | Summ. ROUGE-L (↑) | GSM8K Acc (↑) |
| $q_\theta$ | 74.21 | 34.88 | 24.26 | 58.07 | 31.67 | 33.13 |
| AMiD ($\alpha = -1$) | 52.83 | 26.51 | 00.00 | 57.23 | 32.27 | 35.63 |
| AMiD ($\alpha = 1$) | 74.20 | 34.93 | 24.49 | 58.17 | 31.65 | 33.28 |
| **AMiD ($\alpha \neq \pm 1$)** | **74.78** | **35.22** | **24.94** | **58.31** | **32.51** | **36.24** |

Please refer to Appendix A.5 for the proof. Proposition 3.5 implies the following properties. First, $w$ controls the magnitude of the instance-wise gradient $\nabla_\theta \log q_\theta(y_l \mid x, y_{<l})$. While $w$ does not affect the individual gradient direction due to $0 \leq w \leq 1$, its weighting effect may shift the batch-wise gradient direction. Second, the $\alpha$ plays a crucial role in enabling $w$ to perform instance-wise weighting based on the density ratio $\frac{p(y_l \mid x, y_{<l})}{q_\theta(y_l \mid x, y_{<l})}$. This property originates from the unique characteristic of the $\alpha$-mixture assistant distribution that cannot be achieved by $\lambda$ or learning rate scheduling. Third, $\alpha$ (relatively) adjusts the *mode-covering* and *mode-seeking* behavior of the optimized student distribution. Let us assume $\frac{1-\alpha}{2} \geq 0$, i.e., $\alpha \leq 1$.[2] When $p \geq q_\theta$, a larger $\alpha$ produces a correspondingly larger value of $w$. Thus, it amplifies the gradient magnitude in regions where the student underestimates the teacher. As a result, choosing a large $\alpha$ (relatively) encourages the student distribution $q_\theta$ to exhibit a *mode-covering* behavior. In contrast, employing the small $\alpha$ in $p < q_\theta$ results in a large $w$ value. It assigns a large gradient magnitude to the area where the student overestimates, ultimately exhibiting that the small $\alpha$ (relatively) reinforces *mode-seeking* property.

To support the results from the gradient analysis, we investigate the property of the optimized student distribution $q_\theta^*$ through the toy experiments. Figure 3c shows that the optimized student distribution $q_\theta^*$ with small $\alpha$ converges to one of the peak, which indicates the mode-seeking. As increasing $\alpha$, the $q_\theta^*$ gradually have thick tails while moving towards the average of $p$, which implies the mode-covering. These analyses indicate that the balance between mode-covering and mode-seeking, often attributed to divergence selection, might be controlled by $\alpha$ in the $\alpha$-mixture assistant distribution.

[2]Although $\alpha$ can be any real number, we assume $\alpha \leq 1$ for the sake of simplicity.

Table 3: ROUGE-L scores (↑) with various divergences $D$ and $\alpha$. We utilize GPT-2 XL (1.5B) → GPT-2 (0.1B). We use $\lambda = 0.1$ for AMiD. $q_\theta$ indicates the no-assistant baseline. $\alpha = -1$ and $\alpha = 1$ correspond to the assistant distribution employed in DistiLLM and TAID, respectively.

| Divergence $D$ | Assistant $r_\theta^{(\alpha,\lambda)}$ | Val. (↑) | Dolly Eval (↑) | Self Inst (↑) | Vicuna (↑) | Super NI (↑) | UnNI (↑) | Avg. (↑) |
|---|---|---|---|---|---|---|---|---|
| $D_{\mathrm{KL}}(p\|r_\theta^{(\alpha,\lambda)})$ | $q_\theta$ | 25.25 | 22.96 ±0.23 | 10.54 ±0.14 | 15.33 ±0.13 | 18.10 ±0.26 | 21.10 ±0.16 | 17.61 |
| | **AMiD ($\alpha = -5.0$)** | **28.99** | **25.86** ±0.10 | **13.72** ±0.42 | 15.90 ±0.29 | **28.32** ±0.24 | **29.52** ±0.06 | **22.66** |
| | AMiD ($\alpha = -3.0$) | 28.47 | 25.72 ±0.17 | 13.68 ±0.19 | **16.71** ±0.30 | 27.30 ±0.30 | 29.03 ±0.12 | 22.49 |
| | AMiD ($\alpha = -1.0$) | 27.88 | 25.50 ±0.28 | 12.35 ±0.39 | 16.10 ±0.22 | 23.87 ±0.39 | 26.16 ±0.06 | 20.80 |
| | AMiD ($\alpha = -0.5$) | 27.37 | 24.17 ±0.37 | 12.15 ±0.49 | 16.37 ±0.38 | 24.34 ±0.20 | 24.36 ±0.06 | 20.28 |
| | AMiD ($\alpha = 0.0$) | 26.37 | 24.08 ±0.25 | 10.65 ±0.20 | 16.27 ±0.24 | 20.09 ±0.20 | 22.71 ±0.13 | 18.76 |
| | AMiD ($\alpha = 0.5$) | 25.56 | 22.81 ±0.22 | 10.77 ±0.40 | 16.24 ±0.23 | 18.96 ±0.45 | 22.13 ±0.10 | 18.18 |
| | AMiD ($\alpha = 1.0$) | 25.31 | 22.99 ±0.12 | 11.17 ±0.51 | 15.97 ±0.46 | 18.74 ±0.40 | 21.94 ±0.16 | 18.16 |
| $D_{\mathrm{RKL}}(p\|r_\theta^{(\alpha,\lambda)})$ | $q_\theta$ | 28.85 | **26.67** ±0.50 | 12.32 ±0.43 | 17.48 ±0.30 | 24.25 ±0.19 | 26.56 ±0.19 | 21.46 |
| | AMiD ($\alpha = -5.0$) | 28.39 | 26.16 ±0.33 | **12.95** ±0.57 | 17.39 ±0.39 | 24.59 ±0.22 | 27.17 ±0.09 | 21.65 |
| | **AMiD ($\alpha = -3.0$)** | 28.75 | 26.47 ±0.12 | 12.71 ±0.21 | 17.17 ±0.64 | **27.00** ±0.11 | **28.16** ±0.12 | **22.30** |
| | AMiD ($\alpha = -1.0$) | **28.97** | 22.96 ±0.23 | 12.34 ±0.24 | 17.27 ±0.64 | 22.44 ±0.36 | 25.68 ±0.10 | 20.83 |
| | AMiD ($\alpha = -0.5$) | 28.25 | 26.15 ±0.30 | 11.81 ±0.29 | 16.54 ±0.22 | 23.49 ±0.25 | 25.87 ±0.06 | 20.77 |
| | AMiD ($\alpha = 0.0$) | 28.80 | 25.84 ±0.22 | 12.06 ±0.13 | **17.71** ±0.65 | 22.72 ±0.24 | 25.36 ±0.10 | 20.74 |
| | AMiD ($\alpha = 0.5$) | 28.45 | 25.42 ±0.11 | 11.45 ±0.26 | 17.31 ±0.38 | 21.58 ±0.24 | 24.43 ±0.10 | 20.04 |
| | AMiD ($\alpha = 1.0$) | 0.16 | 4.27 ±0.01 | 2.81 ±0.02 | 9.12 ±0.06 | 1.64 ±0.00 | 1.84 ±0.0 | 3.94 |
| $D_{\mathrm{AB}}(p\|r_\theta^{(\alpha,\lambda)})$ | $q_\theta$ | 28.61 | 25.49 ±0.24 | 12.52 ±0.52 | 17.36 ±0.55 | 26.07 ±0.14 | 27.36 ±0.10 | 21.76 |
| | **AMiD ($\alpha = -5.0$)** | **29.24** | **26.44** ±0.12 | **13.74** ±0.49 | 16.76 ±0.24 | **29.71** ±0.08 | **30.35** ±0.09 | **23.40** |
| | AMiD ($\alpha = -3.0$) | 29.07 | 26.38 ±0.18 | 13.58 ±0.57 | 16.11 ±0.18 | 29.27 ±0.14 | 30.14 ±0.06 | 23.10 |
| | AMiD ($\alpha = -1.0$) | 28.70 | 26.10 ±0.24 | 13.34 ±0.25 | 16.71 ±0.27 | 26.55 ±0.17 | 29.55 ±0.11 | 22.45 |
| | AMiD ($\alpha = -0.5$) | 28.70 | 26.37 ±0.27 | 13.59 ±0.25 | **17.02** ±0.34 | 27.06 ±0.31 | 28.50 ±0.16 | 22.51 |
| | AMiD ($\alpha = 0.0$) | 28.86 | 25.77 ±0.34 | 13.57 ±0.22 | 16.14 ±0.36 | 27.26 ±0.27 | 28.52 ±0.20 | 22.25 |
| | AMiD ($\alpha = 0.5$) | 28.46 | 25.80 ±0.32 | 12.94 ±0.36 | 16.59 ±0.34 | 26.29 ±0.22 | 27.73 ±0.08 | 21.87 |
| | AMiD ($\alpha = 1.0$) | 24.93 | 22.36 ±0.22 | 9.72 ±0.58 | 16.29 ±0.30 | 15.09 ±0.19 | 16.15 ±0.12 | 15.92 |

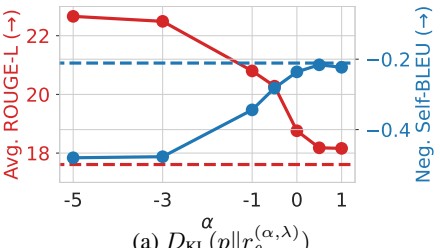
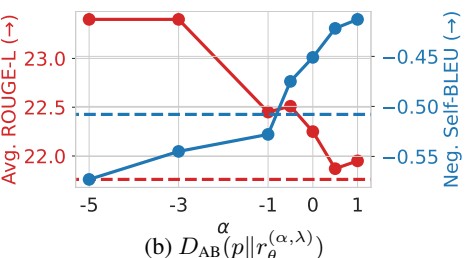

(a) $D_{\mathrm{KL}}(p\|r_\theta^{(\alpha,\lambda)})$ (b) $D_{\mathrm{AB}}(p\|r_\theta^{(\alpha,\lambda)})$

Figure 5: Performance curve on ROUGE-L (quality) and Self-BLEU (diversity). Colored dashed lines: no-assistant baseline ($q_\theta$). We use $\lambda = 0.1$.

## 4 EXPERIMENTS

We consider both general instruction-following distillation and task-specific distillation to validate the effectiveness of AMiD. AMiD is primarily compared with methodologies, which inherently include the assistant distribution, such as GKD (Agarwal et al., 2024), DistiLLM (Ko et al., 2024), TAID (Shing et al., 2025); and the state-of-the-art ABKD (Wang et al., 2025). Please refer to Table 15 in Appendix C for further performance comparisons with additional baselines. We use $\alpha_{AB}$-$\beta_{AB}$-divergence with $\alpha_{AB} = 0.2, \beta_{AB} = 0.7$ and adopt adaptive off-policy training (Ko et al., 2024) as default. We explore $\alpha$ over the range $\{-5, -3, -1, -0.5, 0, 0.5, 1.0\}$ and employ $\lambda = 0.1$ as default by following prior work (Ko et al., 2024). Additional details on datasets, models, and training details are provided in Appendix B.

### 4.1 PERFORMANCE COMPARISON

**Instruction-following Experiments.** Table 1 reports results on the GPT-2 family (Radford et al., 2019) across different model sizes. AMiD consistently achieves the best performance in most evaluation settings, surpassing prior methods such as GKD, TAID, and DistiLLM, which also exploit assistant distributions. Notably, AMiD delivers substantial gains on SuperNI and UnNI, benchmarks requiring generalization to diverse and unseen instructions (Wang et al., 2022). These improvements suggest that AMiD promotes superior mode coverage and distributional alignment, thereby enhancing out-of-distribution generalization. Even when the capacity gap narrows for larger models (e.g., GPT-2 Large), AMiD continues to yield significant improvements, demonstrating that the $\alpha$-mixture assistant distribution benefits not only small students but also stronger ones, thereby validating its scalability and robustness. Figure 4 further shows that AMiD consistently envelopes the baseline's

Table 4: ROUGE-L scores (↑) with various SGOs. We utilize GPT-2 XL (1.5B) → GPT-2 (0.1B). We use $\lambda = 0.1$ for these experiments. $q_\theta$ indicates no-assistant baseline ABKD for these experiments. We use $D_{AB}$ for AMiD. $\alpha = -1$ and $\alpha = 1$ correspond to the assistant distribution employed in DistiLLM and TAID, respectively.

| Dataset $\mathcal{D}$ | Assistant $r_\theta^{(\alpha,\lambda)}$ | Val. (↑) | Dolly Eval (↑) | Self Inst (↑) | Vicuna (↑) | Super NI (↑) | UnNI (↑) | Avg. (↑) |
|---|---|---|---|---|---|---|---|---|
| Fixed (Hinton et al., 2015) | $q_\theta$ | 27.06 | 24.81 ±0.28 | 11.25 ±0.26 | 15.05 ±0.21 | 21.78 ±0.15 | 23.73 ±0.09 | 19.32 |
| | AMiD ($\alpha = -1$) | 27.35 | **25.34** ±0.28 | 11.54 ±0.26 | 15.34 ±0.30 | 21.77 ±0.19 | 24.45 ±0.06 | 19.69 |
| | AMiD ($\alpha = 1$) | 27.09 | 24.36 ±0.34 | 12.40 ±0.41 | 14.37 ±0.31 | 24.36 ±0.28 | 26.28 ±0.05 | 20.35 |
| | **AMiD ($\alpha \neq \pm1$)** | **27.84** | 25.32 ±0.15 | **13.44** ±0.41 | **15.44** ±0.11 | **27.03** ±0.12 | **28.19** ±0.06 | **21.88** |
| On-policy (Lin et al., 2020) | $q_\theta$ | 28.25 | 25.70 ±0.20 | 13.03 ±0.17 | 16.86 ±0.34 | 24.67 ±0.16 | 27.47 ±0.15 | 21.55 |
| | AMiD ($\alpha = -1$) | 28.60 | 25.43 ±0.34 | 12.96 ±0.58 | 16.59 ±0.39 | 27.35 ±0.16 | 29.93 ±0.07 | 22.45 |
| | AMiD ($\alpha = 1$) | 28.60 | 25.12 ±0.55 | 13.28 ±0.38 | 17.08 ±0.52 | 24.35 ±0.23 | 26.94 ±0.19 | 21.35 |
| | **AMiD ($\alpha \neq \pm1$)** | **28.90** | **26.22** ±0.31 | **14.31** ±0.22 | **17.37** ±0.22 | **28.59** ±0.27 | **31.00** ±0.08 | **23.50** |
| Mixed (Agarwal et al., 2024) | $q_\theta$ | 29.08 | 25.67 ±0.16 | 12.38 ±0.29 | 17.15 ±0.52 | 22.98 ±0.26 | 26.20 ±0.14 | 20.88 |
| | AMiD ($\alpha = -1$) | 28.79 | 25.65 ±0.25 | 11.98 ±0.28 | 16.94 ±0.13 | 23.82 ±0.17 | 26.25 ±0.06 | 20.93 |
| | AMiD ($\alpha = 1$) | 28.06 | 25.68 ±0.39 | 12.81 ±0.20 | 16.97 ±0.27 | 24.91 ±0.21 | 26.52 ±0.08 | 21.38 |
| | **AMiD ($\alpha \neq \pm1$)** | **29.24** | **26.46** ±0.16 | **13.62** ±0.27 | 16.91 ±0.30 | **28.13** ±0.06 | **29.39** ±0.07 | **22.90** |
| Adaptive off-policy (Ko et al., 2024) | $q_\theta$ | 28.61 | 25.49 ±0.24 | 12.52 ±0.52 | **17.36** ±0.55 | 26.07 ±0.14 | 27.36 ±0.10 | 21.76 |
| | AMiD ($\alpha = -1$) | 28.70 | 26.10 ±0.24 | 13.34 ±0.25 | 16.71 ±0.27 | 26.55 ±0.17 | 29.55 ±0.11 | 22.45 |
| | AMiD ($\alpha = 1$) | 27.80 | 25.78 ±0.44 | 13.74 ±0.19 | 16.42 ±0.22 | 26.04 ±0.22 | 27.79 ±0.09 | 21.95 |
| | **AMiD ($\alpha \neq \pm1$)** | **29.24** | **26.44** ±0.12 | 13.74 ±0.49 | 16.76 ±0.24 | **29.71** ±0.08 | **30.35** ±0.09 | **23.40** |

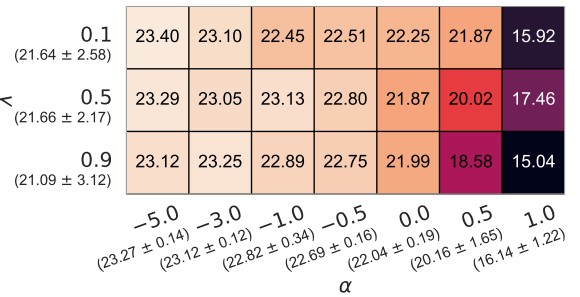

Figure 6: Relationship between $\alpha$ and $\lambda$ under $D_{AB}$.

Table 5: Win Rate (WR, %) scores on three instruction-following benchmarks. We utilize Qwen2.5-14B-Instruct as a teacher and Qwen2.5-1.5B-Instruct as a student.

| Model | AlpacaEval WR (↑) | Evol-Inst WR (↑) | UltraFeed WR (↑) | Avg. (↑) |
|---|---|---|---|---|
| Teacher | 95.7 | 92.2 | 84.9 | 90.9 |
| Student | 64.3 | 47.2 | 40.4 | 50.6 |
| DistiLLM-2 ($\alpha = -1$) | 80.2 | 70.4 | 61.7 | 70.8 |
| DistiLLM-2 ($\alpha = -5$) | **81.3** | **71.1** | **63.6** | **72.0** |

validation ROUGE-L curve, indicating both efficient and stable optimization. Additional comparisons with other baselines and experiments on OpenLLaMA2 are provided in Appendix C.

**Task-specific Experiments.** To further investigate the effectiveness of AMiD, we consider various task-specific distillation, such as translation, summarization, and reasoning tasks. We adopt the implementation of SKD (Xu et al., 2025) and employ the fixed dataset strategy. As shown in Table 2, using an assistant distribution achieves higher performance compared to the no-assistant baseline ABKD. Nevertheless, in the previous framework, where only $\alpha = \pm1$ is available, it exhibits mixed performance depending on the task and network. Our proposed framework, AMiD, which extends the range of $\alpha$, allows us to discover the high-performance assistant distribution. This generalization leads to consistent improvements over the baselines, achieving the best performance on all tasks.

## 4.2 ABLATION STUDY AND ADDITIONAL ANALYSIS

**Balancing Mode-covering and Mode-seeking via $\alpha$.** As mentioned in Section 3.3, we have demonstrated that even when employing the same divergence, adjusting the $\alpha$ of the $\alpha$-mixture assistant distribution allows us to control the mode-covering or mode-seeking property of the optimized student distribution. To further substantiate this analysis, we examine the trends of ROUGE-L, representing quality, and (Negative) Self-BLEU, representing diversity, by adjusting $\alpha$ with a fixed divergence. Figure 5 exhibits a clear quality-diversity trade-off for both KL divergence $D_{KL}$ and $\alpha$-$\beta$ divergence $D_{AB}$. Specifically, as $\alpha$ increases, quality decreases and diversity increases, supporting the theoretical analysis that the mode-covering property is enhanced. Decreasing $\alpha$ shows the opposite effect, aligning with the mode of teacher distribution, which indicates mode-seeking. These results demonstrate that, even under a fixed divergence, $\alpha$ serves as an effective control knob to balance quality and diversity.

**Relationship between $\alpha$ and $\lambda$.** To further examine the relationship between $\alpha$ and $\lambda$, we conduct the experiments on $\lambda = 0.1, 0.5, 0.9$ with $D_{AB}(p||r_\theta^{(\alpha,\lambda)})$. Figure 6 presents the average performance of five instruction-following datasets among the various $\alpha$ and $\lambda$ combinations. The analyses of experimental results are as follows: (1) Across all tested values of $\lambda$, using a smaller $\alpha$ consistently achieves higher performance. This observation aligns with our theoretical analysis, indicating

Table 6: Performances on instruction-following, code generation, and mathematical reasoning. We utilize the Qwen2.5-Instruct models, which are appropriate for each task, employing a 7B model as the teacher and a 1.5B model as the student. For evaluation, we employ LLM-as-a-Judge (Zheng et al., 2023) with GPT-4o for AlpacaEval and Evol-Instruct, GPT-4o-mini for UltraFeedback.

| Model | Instruction-following | | | | Code generation | | | Math reasoning |
|---|---|---|---|---|---|---|---|---|
| | AlpacaEval WR (↑) | Evol-Inst WR (↑) | UltraFeed WR (↑) | Avg. WR (↑) | HumanEval pass@1 (↑) | MBPP pass@1 (↑) | Avg. pass@1 (↑) | GSM8K (↑) |
| Teacher | 93.7 | 89.6 | 80.8 | 88.0 | 90.9 | 83.1 | 87.0 | 89.3 |
| Student | 64.2 | 46.2 | 40.0 | 50.1 | 70.7 | 69.3 | 70.0 | 74.3 |
| DistiLLM-2 ($\alpha = -1$) | 79.5 | 69.0 | 62.6 | 70.4 | 72.0 | **74.6** | 73.3 | 76.9 |
| DistiLLM-2 ($\alpha = -5$) | **80.7** | **71.0** | **63.3** | **71.7** | **73.2** | 73.5 | **73.4** | **77.4** |

that a smaller $\alpha$ relatively induces a more mode-seeking behavior. (2) When $\lambda$ is too large ($\lambda = 0.9$), performance slightly degrades and exhibits a larger standard deviation. We attribute this to the assistant distribution being overly close to the teacher distribution, which 1) limits effective knowledge transfer and 2) makes the optimization more sensitive to curvature variations. In contrast, $\lambda = 0.5$ achieves both high and stable performance, demonstrating that choosing a midpoint provides robustness against curvature changes. (3) Compared to $\lambda$, the performance is generally more robust to changes in $\alpha$, as indicated by lower standard deviations. However, the $\alpha$ values close to 1 show noticeably higher standard deviations. We conjecture that this instability arises because the change of mixing coefficient ($\lambda$) between the teacher and student distributions makes hard mode-covering.

**Compatibility with Divergences.** As mentioned in Section 3.3, AMiD allows the use of the arbitrary divergence $D$ since the $\alpha$-mixture assistant distribution $r_\theta^{(\alpha,\lambda)}$ is a valid distribution. Therefore, we conduct performance comparisons across various combinations of divergences and $\alpha$ values to demonstrate the versatility of AMiD. In Table 3, we observed that AMiD generally achieves higher performance than the no-assistant baseline regardless of divergence in most settings. Notably, the highest-performing $\alpha$ was discovered in $\alpha \neq \pm 1$ regions beyond the limited scope of prior works, and its values are generally small. These results confirm the universality of AMiD to generic divergences, representing its wide adaptiveness and flexibility. Please refer to Appendix C.3 for the student-assistant cases $D(q_\theta, r_\theta^{(\alpha,\lambda)})$.

**Universality to SGOs.** AMiD is a generalized framework from the view of assistant distribution $r_\theta^{(\alpha,\lambda)}$ and divergence $D$, and therefore is not constrained by the dataset $\mathcal{D}$. In Table 4, we confirm the universality of AMiD across various student-generated output (SGO) strategies. AMiD ($\alpha \neq \pm 1$) outperforms the no-assistant baseline and previous mixtures ($\alpha = \pm 1$) by a significant margin across almost all metrics. These results indicate that AMiD is compatible with diverse SGO pipelines and remains effective regardless of how the datasets are collected.

**Cooperation with Contrastive-based Distillation.** The proposed $\alpha$-mixture assistant distribution $r_\theta^{(\alpha,\lambda)}$ is a theoretically valid probability distribution for any $\alpha$ and $\lambda$. Therefore, the $r_\theta^{(\alpha,\lambda)}$ can be combined with the contrastive-based distillation methods beyond the divergence-based distillation methods. To validate this applicability, we replace the assistant distribution of DistiLLM-2 (Ko et al., 2025), which utilizes $m$-mixture ($\alpha = -1$), with our $\alpha$-mixture assistant distribution. As shown in Table 6, incorporating our $\alpha$-mixture assistant distribution exhibits further performance improvements or competitive performances over the base contrastive method, DistiLLM-2.

**Scalability to Large Teacher.** To demonstrate the extensibility, we conduct an experiment by distilling Qwen2.5-14B-Instruct into Qwen2.5-1.5B-Instruct under the instruction-following task. As shown in Table 5, integrating $\alpha$-mixture assistant distribution exhibits performance improvements over the DistiLLM-2. These results indicate that AMiD remains effective in a large teacher model.

## 5 CONCLUSION

This work introduces a unified framework for KD in LLMs by proposing the $\alpha$-mixture assistant distribution and the corresponding distillation method, AMiD. Our approach systematically generalizes previous fragmented methods and enables flexible interpolation between teacher and student. Theoretical and empirical analyses congruently demonstrate that the design parameter $\alpha$ controls the mode-seeking vs. mode-covering behavior. AMiD consistently outperforms prior KD methods across diverse settings and establishes a new foundation for assistant-guided KD for LLMs.

ACKNOWLEDGMENTS

This work was supported by the IITP (Institute of Information & Communications Technology Planning & Evaluation)-ITRC (Information Technology Research Center) grant funded by the Korea government (Ministry of Science and ICT) (IITP-2026-RS-2024-00437268), with a 50% contribution rate. This research was also supported by AI Technology Development for Commonsense Extraction, Reasoning, and Inference from Heterogeneous Data(IITP) funded by the Ministry of Science and ICT(RS-2022-II220077), with a 50% contribution rate.

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

## A DEFINITION AND PROOF

### A.1 AMARI'S $\alpha$-DIVERGENCE

**Definition.** Let $\alpha \in \mathbb{R}$ be a real parameter. The $\alpha$-function $f_\alpha(u)$ is defined for $\alpha \neq \pm 1$ by

$$f_\alpha(u) = \frac{4}{1 - \alpha^2} \left( 1 - u^{\frac{1+\alpha}{2}} \right). \tag{12}$$

Using this $\alpha$-function, the $\alpha$-divergence between two probability distributions $p = \{p_i\}$ and $q = \{q_i\}$ is defined as

$$D_\alpha[p||q] = \frac{4}{1 - \alpha^2} \left( 1 - \sum_i p_i^{\frac{1+\alpha}{2}} q_i^{\frac{1-\alpha}{2}} \right), \qquad \alpha \neq \pm 1. \tag{13}$$

The dual divergence is obtained by replacing $\alpha$ with $-\alpha$:

$$D_\alpha[p||q] = D_{-\alpha}[q||p]. \tag{14}$$

We also define the limiting cases at $\alpha = \pm 1$ by continuity. When $\alpha = \pm 1$, the $\alpha$-function becomes

$$f_\alpha(u) = \begin{cases} u \log u, & \alpha = 1, \\ -\log u, & \alpha = -1. \end{cases} \tag{15}$$

**Connection to KL and reverse KL.** Correspondingly, the $\alpha$-divergence reduces to

$$D_\alpha[p||q] = \begin{cases} \sum_i q_i \log \frac{q_i}{p_i}, & \alpha = 1, \\ \sum_i p_i \log \frac{p_i}{q_i}, & \alpha = -1, \end{cases} \tag{16}$$

which correspond to the KL divergence ($\alpha = -1$) and the reverse KL divergence ($\alpha = 1$).

**Relation to Rényi divergence.** Since the Amari $\alpha$-divergence includes KL and reverse KL as the limiting cases, it can be viewed as a one-parameter generalization of the KL divergence. Moreover, it is closely related to the Rényi divergence. If we define a reparameterized order

$$\beta = \frac{1 + \alpha}{2},$$

then the quantity appearing in the Amari divergence,

$$\sum_i p_i^{\frac{1+\alpha}{2}} q_i^{\frac{1-\alpha}{2}},$$

is exactly the same expression inside the discrete Rényi divergence,

$$R_\beta[p||q] = \frac{1}{\beta - 1} \log \left( \sum_i p_i^\beta q_i^{1-\beta} \right). \tag{17}$$

Thus, the Amari and Rényi divergences represent the same underlying one-parameter family through the invertible reparameterization $\beta = (1 + \alpha)/2$

### A.2 PROOF OF PROPOSITION 3.1

**Proposition 3.1.** *The assistant distribution of TAID (Shing et al., 2025) is e-mixture of $p$ and $q_\theta$:*[3]

$$r_\theta := softmax((1 - \lambda) \cdot logit(q_\theta) + \lambda \cdot logit(p)) \propto p^\lambda q_\theta^{1-\lambda} \tag{6}$$

---

[3]We omit the time index $t$ and detached notation for the sake of uniformity.

*Proof.* Note that $p = \text{softmax}(\text{logit}(p)) = \frac{1}{Z_p}\exp(\text{logit}(p))$ where $\text{logit}(p)$ is logit of $p$ and $Z_p$ is normalization constant of $p$. Therefore, $\text{logit}(p) = \log(p \cdot Z_p)$.

$$r = \text{softmax}(\lambda\text{logit}(p) + (1-\lambda)\text{logit}(q)) \tag{18}$$

$$= \frac{1}{Z_r}\exp(\lambda\text{logit}(p) + (1-\lambda)\text{logit}(q)) \tag{19}$$

$$= \frac{1}{Z_r}\exp(\lambda\log(pZ_p) + (1-\lambda)\log(qZ_q)) \tag{20}$$

$$= \frac{1}{Z_r}\exp\left(\log\left(p^\lambda q^{1-\lambda}\right) + \log\left(Z_p^\lambda Z_q^{1-\lambda}\right)\right) \tag{21}$$

$$= \frac{Z_p^\lambda Z_q^{1-\lambda}}{Z_r} p^\lambda q^{1-\lambda} \tag{22}$$

$$= \frac{1}{Z'} p^\lambda q^{1-\lambda}, \quad \text{where } Z' := \frac{Z_r}{Z_p^\lambda Z_q^{1-\lambda}}. \tag{23}$$

Now, it is sufficient to show that $Z' = \sum p^\lambda q^{1-\lambda}$:

$$Z' = \frac{1}{Z_p^\lambda Z_q^{1-\lambda}} \sum \exp(\lambda\text{logit}(p) + (1-\lambda)\text{logit}(q)) \tag{24}$$

$$= \sum \left(\frac{1}{Z_p}\exp(\text{logit}(p))\right)^\lambda \left(\frac{1}{Z_q}\exp(\text{logit}(q))\right)^{1-\lambda} \tag{25}$$

$$= \sum p^\lambda q^{1-\lambda}. \tag{26}$$

Therefore, $r_\theta \propto p^\lambda q^{1-\lambda}$ and it is a valid distribution with normalization. $\square$

## A.3 PROOF OF PROPOSITION 3.3

**Proposition 3.3.** *(Continuity) Assume that $p$ and $q_\theta$ are not both zero. Then, $r_\theta^{(\alpha,\lambda)}$ is continuous function w.r.t $\alpha$ under the fixed $\lambda \in [0,1]$.*

*Proof.* We begin with the proof for the continuity of the unnormalized assistant distribution $\tilde{r}_\theta^{(\alpha,\lambda)}$. The $\alpha \neq 1$ case is trivial since it is a composition of continuous functions. For the $\alpha = 1$ case, it is a well-known fact that the power mean is a continuous function (Bullen, 2013). Especially, we can show that as follows:

$$\lim_{\alpha\to 1} \log \tilde{r}_\theta^{(\alpha,\lambda)} = \lim_{\alpha\to 1} \frac{2}{1-\alpha} \log\left(\lambda p^{\frac{1-\alpha}{2}} + (1-\lambda) q^{\frac{1-\alpha}{2}}\right) \tag{27}$$

$$= \lim_{\alpha\to 1} \frac{\lambda p^{\frac{1-\alpha}{2}} \log p + (1-\lambda) q^{\frac{1-\alpha}{2}} \log q}{\lambda p^{\frac{1-\alpha}{2}} + (1-\lambda) q^{\frac{1-\alpha}{2}}} \tag{28}$$

$$= \lambda \log p + (1-\lambda) \log q \tag{29}$$

$$= \log\left(p^\lambda q^{1-\lambda}\right). \tag{30}$$

By the continuity of the exponential function, we can get $\lim_{\alpha\to 1} \tilde{r}_\theta^{(\alpha,\lambda)} = p^\lambda q^{1-\lambda}$. We use L'Hôpital's rule in the second equality. Note that $Z = \sum_i \tilde{r}_\theta^{(\alpha,\lambda)}(i)$ is continuous function w.r.t $\alpha$ since it is a finite sum of continuous functions w.r.t $\alpha$. Also, since $p$ and $q$ cannot be both zero, $\tilde{r}_\theta^{(\alpha,\lambda)}$ is not zero, so $Z > 0$. Therefore, the $r_\theta^{(\alpha,\lambda)} = \frac{1}{Z}\tilde{r}_\theta^{(\alpha,\lambda)}$ is contiuous function w.r.t. $\alpha$. $\square$

## A.4 PROOF OF THEOREM 3.4

**Theorem 3.4.** *(Optimality) Let $D$ be any proper divergence and $\alpha \in \mathbb{R}$.*

- If $\lambda \in [0, 1)$ and $\exists \theta$ s.t. $D(p, r_\theta^{(\alpha,\lambda)}) = 0$, then $D(p, r_\theta^{(\alpha,\lambda)}) = 0$ if and only if $p = q_\theta$.

- If $\lambda \in (0, 1]$ and $\exists \theta$ s.t. $D(q_\theta, r_\theta^{(\alpha,\lambda)}) = 0$, then $D(q_\theta, r_\theta^{(\alpha,\lambda)}) = 0$ if and only if $p = q_\theta$.

*Proof.* We first prove that $D(p, r_\theta^{(\alpha,\lambda)})$ implies $p = q_\theta$. By the definition of divergence, we have that $D(p, r_\theta^{(\alpha,\lambda)}) = 0$ if and only if $p = r_\theta^{(\alpha,\lambda)}$.

**Case $\alpha = 1$.** In this case, $r_\theta^{(\alpha,\lambda)} = \frac{1}{Z} p^\lambda q_\theta^{(1-\lambda)}$.

$$p = \frac{1}{Z} p^\lambda q_\theta^{1-\lambda} \Leftrightarrow Z p^{1-\lambda} = q_\theta^{1-\lambda} \Leftrightarrow Z^{\frac{1}{1-\lambda}} p = q_\theta \tag{31}$$

By integrating both sides, $Z^{\frac{1}{1-\lambda}} = 1$ which implies $Z = 1$. Therefore, $p = q_\theta$

**Case $\alpha \neq 1$.** In this case, $r_\theta^{(\alpha,\lambda)} = \frac{1}{Z} \left\{ \lambda p^{\frac{1-\alpha}{2}} + (1-\lambda) q_\theta^{\frac{1-\alpha}{2}} \right\}^{\frac{2}{1-\alpha}}$

$$p = \frac{1}{Z} \left\{ \lambda p^{\frac{1-\alpha}{2}} + (1-\lambda) q_\theta^{\frac{1-\alpha}{2}} \right\}^{\frac{2}{1-\alpha}} \Leftrightarrow Z^{\frac{1-\alpha}{2}} p^{\frac{1-\alpha}{2}} = \lambda p^{\frac{1-\alpha}{2}} + (1-\lambda) q_\theta^{\frac{1-\alpha}{2}} \tag{32}$$

$$\Leftrightarrow (Z^{\frac{1-\alpha}{2}} - \lambda) p^{\frac{1-\alpha}{2}} = (1-\lambda) q_\theta^{\frac{1-\alpha}{2}} \tag{33}$$

$$\Leftrightarrow C p^{\frac{1-\alpha}{2}} = q_\theta^{\frac{1-\alpha}{2}}, \quad \text{where} \quad C := \frac{Z^{\frac{1-\alpha}{2}} - \lambda}{1 - \lambda} \tag{34}$$

$$\Leftrightarrow C^{\frac{2}{1-\alpha}} p = q_\theta \tag{35}$$

By integrating both sides, $C^{\frac{2}{1-\alpha}} = 1$ which implies $C = 1$. Therefore, $p = q_\theta$.

$D(q_\theta, r_\theta^{(\alpha,\lambda)})$ is similar. $\qquad\square$

## A.5 PROOF OF PROPOSITION 3.5

**Proposition 3.5.** *(Gradient analysis) The gradient of $f$-divergence $D_f(p||r_\theta^{(\alpha,\lambda)})$ be expressed as:*

$$\nabla_\theta D_f\left(p||r_\theta^{(\alpha,\lambda)}\right) = \mathbb{E}_{r_\theta^{(\alpha,\lambda)}}\left[w \cdot \left\{\psi_f\left(\frac{p}{r_\theta^{(\alpha,\lambda)}}\right) - \mathbb{E}_{r_\theta^{(\alpha,\lambda)}}\left[\psi_f\left(\frac{p}{r_\theta^{(\alpha,\lambda)}}\right)\right]\right\} \cdot \nabla_\theta \log q_\theta\right] \tag{11}$$

*where $w := \frac{(1-\lambda)q_\theta^{\frac{1-\alpha}{2}}}{\lambda p^{\frac{1-\alpha}{2}} + (1-\lambda)q_\theta^{\frac{1-\alpha}{2}}}$ and $\psi_f(v) := f(v) - vf'(v)$.*

*Proof.* From the basic calculus, we can derive the following equation for fixed $p$:

$$\frac{\partial}{\partial r}\left[r f\left(\frac{p}{r}\right)\right] = f\left(\frac{p}{r}\right) - \frac{p}{r} f'\left(\frac{p}{r}\right) = \psi_f\left(\frac{p}{r}\right). \tag{36}$$

Hence,

$$\nabla_\theta D_f(p||r_\theta^{(\alpha,\lambda)}) = \sum_{y_t \in \mathcal{V}} \psi_f\left(\frac{p(y_l \mid x, y_{<l})}{r_\theta^{(\alpha,\lambda)}(y_l \mid x, y_{<l})}\right) \cdot \nabla_\theta r_\theta^{(\alpha,\lambda)}(y_l \mid x, y_{<l}) \tag{37}$$

$$= \mathbb{E}_{r_\theta^{(\alpha,\lambda)}}\left[\psi_f\left(\frac{p}{r_\theta^{(\alpha,\lambda)}}\right) \cdot \nabla_\theta \log r_\theta^{(\alpha,\lambda)}\right] \tag{38}$$

Before deriving the gradient of the log probability of $r_\theta^{(\alpha,\lambda)}$, let us first derive $\nabla_\theta \tilde{r}_\theta^{(\alpha,\lambda)}$ as follows:

$$\nabla_\theta \tilde{r}_\theta^{(\alpha,\lambda)} = \nabla_\theta \left\{ h_\alpha^{-1}\big(\lambda\, h_\alpha(p) + (1-\lambda)\, h_\alpha(q_\theta)\big) \right\} \tag{39}$$

$$= \frac{1}{h_\alpha'\big(\tilde{r}_\theta^{(\alpha,\lambda)}\big)} (1-\lambda)\, \nabla_\theta h_\alpha(q_\theta) \tag{40}$$

$$= \frac{1}{h_\alpha'\big(\tilde{r}_\theta^{(\alpha,\lambda)}\big)} (1-\lambda)\, h_\alpha'(q_\theta)\, \nabla_\theta q_\theta \tag{41}$$

$$= \frac{1}{h_\alpha'\big(\tilde{r}_\theta^{(\alpha,\lambda)}\big)} (1-\lambda)\, h_\alpha'(q_\theta)\, q_\theta\, \nabla_\theta \log q_\theta \tag{42}$$

$$= (1-\lambda)\, \frac{h_\alpha'(q_\theta)\, q_\theta}{h_\alpha'\big(\tilde{r}_\theta^{(\alpha,\lambda)}\big)\, \tilde{r}_\theta^{(\alpha,\lambda)}}\, \tilde{r}_\theta^{(\alpha,\lambda)}\, \nabla_\theta \log q_\theta \tag{43}$$

$$= (1-\lambda) \left( \frac{q_\theta}{\tilde{r}_\theta^{(\alpha,\lambda)}} \right)^{\frac{1-\alpha}{2}} \tilde{r}_\theta^{(\alpha,\lambda)}\, \nabla_\theta \log q_\theta \tag{44}$$

$$= \frac{(1-\lambda)\, q_\theta^{\frac{1-\alpha}{2}}}{\lambda\, p^{\frac{1-\alpha}{2}} + (1-\lambda)\, q_\theta^{\frac{1-\alpha}{2}}}\, \tilde{r}_\theta^{(\alpha,\lambda)}\, \nabla_\theta \log q_\theta \tag{45}$$

$$= w \cdot \tilde{r}_\theta^{(\alpha,\lambda)} \cdot \nabla_\theta \log q_\theta\,. \tag{46}$$

Therefore,

$$\nabla_\theta \log r_\theta^{(\alpha,\lambda)} = \frac{\nabla_\theta \tilde{r}_\theta^{(\alpha,\lambda)}}{\tilde{r}_\theta^{(\alpha,\lambda)}} - \frac{1}{Z_r} \sum_k \nabla_\theta \tilde{r}_\theta^{(\alpha,\lambda)}(k) \tag{47}$$

$$= w \cdot \nabla_\theta \log q_\theta - \mathbb{E}_{r_\theta^{(\alpha,\lambda)}}\big[ w \cdot \nabla_\theta \log q_\theta \big] \tag{48}$$

Lastly, placing Eq. (48) into Eq. (38) and rearranging will yield the final result.

$$\nabla_\theta D_f(p \| r_\theta^{(\alpha,\lambda)}) = \mathbb{E}_{r_\theta^{(\alpha,\lambda)}} \left[ \psi_f\left( \frac{p}{r_\theta^{(\alpha,\lambda)}} \right) \cdot \nabla_\theta \log r_\theta^{(\alpha,\lambda)} \right] \tag{49}$$

$$= \mathbb{E}_{r_\theta^{(\alpha,\lambda)}} \left[ \psi_f\left( \frac{p}{r_\theta^{(\alpha,\lambda)}} \right) \cdot \left\{ w \cdot \nabla_\theta \log q_\theta - \mathbb{E}_{r_\theta^{(\alpha,\lambda)}}\big[ w \cdot \nabla_\theta \log q_\theta \big] \right\} \right] \tag{50}$$

$$= \mathbb{E}_{r_\theta^{(\alpha,\lambda)}} \left[ w \cdot \left\{ \psi_f\left( \frac{p}{r_\theta^{(\alpha,\lambda)}} \right) - \mathbb{E}_{r_\theta^{(\alpha,\lambda)}}\left[ \psi_f\left( \frac{p}{r_\theta^{(\alpha,\lambda)}} \right) \right] \right\} \cdot \nabla_\theta \log q_\theta \right] \tag{51}$$

$$\square$$

## B  EXPERIMENTAL DETAILS

### B.1  TOY EXPERIMENT

We employ a two-modal Gaussian mixture for the teacher $p = 0.7\mathcal{N}(-3, 2) + 0.3\mathcal{N}(3, 0.8)$ and a unimodal Gaussian for the student $q_\theta = \mathcal{N}(\mu, \sigma^2)$ with $\mu_0 = 0, \sigma_0^2 = 1$. We optimize $\theta = \{\mu, \sigma^2\}$ by minizing $D_{\mathrm{KL}}(p \| r_\theta^{(\alpha,\lambda)})$ with Adam optimizer (Kingma & Ba, 2014) with 5000 steps and 5e-2 learning rate.

## B.2 DATASETS

- **databricks-dolly-15k** (Conover et al., 2023): An open-source dataset of instruction–response pairs created by thousands of Databricks employees. It covers diverse behavioral categories defined in Ouyang et al. (2022), including brainstorming, classification, closed QA, generation, information extraction, open QA, and summarization.

- **Self-instruct** (Wang et al., 2023): A framework for improving instruction-following ability by iteratively using model outputs to generate new instructional data. The dataset contains 52K instructions and 82K input–output pairs for tuning, 252 expert-written tasks for practical evaluation, and 50K additional examples from public datasets for benchmarking.

- **Vicuna** (Chiang et al., 2023): A benchmark consisting of 80 challenging open-ended questions originally used to assess Vicuna. It provides a compact but difficult testbed for evaluating instruction-following performance.

- **Super-Natural Instructions** (Wang et al., 2022): A large-scale benchmark comprising 1,616 expert-written NLP tasks spanning 76 task categories. Its test set includes 9K examples drawn from 119 tasks, covering a wide spectrum of instruction types.

- **Unnatural Instructions** (Honovich et al., 2023): An AI-generated dataset containing 240K instructions created with minimal human intervention. The collection demonstrates that synthetic data can serve as an effective substitute for human-curated data. Its core subset includes 60K examples.

- **AlpacaEval** (Dubois et al., 2023): AlpacaEval is derived from the AlpacaFarm evaluation suite but includes simplified formatting. In particular, Dubois et al. (2023) combined the original instruction and input fields into one unified instruction, a change that influences about a quarter of the samples sourced from Self-Instruct (Wang et al., 2023). The dataset ultimately comprises 805 difficult instruction-following queries.

- **Evol-Instruct Evaluation** (Xu et al., 2024): This evaluation set includes 218 prompts produced through the Evol-Instruct generation pipeline. The questions span a wide range of topics and serve as a compact benchmark for instruction-following ability.

- **UltraFeedback** (Cui et al., 2024): UltraFeedback is a large and detailed preference dataset tailored for building high-quality reward and critic models. It contains roughly 64k prompts collected from UltraChat, ShareGPT, and Evol-Instruct. Each prompt was used to elicit four responses from different LLMs, and GPT-4 subsequently annotated these outputs based on dimensions such as following instructions, factual correctness, honesty, and helpfulness.

- **MetaMathQA** (Yu et al., 2024): MetaMathQA was created to strengthen mathematical reasoning in LLMs. The dataset is generated through a bootstrapping method in which each math question is re-expressed using multiple reasoning viewpoints, including forward reasoning, backward reasoning, and paraphrased formulations.

- **GSM8K** (Cobbe et al., 2021): GSM8K consists of 8.5K meticulously written grade-school math word problems. Designed to require multi-step inference, it is widely used as a standard benchmark for evaluating basic mathematical reasoning skills.

- **WizardCoder** (Luo et al., 2024): WizardCoder is an instruction-tuned code dataset built via the Evol-Instruct method. Starting with the 20K-sample Code Alpaca corpus, the creators iteratively evolved prompts by increasing complexity, adding constraints, inserting misleading code, and introducing time/space complexity requirements. The final dataset contains about 78K evolved examples, which were used to fine-tune StarCoder and substantially improve its coding performance.

- **HumanEval** (Chen, 2021): HumanEval provides 164 hand-crafted programming tasks with function signatures, natural-language descriptions, and unit tests. It is one of the primary benchmarks for assessing code generation and was explicitly designed to avoid overlap with existing training data.

- **MBPP** (Austin et al., 2021): MBPP includes roughly 1,000 Python programming exercises aimed at novice programmers. Each problem comes with a textual description, a reference code solution, and three automatic test cases. Portions of the dataset were manually validated to ensure consistency and correctness.

## B.3 IMPLEMENTATION SETTINGS

**Training.** For instruction-following distillation, we use databricks-dolly-15K (Conover et al., 2023) for the distillation loss and OpenWebText (Gokaslan & Cohen, 2019) for the pretraining loss. Teacher models include GPT-2 XL (1.5B) with SFT, and students are GPT-2 (0.1B), GPT-2 Medium (0.3B), and GPT-2 Large (0.8B). To test scalability, OpenLLaMA2-7B (Geng & Liu, 2023) is distilled into OpenLLaMA2-3B using LoRA.

For task-specific evaluation, we use Flores-200 (Costa-Jussà et al., 2022) for translation, Dialog-Sum (Chen et al., 2021) for summarization, and GSM8K (Cobbe et al., 2021) for mathematical reasoning. Teacher models are Gemma-7B-It (Team et al., 2024) and Qwen2-7B-Instruct (Team, 2024), while Gemma-2B-It and Qwen2-0.5B-Instruct serve as students. Teachers are fine-tuned on the full dataset, while students are trained with about 1,000 samples.

We use $\alpha_{AB}$-$\beta_{AB}$-divergence with $\alpha_{AB} = 0.2, \beta_{AB} = 0.7$ and adopt adaptive off-policy training (Ko et al., 2024) as default. We explore $\alpha$ over the range $\{-5, -3, -1, -0.5, 0, 0.5, 1.0\}$ and employ $\lambda = 0.1$ as default by following prior work (Ko et al., 2024). We utilize the AdamW optimizer and cosine learning rate scheduling by following the previous work (Ko et al., 2024; Wang et al., 2025). We search the learning rate over the range $\{0.0005, 0.0001, 0.00005\}$ except for the cooperation with contrastive-based distillation experiments, which use the default setting of DistiLLM-2 (Ko et al., 2025).

**Theoretical insights based $\alpha$ tuning guidelines.** Herein, we provide principled tuning guidelines grounded in the theoretical properties of $\alpha$. First, in Section 3.2, we show that the support of the assistant distribution varies with $\alpha$: $supp(r_\theta^{(\alpha,\lambda)}) = supp(p) \cup supp(q_\theta)$ when $\alpha < 1$, $supp(r_\theta^{(\alpha,\lambda)}) = supp(p) \cap supp(q_\theta)$ when $\alpha \geq 1$. In KD for LLMs, the teacher and student often exhibit a capacity gap and produce high-dimensional outputs with many near-zero probabilities. As a result, they do not share a sufficiently large common support in general. For this reason, we recommend using $\alpha < 1$ in most practical settings, as this choice improves training stability and enables more reliable knowledge transfer.

Furthermore, our gradient analysis and (toy) experiments demonstrate that $\alpha$ directly controls the trade-off between mode-covering and mode-seeking behavior of the optimized student. Under the $\alpha < 1$, increasing $\alpha$ encourages relatively stronger mode covering, thereby improving output diversity. Conversely, smaller $\alpha$ emphasizes mode seeking, which enhances fidelity to the teacher. Therefore, for enhancing the teacher–student alignment and performance, we suggest using small $\alpha$ values. However, since too small $\alpha$ can induce high curvature in the geometry of the interpolation path, which may reduce optimization efficiency, so such choices should be used with caution.

Based on these theoretical insights, we explore $\alpha$ over the range $\{-5, -3, -1, -0.5, 0, 0.5, 1.0\}$. Table 7 provides the detailed configuration for each task.

Table 7: Configuration of hyperparameters for AMiD

| Task | Dataset | Teacher | Student | Divergence | $\alpha$ |
|---|---|---|---|---|---|
| Instruction following | databricks-dolly-15k | GPT-2 XLarge | GPT-2 Base | $D_{KL}(p\|r)$ | -5.0 |
| | | | | $D_{RKL}(p\|r)$ | -3.0 |
| | | | | $D_{AB}(p\|r)$ | -5.0 |
| | | | | $D_{KL}(q\|r)$ | 0.5 |
| | | | GPT-2 Medium | $D_{AB}(p\|r)$ | -5.0 |
| | | | GPT-2 Large | $D_{AB}(p\|r)$ | -3.0 |
| | | OpenLLaMA2-7B | OpenLLaMA2-3B | $D_{AB}(p\|r)$ | -3.0 |
| | UltraChat200k | Qwen2.5-7B-Instruct | Qwen2.5-1.5B-Instruct | $D_{DistilLLM-2}$ | -5.0 |
| | | Qwen2.5-14B-Instruct | Qwen2.5-1.5B-Instruct | $D_{DistilLLM-2}$ | -5.0 |
| Translation | Flores-200 | Gemma-7B-It | Gemma-2B-It | $D_{AB}(p\|r)$ | -0.5 |
| | | Qwen2-7B-Instruct | Qwen2-0.5B-Instruct | $D_{AB}(p\|r)$ | 0.5 |
| Summarization | DialogSum | Gemma-7B-It | Gemma-2B-It | $D_{AB}(p\|r)$ | 0.5 |
| | | Qwen2-7B-Instruct | Qwen2-0.5B-Instruct | $D_{AB}(p\|r)$ | -3.0 |
| Mathematical reasoning | GSM8k | Gemma-7B-It | Gemma-2B-It | $D_{AB}(p\|r)$ | 0.5 |
| | | Qwen2-7B-Instruct | Qwen2-0.5B-Instruct | $D_{AB}(p\|r)$ | -3.0 |
| | MetaMathQA | Qwen2.5-Math-7B-Instruct | Qwen2.5-Math-1.5B-Instruct | $D_{DistilLLM-2}$ | -5.0 |
| Code generation | WizardCoder | Qwen2.5-Coder-7B-Instruct | Qwen2.5-Coder-1.5B-Instruct | $D_{DistilLLM-2}$ | -5.0 |

**Evaluation.** For evaluating generation quality, we adopt ROUGE-L (Lin, 2004) and Self-BLEU (Zhu et al., 2018). ROUGE-L measures the similarity between the generated output and the reference text by computing the Longest Common Subsequence (LCS). Specifically, recall and precision are defined as

$$R_{\text{LCS}} = \frac{LCS(x,y)}{L_x}, P_{\text{LCS}} = \frac{LCS(x,y)}{L_y}, \tag{52}$$

where $LCS(x,y)$ is the length of the longest common subsequence between the reference $x$ and the generated text $y$, and $L_x$, $L_y$ denote their respective lengths. The final ROUGE-L score is given by the harmonic mean:

$$ROUGE\text{-}L = \frac{2 \cdot R_{\text{LCS}} \cdot P_{\text{LCS}}}{R_{\text{LCS}} + P_{\text{LCS}}}. \tag{53}$$

A higher ROUGE-L score indicates that the generated text more closely matches the reference in terms of sequence overlap.

Self-BLEU evaluates the diversity of generated outputs by leveraging the BLEU metric (Papineni et al., 2002). BLEU computes the geometric mean of modified $n$-gram precisions with a brevity penalty (BP):

$$BP = \begin{cases} 1 & \text{if } c > r, \\ e^{(1-r/c)} & \text{if } c \leq r, \end{cases} \tag{54}$$

$$\text{BLEU}(c, R) = \text{BP} \cdot \exp\left(\sum_{n=1}^{N} w_n \log p_n(c, R)\right), \tag{55}$$

where $c$ is the candidate length, $r$ is the effective reference length, $p_n(c, R)$ denotes the modified $n$-gram precision, and $w_n$ are positive weights summing to one. Building on this definition, Self-BLEU is calculated by treating each generated sample $s_i$ as the hypothesis and the remaining set $S\backslash\{s_i\}$ as references:

$$\text{Self-BLEU}(S) = \frac{1}{M}\sum_{i=1}^{M} \text{BLEU}(s_i, S\backslash\{s_i\}). \tag{56}$$

A higher Self-BLEU score (close to 1) indicates that the outputs are highly similar to each other, reflecting low diversity and more deterministic behavior, while a lower score (close to 0) suggests greater diversity across generations.

## C  ADDITIONAL EXPERIMENTAL RESULTS AND DISCUSSIONS

### C.1  MORE COMPARISON WITH BASELINES

Table 15 presents the complete results on the GPT-2 family, where we extend the comparison to a broader set of baseline methods beyond those reported in the main paper. We observe that AMiD consistently outperforms all competing approaches across different student sizes (0.1B, 0.3B, 0.8B), further validating the robustness of our method. In particular, while methods such as SeqKD, ImitKD, MiniLLM, and AKL yield modest improvements over standard knowledge distillation (KD), they still fall short of strong assistant-based methods like GKD, TAID, and DistiLLM. Among these baselines, ABKD often emerges as the strongest competitor. Nevertheless, AMiD achieves clear performance gains over ABKD in nearly every evaluation setting.

### C.2  RESULTS ON OPENLLAMA2

Table 8 reports results on the OpenLLaMA2 family, where a 7B teacher is distilled into a 3B student. Consistent with our findings on the GPT-2 series, AMiD achieves the best overall performance across most evaluation benchmarks. In particular, AMiD surpasses prior assistant-based approaches such as TAID and DistiLLM (both SKL and SRKL variants), as well as the strong baseline ABKD.

Table 8: ROUGE-L scores (↑) on OpenLLaMA2-7B → OpenLLaMA2-3B. **Bold** and Underline mean the best and second-best performance of each column, except the teacher, respectively. All results are based on our own re-implementation. We conduct the evaluation with five random seeds.

| Model | Val. (↑) | Dolly Eval (↑) | Self Inst (↑) | Vicuna (↑) | Super NI (↑) | UnNI (↑) | Avg. (↑) |
|---|---|---|---|---|---|---|---|
| Teacher | – | 27.60 $_{\pm 0.34}$ | 18.17 $_{\pm 0.80}$ | 17.85 $_{\pm 0.48}$ | 31.05 $_{\pm 0.31}$ | 32.40 $_{\pm 0.28}$ | 25.41 |
| ***OpenLLaMA2-7B → OpenLLaMA2-3B*** | | | | | | | |
| TAID | 30.85 | 26.53 $_{\pm 0.23}$ | 17.73 $_{\pm 0.69}$ | 18.14 $_{\pm 0.39}$ | 31.93 $_{\pm 0.23}$ | 31.55 $_{\pm 0.12}$ | 25.18 |
| DistiLLM (SKL) | 33.07 | 28.63 $_{\pm 0.28}$ | 20.20 $_{\pm 0.66}$ | 19.15 $_{\pm 0.32}$ | 35.31 $_{\pm 0.19}$ | 34.74 $_{\pm 0.10}$ | 27.61 |
| DistiLLM (SRKL) | 33.18 | 28.83 $_{\pm 0.41}$ | 20.76 $_{\pm 0.37}$ | 19.37 $_{\pm 0.15}$ | 36.82 $_{\pm 0.14}$ | 35.76 $_{\pm 0.13}$ | 28.31 |
| ABKD | 33.91 | 29.43 $_{\pm 0.42}$ | 20.46 $_{\pm 0.28}$ | 20.42 $_{\pm 0.12}$ | **39.51** $_{\pm 0.25}$ | **38.07** $_{\pm 0.08}$ | 29.58 |
| **AMiD (Ours)** | **34.39** | **29.69** $_{\pm 0.47}$ | **20.99** $_{\pm 0.37}$ | **21.03** $_{\pm 0.40}$ | 39.06 $_{\pm 0.21}$ | 37.31 $_{\pm 0.11}$ | **29.62** |

## C.3 COMPATIBILITY WITH DIVERGENCES $D_{\text{KL}}(q_\theta \| r_\theta^{(\alpha,\lambda)})$

Table 9 provides the complementary results when employing the divergence $D_{\text{KL}}(q_\theta \| r_\theta^{(\alpha,\lambda)})$, contrasting the student distribution against the $\alpha$-mixture assistant. Similar to the findings in the main text (Table 3), AMiD consistently outperforms the no-assistant baseline across most evaluation benchmarks, confirming that the proposed method is broadly compatible with different divergence directions.

Table 9: ROUGE-L scores (↑) with $D_{\text{KL}}(q_\theta \| r_\theta^{(\alpha,\lambda)})$ and various $\alpha$. We utilize GPT-2 XL (1.5B) → GPT-2 (0.1B). We use a fixed $\lambda = 0.9$ for these experiments.

| Divergence $D$ | Assistant $r_\theta^{(\alpha,\lambda)}$ | Val. (↑) | Dolly Eval (↑) | Self Inst (↑) | Vicuna (↑) | Super NI (↑) | UnNI (↑) | Avg. (↑) |
|---|---|---|---|---|---|---|---|---|
| $D_{\text{KL}}(q_\theta \| r_\theta^{(\alpha,\lambda)})$ | $q_\theta$ | 28.71 | 26.22 $_{\pm 0.35}$ | 12.57 $_{\pm 0.16}$ | 16.97 $_{\pm 0.34}$ | 24.75 $_{\pm 0.20}$ | 26.59 $_{\pm 0.14}$ | 21.42 |
| | AMiD ($\alpha = -5.0$) | 27.54 | 24.23 $_{\pm 0.23}$ | 12.59 $_{\pm 0.32}$ | 15.80 $_{\pm 0.43}$ | 24.50 $_{\pm 0.14}$ | 26.38 $_{\pm 0.07}$ | 20.70 |
| | AMiD ($\alpha = -3.0$) | 27.96 | 25.13 $_{\pm 0.29}$ | 12.80 $_{\pm 0.48}$ | 16.32 $_{\pm 0.45}$ | 25.54 $_{\pm 0.30}$ | 26.86 $_{\pm 0.14}$ | 21.33 |
| | AMiD ($\alpha = -1.0$) | 28.21 | 25.74 $_{\pm 0.20}$ | 12.13 $_{\pm 0.23}$ | 16.34 $_{\pm 0.15}$ | 25.40 $_{\pm 0.10}$ | 26.91 $_{\pm 0.12}$ | 21.30 |
| | AMiD ($\alpha = -0.5$) | 28.89 | 26.01 $_{\pm 0.32}$ | 12.84 $_{\pm 0.59}$ | **17.04** $_{\pm 0.05}$ | 27.43 $_{\pm 0.14}$ | 27.59 $_{\pm 0.03}$ | 22.18 |
| | AMiD ($\alpha = 0.0$) | 28.84 | **26.70** $_{\pm 0.33}$ | 13.36 $_{\pm 0.36}$ | 15.95 $_{\pm 0.36}$ | 26.23 $_{\pm 0.17}$ | 27.70 $_{\pm 0.10}$ | 21.99 |
| | **AMiD ($\alpha = -0.5$)** | **29.02** | 26.48 $_{\pm 0.17}$ | **13.73** $_{\pm 0.44}$ | 16.78 $_{\pm 0.30}$ | 26.78 $_{\pm 0.34}$ | **28.65** $_{\pm 0.10}$ | **22.48** |
| | AMiD ($\alpha = 1.0$) | 28.40 | 26.01 $_{\pm 0.34}$ | 12.03 $_{\pm 0.33}$ | 16.96 $_{\pm 0.31}$ | 24.84 $_{\pm 0.24}$ | 27.01 $_{\pm 0.11}$ | 21.37 |

## C.4 OVERLAP-BASED ADAPTIVE $\alpha$ SCHEDULING

The theoretical insights based $\alpha$ tuning guidelines efficiently exclude low potential candidates. However, applying a single fixed global $\alpha$ value can still be sub-optimal in certain cases.

To address this concern, we introduce a curriculum-based adaptive $\alpha$ scheduling based on the degree of overlap between token-level teacher distribution $p(y_l|y_{<l}, x)$ and student distribution $q_\theta(y_l|y_{<l}, x)$. The intuition is that when the teacher and student distributions are highly overlapped, we encourage mode-covering to align further, whereas when the overlap is low, we enhance mode-seeking to find the mode first.

We define token-level overlap as $ovl_{i,l} := \sum_{y_l} \min(p(y_l|y_{<l}, x), q_\theta(y_l|y_{<l}, x))$. Obviously, the overlap value $ovl_{i,l}$ is bounded into $[0, 1]$. Also, $ovl_{i,l}$ approaches 0 when $p(y_l|y_{<l}, x)$ and $q_\theta(y_l|y_{<l}, x)$ significantly differ, and approaches 1 when they are well-aligned. Furthermore, $ovl_{i,l}$ can be expressed in terms of total variation distance $1 - TVD(p(y_l|y_{<l}, x), q_\theta(y_l|y_{<l}, x))$, which provides same interpretation.

Given the predefined $\alpha_{min}$ and $\alpha_{max}$, we set the token-level $\alpha_{i,l}$ as the linearly increasing value along the line passing through $(0, \alpha_{min})$ and $(1, \alpha_{max})$ i.e. $\alpha_{i,l} \leftarrow (\alpha_{max} - \alpha_{min}) * ovl_{i,l} + \alpha_{min}$. Under this idea, when the teacher and student distributions differ substantially, the $ovl_{i,l}$ becomes small, leading to a smaller assigned $\alpha$, which strengthens mode-seeking. Conversely, when the teacher and student distributions are similar, both $ovl_{i,l}$ and $\alpha$ have larger values, thereby reinforcing mode-covering. This mechanism systematically determines $\alpha$ by combining the degree of alignment between the teacher and student distribution, which continuously changes through training, with the theoretical characteristics of $\alpha$.

Table 10: ROUGE-L scores ($\uparrow$) on five task-agnostic instruction-following datasets with fixed $\alpha$ versus adaptive $\alpha$ scheduling. **Bold** means the best performance of each column. We use $D_{AB}$ and $\lambda = 0.1$ for AMiD.

| Assistant | Val. ($\uparrow$) | Dolly Eval ($\uparrow$) | Self Inst ($\uparrow$) | Vicuna ($\uparrow$) | Super NI ($\uparrow$) | UnNI ($\uparrow$) | Avg. ($\uparrow$) |
|---|---|---|---|---|---|---|---|
| AMiD (Fixed $\alpha$) | 29.24 | 26.44 | 13.74 | 16.76 | 29.71 | 30.35 | 23.40 |
| AMiD (Adaptive $\alpha$) | 29.31 | 26.50 | 14.02 | 16.59 | 29.87 | 30.60 | **23.52** |

## C.5 ROBUSTNESS TO OPTIMIZER

To investigate whether the effectiveness of AMiD depends on a particular optimization setup, we evaluate its performance under the Lion optimizer. Table 11 exhibits the robustness of AMiD w.r.t. the optimizer.

Table 11: ROUGE-L scores ($\uparrow$) on five task-agnostic instruction-following datasets under the Lion optimizer across different $\alpha$-mixture configurations. **Bold** means the best performance of each column. We use $D_{AB}$ and $\lambda = 0.1$ for AMiD.

| | Assistant | Val. ($\uparrow$) | Dolly Eval ($\uparrow$) | Self Inst ($\uparrow$) | Vicuna ($\uparrow$) | Super NI ($\uparrow$) | UnNI ($\uparrow$) | Avg. ($\uparrow$) |
|---|---|---|---|---|---|---|---|---|
| | No assistant | 28.38 | 26.02 $\pm 0.28$ | 12.19 $\pm 0.34$ | 17.24 $\pm 0.37$ | 26.29 $\pm 0.19$ | 28.88 $\pm 0.10$ | 22.12 |
| Lion | AMiD ($\alpha = -1$) | 28.68 | 25.29 $\pm 0.16$ | 12.21 $\pm 0.25$ | 17.81 $\pm 0.36$ | 24.82 $\pm 0.13$ | 27.87 $\pm 0.08$ | 21.60 |
| | AMiD ($\alpha = +1$) | 24.98 | 22.33 $\pm 0.27$ | 9.50 $\pm 0.28$ | 15.50 $\pm 0.52$ | 15.71 $\pm 0.34$ | 18.01 $\pm 0.08$ | 16.21 |
| | AMiD ($\alpha = \pm 1$) | 27.85 | 26.14 $\pm 0.21$ | 12.55 $\pm 0.12$ | 17.25 $\pm 0.48$ | 28.28 $\pm 0.26$ | 29.69 $\pm 0.06$ | **22.78** |

## C.6 ROBUSTNESS TO LEARNING RATE SCHEDULING

To further assess the robustness of AMiD to optimization hyperparameters, we evaluate its performance under the Noam learning rate schedule, originally designed for transformer architectures. Table 12 also support the robustness of AMiD.

Table 12: ROUGE-L scores ($\uparrow$) on five task-agnostic instruction-following datasets under the Noam learning rate schedule across different $\alpha$-mixture configurations. **Bold** means the best performance of each column. We use $D_{AB}$ and $\lambda = 0.1$ for AMiD.

| | Assistant | Val. ($\uparrow$) | Dolly Eval ($\uparrow$) | Self Inst ($\uparrow$) | Vicuna ($\uparrow$) | Super NI ($\uparrow$) | UnNI ($\uparrow$) | Avg. ($\uparrow$) |
|---|---|---|---|---|---|---|---|---|
| | No assistant | 28.42 | 25.83 $\pm 0.17$ | 13.35 $\pm 0.54$ | 16.43 $\pm 0.28$ | 28.30 $\pm 0.24$ | 29.90 $\pm 0.11$ | 22.76 |
| Noam | AMiD ($\alpha = -1$) | 28.91 | 26.02 $\pm 0.34$ | 14.07 $\pm 0.25$ | 17.09 $\pm 0.19$ | 27.78 $\pm 0.11$ | 29.49 $\pm 0.04$ | 22.93 |
| | AMiD ($\alpha = +1$) | 28.33 | 25.59 $\pm 0.25$ | 13.61 $\pm 0.49$ | 16.25 $\pm 0.34$ | 26.42 $\pm 0.20$ | 28.26 $\pm 0.19$ | 22.09 |
| | AMiD ($\alpha = \pm 1$) | 29.39 | 26.12 $\pm 0.35$ | 13.07 $\pm 0.52$ | 16.53 $\pm 0.46$ | 29.06 $\pm 0.14$ | 30.86 $\pm 0.09$ | **23.19** |

## C.7 MITIGATE THE CONFLICT VIA TEMPERATURE SCALING

As discussed in Section 3.2, combining an assistant distribution with a narrow support and a divergence that requires the expectation w.r.t. the assistant distribution can lead to training instability and poor knowledge transfer.

Since the primary cause of this issue is the narrow support of the assistant distribution, we conjecture that applying distribution softening technique could alleviate the instability even for such problematic combinations. To verify this conjecture, we employ temperature $T > 1$, which is a widely used flattening technique in various area. The table below shows the performance when using various temperature values under $D_{RKL}(p||r_\theta^{(\alpha, \lambda)})$ with $\alpha = 1$. The results exhibit that introducing the temperature leads to stable training and can even yield strong performance with an appropriately chosen temperature value. However, large temperature causes an over-flattening effect, inducing large shift in the assistant distribution and consequently degrading performance. Overall, we demonstrate that temperature scaling can empirically mitigate the instability associated with problematic combinations under the appropriate temperature value.

Table 13: ROUGE-L scores ($\uparrow$) on five task-agnostic instruction-following datasets under $D_{\text{RKL}}(p\|r_\theta^{(\alpha,\lambda)})$ with $\alpha = 1$ when applying temperature scaling to soften the assistant distribution.

| Assistant | Val. ($\uparrow$) | Dolly Eval ($\uparrow$) | Self Inst ($\uparrow$) | Vicuna ($\uparrow$) | Super NI ($\uparrow$) | UnNI ($\uparrow$) | Avg. ($\uparrow$) |
|---|---|---|---|---|---|---|---|
| AMiD ($\alpha = 1.0, T = 1.0$) | 0.16 | 4.27 | 2.81 | 9.12 | 1.64 | 1.84 | 3.94 |
| AMiD ($\alpha = 1.0, T = 1.5$) | 27.40 | 24.61 | 11.55 | 17.11 | 21.41 | 23.39 | 19.61 |
| AMiD ($\alpha = 1.0, T = 2.0$) | 28.64 | 26.83 | 12.74 | 17.62 | 23.56 | 27.01 | 21.55 |
| AMiD ($\alpha = 1.0, T = 5.0$) | 28.78 | 26.78 | 12.43 | 17.30 | 25.87 | 27.74 | 22.02 |
| AMiD ($\alpha = 1.0, T = 10.0$) | 27.33 | 24.73 | 12.11 | 16.95 | 23.65 | 26.80 | 20.85 |

Table 14: ROUGE-L scores ($\uparrow$) across different combinations of $\lambda$ and the $\alpha$-mixture assistant distribution.

| $\lambda$ | Assistant | Val. ($\uparrow$) | Dolly Eval ($\uparrow$) | Self Inst ($\uparrow$) | Vicuna ($\uparrow$) | Super NI ($\uparrow$) | UnNI ($\uparrow$) | Avg. ($\uparrow$) |
|---|---|---|---|---|---|---|---|---|
| | No Assistant | 28.61 | 25.49 | 12.52 | 17.36 | 26.07 | 27.36 | 21.76 |
| | AMiD ($\alpha = -5.0$) | 29.24 | 26.44 | 13.74 | 16.76 | 29.71 | 30.35 | **23.40** |
| | AMiD ($\alpha = -3.0$) | 29.07 | 26.38 | 13.58 | 16.11 | 29.27 | 30.14 | 23.10 |
| | AMiD ($\alpha = -1.0$) | 28.70 | 26.10 | 13.34 | 16.71 | 26.55 | 29.55 | 22.45 |
| 0.1 | AMiD ($\alpha = -0.5$) | 28.70 | 26.37 | 13.59 | 17.02 | 27.06 | 28.50 | 22.51 |
| | AMiD ($\alpha = 0.0$) | 28.86 | 25.77 | 13.57 | 16.14 | 27.26 | 28.52 | 22.25 |
| | AMiD ($\alpha = 0.5$) | 28.46 | 25.80 | 12.94 | 16.59 | 26.29 | 27.73 | 21.87 |
| | AMiD ($\alpha = 1.0$) | 24.93 | 22.36 | 9.72 | 16.29 | 15.09 | 16.15 | 15.92 |
| | AMiD ($\alpha = -5.0$) | 29.38 | 26.41 | 13.81 | 16.44 | 29.19 | 30.58 | **23.29** |
| | AMiD ($\alpha = -3.0$) | 29.38 | 26.45 | 13.71 | 16.43 | 28.23 | 30.44 | 23.05 |
| | AMiD ($\alpha = -1.0$) | 29.11 | 26.31 | 14.09 | 16.70 | 28.68 | 29.89 | 23.13 |
| 0.5 | AMiD ($\alpha = -0.5$) | 28.79 | 26.55 | 13.85 | 16.30 | 27.85 | 29.46 | 22.80 |
| | AMiD ($\alpha = 0.0$) | 28.74 | 25.68 | 13.01 | 16.51 | 26.32 | 27.83 | 21.87 |
| | AMiD ($\alpha = 0.5$) | 26.73 | 24.13 | 12.06 | 16.19 | 22.88 | 24.82 | 20.02 |
| | AMiD ($\alpha = 1.0$) | 22.88 | 21.31 | 10.41 | 14.87 | 19.35 | 21.34 | 17.46 |
| | AMiD ($\alpha = -5.0$) | 29.26 | 26.64 | 13.49 | 16.40 | 28.65 | 30.40 | 23.12 |
| | AMiD ($\alpha = -3.0$) | 29.14 | 26.02 | 13.62 | 17.03 | 28.99 | 30.59 | **23.25** |
| | AMiD ($\alpha = -1.0$) | 29.32 | 26.19 | 13.24 | 15.83 | 29.23 | 29.97 | 22.89 |
| 0.9 | AMiD ($\alpha = -0.5$) | 29.12 | 26.19 | 13.32 | 16.40 | 28.15 | 29.68 | 22.75 |
| | AMiD ($\alpha = 0.0$) | 28.28 | 25.23 | 13.01 | 15.51 | 27.52 | 28.66 | 21.99 |
| | AMiD ($\alpha = 0.5$) | 22.05 | 19.92 | 10.76 | 12.33 | 25.09 | 24.78 | 18.58 |
| | AMiD ($\alpha = 1.0$) | 21.52 | 19.17 | 9.16 | 13.60 | 15.53 | 17.73 | 15.04 |

# D    DISCUSSION OF OPTIMALITY

Theorem 3.4 guarantees the optimality of AMiD, yet experimentally demonstrated extremely poor performance for the reverse KL divergence $D_{\text{RKL}}(p\|r_\theta^{(\alpha,\lambda)})$ and $\alpha = 1$ in Table 3. We conjecture that it is caused by the conflict between RKL and the support intersection property, which leads to instability. RKL includes the expectation of the assistant distribution $\mathbb{E}_{r_\theta^{(\alpha,\lambda)}}[\cdot]$ by definition. However, when $\alpha = 1$, since $\text{supp}(r_\theta^{(\alpha,\lambda)})$ is $\text{supp}(p) \cap \text{supp}(q_\theta)$ (see Section 3.2), $\mathbb{E}_{r_\theta^{(\alpha,\lambda)}}[\cdot]$ is conducted on an unstable and narrow region, and this phenomenon intensifies further in the early stages of optimization. In addition, we experimentally find that the combination of $D_{\text{RKL}}(p\|r_\theta^{(\alpha,\lambda)})$ and $\alpha = 1$ produces highly unstable loss and gradient within a few early steps. In conclusion, while AMiD theoretically guarantees optimality, it might be necessary to employ appropriate divergence and alpha values, taking into account the imperfect optimization.

# E    THE USE OF LARGE LANGUAGE MODELS (LLMS)

We employed the LLM to polish the paper writing. Specifically, it was used to request grammatical corrections once the author had drafted the text.

Table 15: ROUGE-L scores (↑) on five task-agnostic instruction-following datasets. **Bold** and Underline mean the best and second-best performance of each column, except the teacher, respectively. All results are based on our own re-implementation. We conduct the evaluation with five random seeds.

| Model | Val. (↑) | Dolly Eval (↑) | Self Inst (↑) | Vicuna (↑) | Super NI (↑) | UnNI (↑) | Avg. (↑) |
|---|---|---|---|---|---|---|---|
| Teacher | – | $27.14_{\pm 0.15}$ | $14.55_{\pm 0.82}$ | $16.12_{\pm 0.31}$ | $27.21_{\pm 0.25}$ | $31.41_{\pm 0.06}$ | 23.29 |
| *GPT-2 XL (1.5B) → GPT-2 (0.1B)* | | | | | | | |
| SFT | 25.81 | $23.54_{\pm 0.42}$ | $9.62_{\pm 0.21}$ | $14.79_{\pm 0.56}$ | $18.42_{\pm 0.23}$ | $19.33_{\pm 0.13}$ | 17.14 |
| KD | 25.25 | $23.44_{\pm 0.33}$ | $10.12_{\pm 0.28}$ | $14.93_{\pm 0.29}$ | $16.88_{\pm 0.24}$ | $18.87_{\pm 0.16}$ | 16.85 |
| SeqKD | 26.07 | $24.20_{\pm 0.31}$ | $11.12_{\pm 0.09}$ | $15.82_{\pm 0.37}$ | $19.29_{\pm 0.09}$ | $22.74_{\pm 0.05}$ | 18.63 |
| ImitKD | 23.91 | $22.02_{\pm 0.29}$ | $10.34_{\pm 0.53}$ | $15.32_{\pm 0.26}$ | $17.34_{\pm 0.26}$ | $19.68_{\pm 0.15}$ | 16.94 |
| GKD | 27.06 | $24.58_{\pm 0.13}$ | $11.78_{\pm 0.44}$ | $14.60_{\pm 0.37}$ | $22.84_{\pm 0.12}$ | $25.04_{\pm 0.09}$ | 19.77 |
| MiniLLM | - | $24.47_{\pm 0.18}$ | $12.83_{\pm 0.50}$ | $16.94_{\pm 0.40}$ | $25.58_{\pm 0.33}$ | $26.38_{\pm 0.17}$ | 21.24 |
| AKL | 25.62 | $23.23_{\pm 0.35}$ | $11.18_{\pm 0.21}$ | $14.94_{\pm 0.23}$ | $19.36_{\pm 0.39}$ | $22.41_{\pm 0.08}$ | 18.22 |
| TAID | 28.37 | $25.74_{\pm 0.27}$ | $12.91_{\pm 0.31}$ | $17.09_{\pm 0.18}$ | $23.66_{\pm 0.31}$ | $26.82_{\pm 0.05}$ | 21.24 |
| DistiLLM (SKL) | 27.88 | $25.50_{\pm 0.28}$ | $12.35_{\pm 0.39}$ | $16.10_{\pm 0.22}$ | $23.87_{\pm 0.39}$ | $26.16_{\pm 0.06}$ | 20.80 |
| DistiLLM (SRKL) | 28.21 | $25.74_{\pm 0.20}$ | $12.13_{\pm 0.23}$ | $16.34_{\pm 0.15}$ | $25.40_{\pm 0.10}$ | $26.91_{\pm 0.12}$ | 21.30 |
| ABKD | 28.61 | $25.49_{\pm 0.24}$ | $12.52_{\pm 0.52}$ | **$17.36_{\pm 0.55}$** | $26.07_{\pm 0.14}$ | $27.36_{\pm 0.10}$ | 21.76 |
| **AMiD (Ours)** | **29.24** | **$26.44_{\pm 0.12}$** | **$13.74_{\pm 0.49}$** | $16.76_{\pm 0.24}$ | **$29.71_{\pm 0.08}$** | **$30.35_{\pm 0.09}$** | **23.40** |
| *GPT-2 XL (1.5B) → GPT-2 Medium (0.3B)* | | | | | | | |
| SFT | 27.96 | $25.70_{\pm 0.35}$ | $12.60_{\pm 0.37}$ | $16.51_{\pm 0.19}$ | $24.21_{\pm 0.13}$ | $27.51_{\pm 0.17}$ | 21.31 |
| KD | 26.03 | $24.27_{\pm 0.42}$ | $10.58_{\pm 0.10}$ | $15.59_{\pm 0.10}$ | $18.15_{\pm 0.13}$ | $20.49_{\pm 0.24}$ | 17.82 |
| SeqKD | 28.41 | $26.61_{\pm 0.34}$ | $13.01_{\pm 0.46}$ | $16.42_{\pm 0.63}$ | $23.44_{\pm 0.20}$ | $26.93_{\pm 0.08}$ | 21.28 |
| ImitKD | 25.93 | $24.46_{\pm 0.62}$ | $12.00_{\pm 0.41}$ | $15.56_{\pm 0.46}$ | $20.12_{\pm 0.34}$ | $25.11_{\pm 0.16}$ | 19.45 |
| GKD | 27.90 | $25.06_{\pm 0.55}$ | $12.36_{\pm 0.42}$ | $15.71_{\pm 0.58}$ | $23.83_{\pm 0.26}$ | $27.14_{\pm 0.09}$ | 20.82 |
| MiniLLM | - | $25.80_{\pm 0.57}$ | $14.87_{\pm 0.35}$ | $17.62_{\pm 0.33}$ | $26.78_{\pm 0.26}$ | $30.70_{\pm 0.11}$ | 23.15 |
| AKL | 27.81 | $25.57_{\pm 0.10}$ | $12.06_{\pm 0.56}$ | $15.98_{\pm 0.17}$ | $22.22_{\pm 0.20}$ | $26.17_{\pm 0.13}$ | 20.40 |
| TAID | 29.45 | $27.01_{\pm 0.27}$ | $14.53_{\pm 0.47}$ | $17.58_{\pm 0.20}$ | $25.14_{\pm 0.15}$ | $29.79_{\pm 0.14}$ | 22.81 |
| DistiLLM (SKL) | 29.65 | $26.87_{\pm 0.13}$ | $14.11_{\pm 0.29}$ | $16.85_{\pm 0.54}$ | $25.59_{\pm 0.22}$ | $28.84_{\pm 0.03}$ | 22.45 |
| DistiLLM (SRKL) | 29.72 | $26.50_{\pm 0.20}$ | $13.79_{\pm 0.71}$ | $17.14_{\pm 0.52}$ | $26.25_{\pm 0.11}$ | $29.31_{\pm 0.16}$ | 22.60 |
| ABKD | 29.64 | $26.93_{\pm 0.17}$ | $13.69_{\pm 0.32}$ | $17.45_{\pm 0.27}$ | $28.15_{\pm 0.18}$ | $30.94_{\pm 0.06}$ | 23.43 |
| **AMiD (Ours)** | **30.83** | **$27.34_{\pm 0.18}$** | **$15.26_{\pm 0.46}$** | **$17.69_{\pm 0.27}$** | **$29.04_{\pm 0.20}$** | **$33.15_{\pm 0.13}$** | **24.50** |
| *GPT-2 XL (1.5B) → GPT-2 Large (0.8B)* | | | | | | | |
| SFT | 28.48 | $26.17_{\pm 0.41}$ | $13.78_{\pm 0.21}$ | $16.64_{\pm 0.48}$ | $23.76_{\pm 0.30}$ | $26.64_{\pm 0.12}$ | 21.40 |
| KD | 28.52 | $26.27_{\pm 0.26}$ | $13.72_{\pm 0.44}$ | $16.43_{\pm 0.25}$ | $25.24_{\pm 0.18}$ | $28.94_{\pm 0.09}$ | 22.12 |
| SeqKD | 28.24 | $26.16_{\pm 0.41}$ | $13.93_{\pm 0.56}$ | $16.35_{\pm 0.20}$ | $25.03_{\pm 0.27}$ | $28.58_{\pm 0.06}$ | 22.01 |
| ImitKD | 26.96 | $23.37_{\pm 0.40}$ | $13.26_{\pm 0.60}$ | $16.00_{\pm 0.33}$ | $23.31_{\pm 0.16}$ | $27.59_{\pm 0.14}$ | 20.71 |
| GKD | 29.36 | $26.38_{\pm 0.24}$ | $14.44_{\pm 0.66}$ | $17.02_{\pm 0.46}$ | $26.64_{\pm 0.16}$ | $30.99_{\pm 0.13}$ | 23.09 |
| MiniLLM | - | $26.30_{\pm 0.35}$ | $16.50_{\pm 0.52}$ | $18.14_{\pm 0.49}$ | $29.45_{\pm 0.17}$ | $34.40_{\pm 0.17}$ | 24.96 |
| AKL | 27.69 | $25.45_{\pm 0.40}$ | $13.83_{\pm 0.82}$ | $15.85_{\pm 0.35}$ | $25.41_{\pm 0.25}$ | $28.91_{\pm 0.05}$ | 21.89 |
| TAID | 29.83 | $26.85_{\pm 0.32}$ | $15.07_{\pm 0.31}$ | $17.02_{\pm 0.48}$ | $26.71_{\pm 0.23}$ | $31.09_{\pm 0.17}$ | 23.35 |
| DistiLLM (SKL) | 29.69 | $26.12_{\pm 0.27}$ | $15.69_{\pm 0.75}$ | $16.91_{\pm 0.43}$ | $27.23_{\pm 0.18}$ | $30.73_{\pm 0.12}$ | 23.34 |
| DistiLLM (SRKL) | 30.59 | $27.09_{\pm 0.40}$ | $14.61_{\pm 0.66}$ | $16.39_{\pm 0.27}$ | $28.44_{\pm 0.45}$ | $31.04_{\pm 0.06}$ | 23.51 |
| ABKD | 30.49 | $27.67_{\pm 0.34}$ | $15.46_{\pm 0.81}$ | **$17.43_{\pm 0.25}$** | $30.74_{\pm 0.22}$ | $33.11_{\pm 0.15}$ | 24.88 |
| **AMiD (Ours)** | **31.10** | **$27.86_{\pm 0.29}$** | **$16.46_{\pm 0.41}$** | $16.62_{\pm 0.50}$ | **$32.64_{\pm 0.26}$** | **$35.64_{\pm 0.07}$** | **25.84** |

