# OpenReview forum: "AMiD: Knowledge Distillation for LLMs with $\alpha$-mixture Assistant Distribution"
_ICLR.cc/2026/Conference — ICLR 2026 Poster_

### Official Review · Reviewer_RkqC · 2025-10-30

**Soundness:** 4
**Presentation:** 2
**Contribution:** 3
**Rating:** 6
**Confidence:** 4

**Summary:**

The paper proposes an α-mixture assistant distribution and a corresponding knowledge distillation framework named AMiD for large language models (LLMs). It unifies existing methods (e.g., m-mixture and e-mixture) as special cases within a broader, theoretically grounded family parameterized by α. The framework is supported by theoretical analysis (optimality, gradient behavior, support properties) and extensive experiments showing consistent improvements over prior approaches across various tasks, model scales, divergences, and data generation strategies.

**Strengths:**

Introduces a principled and generalized family of assistant distributions via α-mixture, offering a unified view of prior fragmented methods.
Provides solid theoretical grounding, including optimality guarantees, support analysis, continuity, and gradient-based interpretation of mode-covering vs. mode-seeking behavior.
Comprehensive experiments validate the effectiveness of AMiD across instruction-following, task-specific distillation, different student sizes, divergences, and SGO strategies.
The new design variable α offers practical control over the quality-diversity trade-off, independent of the interpolation weight λ.

**Weaknesses:**

The writing is not very clear and significantly hampers readability.** Despite strong technical content, the exposition is often dense and poorly structured, especially in Section 3. Key concepts (α-mixture, f-mean, α-divergence) are introduced rapidly without sufficient intuition or gradual buildup, making it difficult for readers to follow.
Figures (e.g., Figure 1 and 2) lack detailed captions and fail to fully clarify the geometric impact of α on interpolation paths.
Limited discussion on why α ≠ ±1 performs better in practice beyond empirical results; the gap between theoretical optimality and practical instability (e.g., DRKL with α=1) is noted but not deeply analyzed.
Some notation is ambiguous or inconsistently used (e.g., r vs. r̃, θ vs. θ′), adding unnecessary cognitive load.

**Questions:**

1. How should one choose the optimal α in practice? Are there adaptive or task-aware strategies for tuning α during training?
2. The paper fixes the divergence (e.g., DAB) while varying α. Have the authors explored joint optimization or tuning of both α and divergence parameters (e.g., α_AB, β_AB)?
3. Theorem 3.4 claims optimality for any divergence and α, yet Table 3 shows catastrophic failure for DRKL with α=1. Does this indicate that the “perfect optimization” assumption is too strong, and how should practitioners navigate the theory-practice gap?

---

> ### Author Response · Authors · 2025-11-21
> **Response to Reviewer RkqC [1/6]**
>
> We appreciate the constructive reviews and valuable comments. We address the concerns below.
>
> > **W1. [Less readability]** *The writing is not very clear and significantly hampers readability.** Despite strong technical content, the exposition is often dense and poorly structured, especially in Section 3. Key concepts (α-mixture, f-mean, α-divergence) are introduced rapidly without sufficient intuition or gradual buildup, making it difficult for readers to follow.*
>
> We agree that readability is crucial, especially when introducing a new generalized framework. The Section 3 in the previous manuscript consists of the motivation of our approach (Section 3.1), the definition of the $\alpha$-mixture assistant distribution (Section 3.2), and the formulation of $\alpha$-mixture distillation (Section 3.3). In Sections 3.2 and 3.3, where the generalized framework is newly introduced, we present each framework in the order of (1) its definition, (2) its fundamental properties, and (3) its implications in KD for LLMs. In particular, the properties derived from the gradient analysis are enumerated to enhance readability. We suspect that the factors currently reducing readability are: (1) the separation of content by the page, and (2) unclear boundaries between different properties. Therefore, in the revised manuscript, we reduce unintended separation and add connective explanation to make the transitions between properties clearer.
>
> We would like to emphasize that the manuscript already provides explanations supporting the intuition and progressive development of our framework. In Section 3.1, we clarify that existing assistant distributions correspond to the $m$-mixture and $e$-mixture, and Section 2.2 shows these mixtures as weighted arithmetic and weighted geometric means, respectively. Furthermore, Section 2.3 introduces the generalized $f$-mean, which provides a natural buildup for our framework. Subsequently, Section 3.2 introduces the $\alpha$-mixture assistant distribution by applying the generalized $f$-mean to the teacher and student distributions, and Figure 2a in the revised manuscript (Figure 1a in the previous manuscript) depicts the generalization ability of our framework. Nevertheless, we have revised the manuscript to improve clarity by (1) adding the illustration of generalized $f$-mean (Figure 1 in the revised manuscript), (2) providing an overview paragraph at the beginning of Section 3, (3) stating the motivation for introducing each components, and (4) adding an explanation of $\alpha$-divergence in Appendix A.1.
>
> ---
> > **W2. [Lack of detailed captions]** *Figures (e.g., Figure 1 and 2) lack detailed captions and fail to fully clarify the geometric impact of α on interpolation paths.*
>
> We add the following clarifications in the captions of Figures 2 and 3 in the revised manuscript (Figures 1 and 2 in the previous manuscript).
> * (Figure 2b–f) We explicitly indicate that the support of the $\alpha$-mixture assistant distribution varies depending on the value of $\alpha$.
> * (Figure 3a) We clarify that the curvature of the interpolation path increases as $\alpha$ moves farther away from -1, which exhibits the straight line.
> * (Figure 3b) We note that the parameters updated via AMiD influence the subsequent student distribution.
> * (Figure 3c) We specify that increasing or decreasing $\alpha$ respectively enhances mode-covering or mode-seeking behavior under $\alpha \leq 1$.

---

> ### Author Response · Authors · 2025-11-21
> **Response to Reviewer RkqC [2/6]**
>
> >**W3(a). [Limited discussion on better performance of $\alpha \neq \pm 1$]** *Limited discussion on why α ≠ ±1 performs better in practice beyond empirical results;*
>
> First, as noted in the manuscript, when $\alpha = 1$, the support of $\alpha$-mixture assistant distribution becomes the intersection of the teacher and student distributions. In KD for LLMs, the teacher and student typically exhibit a capacity gap and generate high-dimensional distributions in which many probabilities are unavoidably near zero. Consequently, the support of $\alpha$-mixture assistant distribution is defined within a narrow region, ultimately leading to inefficient knowledge transfer.
>
> In addition, adjusting $\alpha$ allows us to obtain diverse optimization dynamics. As discussed in Theorem 3.2, the $\alpha$-mixture assistant distribution is the interpolation point w.r.t. the $\alpha$-divergence, which induces different optimization paths (Figure 3b in the revised manuscript). By tuning $\alpha$, we can obtain several paths, such as a straight line ($\alpha=1$) and curved paths ($\alpha \neq \pm 1$), leading a various of teacher-student alignment dynamics. These diverse trajectories can circumvent local optima and training instability, enabling the experimental discovery of better optimized solutions.
>
> ---
> >**W3(b). [Discussion of conflict]** *the gap between theoretical optimality and practical instability (e.g., DRKL with α=1) is noted but not deeply analyzed.*
>
> *(The response for W3(b) is provided in global response 2. For your convenience, the same content is attached below. Also, we introduce the mitigation strategy in the global response 2 and the response for Q3.)*
>
> We have discussed this issue in Appendix D of the previous manuscript. In particular, for certain divergences, such as $D_{RKL}$, involve an expectation with respect to $r$ by the definition. When $\alpha = 1$, the support of $\alpha$-mixture assistant distribution equals to the intersection of the teacher and student distribution’s supports, which can be narrow in high-dimensional LLM outputs. Consequently, this small support might induce unstable training and insufficient knowledge transfer. Therefore, restricting $\alpha < 1$ is simple and guarantees the theoretical stability in most cases.
>
> ---
> >**W4. [Ambiguous and inconsistent notion]** *Some notation is ambiguous or inconsistently used (e.g., r vs. r̃, θ vs. θ′), adding unnecessary cognitive load.*
>
> We clarify that $\tilde{r}\_{\theta}^{(\alpha,\lambda)}$ and $r\_{\theta}^{(\alpha,\lambda)}$ denote the unnormalized and normalized $\alpha$-mixture assistant distributions, respectively, as stated in Definition 1. Specifically, $\tilde{r}\_\theta^{(\alpha,\lambda)}$ is the generalized $f_\alpha$-mean of the teacher distribution $p$ and the student distribution $q_\theta$. Because the sum of $\tilde{r}\_\theta^{(\alpha,\lambda)}$ may not equal to 1, it is not necessarily a valid probability distribution. For this reason, we additionally define the normalized form $r\_\theta^{(\alpha,\lambda)}$ as the $\alpha$-mixture assistant distribution used in our manuscript.
>
> $\theta$ and $\theta’$, which are appeared in Figure 3b of the revised manuscript (Figure 2b of the previous manuscript), correspond to the parameters before and after applying the AMiD optimization step, respectively. We have added this clarification to the caption of Figure 3b in the revised manuscript.

---

> ### Author Response · Authors · 2025-11-21
> **Response to Reviewer RkqC [3/6]**
>
> >**Q1. [Strategy for $\alpha$ selection]** *How should one choose the optimal α in practice? Are there adaptive or task-aware strategies for tuning α during training?*
>
> *(The response for Q1 is provided in global responses 1 and 2. For your convenience, the same content is attached below.)*
>
> Our proposed $\alpha$-mixture assistant distribution employs a new distribution design variable $\alpha$, which unifies and generalizes existing assistant distributions into a coherent distribution family. While this additional variable provides substantial flexibility and opens up unexplored assistant distribution candidates, identifying appropriate $\alpha$ values that achieve consistently stable and strong performance is important.
>
> We address this concern by providing tuning guidelines grounded in the theoretical properties of $\alpha$ analyzed in the manuscript. First, in Section 3.2, we show that the support of the $\alpha$-mixture assistant distribution varies with $\alpha$: $supp(r_\theta^{(\alpha,\lambda)}) = supp(p) \cup supp(q_\theta)$ when $\alpha < 1$, $supp(r_\theta^{(\alpha,\lambda)}) = supp(p) \cap supp(q_\theta)$ when $\alpha \geq 1$. In KD for LLMs, the teacher and student models often exhibit a capacity gap and produce high-dimensional outputs with many near-zero probabilities. As a result, they do not share a sufficiently large common support in general. For this reason, we recommend using $\alpha < 1$ in most practical settings, as this choice improves training stability and enables more reliable knowledge transfer.
>
> Furthermore, our gradient analysis and experimental results demonstrate that $\alpha$ enables to control the trade-off between mode-covering and mode-seeking behavior of the optimized student distribution. Under the $\alpha \leq 1$, increasing $\alpha$ relatively encourages mode-covering, thereby improving the diversity of outputs. Conversely, smaller $\alpha$ emphasizes mode-seeking, which enhances fidelity to the teacher. Therefore, for enhancing the teacher-student alignment and performance, we suggest using small $\alpha$ values. However, since too small $\alpha$ can induce high curvature in the geometry of interpolation path, which may reduce optimization efficiency, so such choices should be used with caution.
>
> Based on these theoretical insights, we basically consider $\alpha$ over [−5, −3, −1, −0.5, 0, 0.5, 1.0]. We observe consistently stable and strong performance across most of these candidates.
>
> ---
> The theoretical insights based $\alpha$ tuning guidelines efficiently exclude low potential candidates. However, applying a single fixed global $\alpha$ value can still be sub-optimal in certain cases.
>
> To address this concern, we introduce a curriculum-based adaptive $\alpha$ scheduling based on the degree of overlap between token-level teacher distribution $p(y_l | y_{<l}, x)$ and student distribution $q_\theta(y_l | y_{<l}, x)$. The intuition is that when the teacher and student distributions are highly overlapped, we encourage mode-covering to align further, whereas when the overlap is low, we enhance mode-seeking to find the mode first.
>
> We define token-level overlap as $ovl_{i,l} := \sum_{y_l} \min(p(y_l | y_{<l}, x), q_\theta(y_l | y_{<l}, x))$. Obviously, the overlap value $ovl_{i,l}$ is bounded into $[0, 1]$. Also, $ovl_{i,l}$ approaches $0$ when $p(y_l | y_{<l}, x)$ and $q_\theta(y_l | y_{<l}, x)$ significantly differ, and approaches $1$ when they are well-aligned. Furthermore, $ovl_{i,l}$ can be expressed in terms of total variation distance $1-TVD(p(y_l | y_{<l}, x), q_\theta(y_l | y_{<l}, x))$, which provides same interpretation.
>
> Given the predefined $\alpha_{min}$ and $\alpha_{max}$, we set the token-level $\alpha_{i,l}$ as the linearly increasing value along the line passing through $(0, \alpha_{min})$ and $(1, \alpha_{max})$ i.e. $\alpha_{i,l} \leftarrow (\alpha_{max} - \alpha_{min}) * ovl_{i,l} + \alpha_{min}$. Under this idea, when the teacher and student distributions differ substantially, the $ovl_{i,l}$ becomes small, leading to a smaller assigned $\alpha$, which strengthens mode-seeking. Conversely, when the teacher and student distributions are similar, both $ovl_{i,l}$ and $\alpha$ have larger values, thereby reinforcing mode-covering. This mechanism systematically determines $\alpha$ by combining the degree of alignment between the teacher and student distribution, which continuously changes through training, with the theoretical characteristics of $\alpha$. *(Continued below)*

---

> ### Author Response · Authors · 2025-11-21
> **Response to Reviewer RkqC [4/6]**
>
> *(Continued)* Below pseudocode presents the batch-wise training scheme of AMiD with proposed overlap-based adaptive $\alpha$ scheduling.
> ```
> Given: Dataset $\mathcal{D}$, Divergence $D$, $\alpha_min$, $\alpha_max$, $\lambda$
> 1. For each iteration
> 2.     Sample mini-batch $B$ from $\mathcal{D}$
> 3.     Obtain batch-level distribution of teacher $p$ and student $q$    // [B, L, V] shape
> 4.     Calculate batch-level overlap value $ovl \leftarrow \min(p, q).sum(-1)$    // [B, L] shape
> 5.     Calculate batch-level $\alpha \leftarrow (\alpha_{max} - \alpha_{min}) * ovl + \alpha_{min}$    // [B, L] shape
> 6.     Reshape $\alpha$    // [B, L, V] shape
> 7.     Calculate $r_\theta^{(\alpha, \lambda)}$    // [B, L, V] shape
> 8.   Update $\theta$ by minimizing $D(\cdot, r_\theta^{(\alpha, \lambda)})$    // either $p$ or $q_\theta$ is possible.
> ```
>
> The table below presents the performance of fixed $\alpha$ and overlap-based adaptive $\alpha$ scheduling under the $D_{AB}$ with $\alpha_{AB}=0.2, \beta_{AB}=0.7$ and $\lambda = 0.1$. For fixed approach, we set $\alpha = -5.0$ which is best performer among the fixed methods. For adaptive approach, we set $\alpha_{min} = -5.0, \alpha_{max} = -1.0$. As shown in the table below, the overlap-based adaptive $\alpha$ scheduling achieves higher performance than the best fixed global $\alpha$ strategy, demonstrating its effectiveness.
>
> | | Val. (↑) | Dolly Eval (↑) | Self Inst (↑) | Vicuna (↑) | Super NI (↑) | UnNI (↑) | Avg. (↑) |
> |-|-|-|-|-|-|-|-|
> | AMiD (Fixed $\alpha$) | 29.24 | 26.44 | 13.74 | 16.76 | 29.71 | 30.35 | 23.40 |
> | AMiD (Overlap-based adaptive $\alpha$) | 29.31 | 26.50 | 14.02 | 16.59 | 29.87 | 30.60 | **23.52** |
>
> We also compare the wall-clock time for each training step. As shown in the table below, the proposed overlap-based adaptive $\alpha$ scheduling shows a negligible time complexity increase compared to no-assistant baseline (ABKD) and fixed $\alpha$ approach due to the simple structure while theory-based approach.
>
> | | Avg. of wall-clock time of single training step (sec) |
> |-|-|
> | ABKD | 1.17 |
> | AMiD (fixed $\alpha$) | 1.26 |
> | AMiD (overlap-based adaptive $\alpha$) | 1.29 |
>
> We believe that more sophisticated adaptive $\alpha$ scheduling, such as bi-level optimization $\min_{(\alpha, \lambda)} D(p, q_{\theta’})$ where $\theta’ = argmin_{\theta} D(p, r_{\theta}^{(\alpha, \lambda)})$, could further enhance performance. Investigating such adaptive $\alpha$ and/or $\lambda$ scheduling could be interesting research direction.

---

> ### Author Response · Authors · 2025-11-21
> **Response to Reviewer RkqC [5/6]**
>
> >**Q2. [Ablation study for $\alpha_{AB}$ and $\beta_{AB}$]** *The paper fixes the divergence (e.g., DAB) while varying α. Have the authors explored joint optimization or tuning of both α and divergence parameters (e.g., α_AB, β_AB)?*
>
> We follow the default configuration of $\alpha_{AB}=0.2$ and $\beta_{AB}=0.7$ used in ABKD, which is the first work to apply $\alpha_{AB}$-$\beta_{AB}$-divergence to KD for LLMs. To verify that AMiD maintains robust and strong performance across a broader range of divergence settings, we conducted additional experiments using alternative combinations: $(\alpha_{AB},\beta_{AB})=(0.0,0.7)$, $(0.2,0.5)$, and $(0.5,0.5)$ which is Hellinger distance.
>
> As shown in the table below, AMiD with $\alpha \neq \pm 1$ consistently achieves the highest performance across all these divergence settings. Notably, for $(\alpha_{AB},\beta_{AB})=(0.0,0.7)$ and $(0.2,0.5)$, AMiD exhibits even higher performance than the results reported in the previous manuscript. These results repeatedly demonstrate that introducing and extending the new distribution design variable $\alpha$ provides meaningful performance gains, Also, AMiD shows the robustness under divergences designed for different purposes, such as controlling hardness concentration and confidence concentration.
>
> | Assistant    | Val. (↑) | Dolly Eval (↑) | Self Inst (↑) | Vicuna (↑) | Super NI (↑) | UnNI (↑) | Average (↑) |
> |----------------------|-------------|----------------|---------------|------------|--------------|----------|-------------|
> | **$\alpha_{AB} = 0.0, \beta_{AB} = 0.7$** |             |                |               |            |              |          |             |
> | No assistant         | 29.30       | 26.25         | 16.16        | 16.73     | 28.08       | 29.93   | 22.83       |
> | AMiD ($\alpha = -1$)  | 29.35       | 26.72         | 13.34        | 16.93     | 28.54       | 29.83   | 23.07       |
> | AMiD ($\alpha = 1$) | 28.80       | 25.93         | 13.45        | 17.59     | 27.27       | 29.47   | 22.74       |
> | AMiD ($\alpha \neq \pm 1$)  | 29.42       | 26.32         | 13.61        | 17.11     | 29.54       | 31.01   | **23.52**   |
> | **$\alpha_{AB} = 0.2, \beta_{AB} = 0.5$** |             |                |               |            |              |          |             |
> | No assistant         | 28.94       | 26.38         | 13.73        | 16.80     | 28.01       | 29.30   | 22.56       |
> | AMiD ($\alpha = -1$) | 28.94       | 26.92         | 13.06        | 16.35     | 29.08       | 30.98   | 23.15       |
> | AMiD ($\alpha = 1$)  | 28.42       | 25.56         | 13.82        | 16.55     | 27.21       | 29.86   | 22.42       |
> | AMiD ($\alpha \neq \pm 1$)  | 28.48       | 26.27         | 13.13        | 15.85     | 30.42       | 31.04   | **23.50**   |
> | **$\alpha_{AB} = 0.5, \beta_{AB} = 0.5$** |             |                |               |            |              |          |             |
> | No assistant         | 27.87       | 25.41         | 12.68        | 16.54     | 24.03       | 26.29   | 21.06       |
> | AMiD ($\alpha = -1$)  | 28.46       | 25.88         | 13.01        | 16.23     | 26.18       | 27.95   | 21.28       |
> | AMiD ($\alpha = 1$)    | 26.50       | 24.11         | 11.36        | 16.48     | 21.95       | 24.33   | 19.65       |
> | AMiD ($\alpha \neq \pm 1$)   | 28.72       | 26.33         | 13.27        | 16.01     | 27.78       | 29.48   | **22.57**   |

---

> ### Author Response · Authors · 2025-11-21
> **Response to Reviewer RkqC [6/6]**
>
> >**Q3. [Mitigation for catastrophic failure]** *Theorem 3.4 claims optimality for any divergence and α, yet Table 3 shows catastrophic failure for DRKL with α=1. Does this indicate that the “perfect optimization” assumption is too strong, and how should practitioners navigate the theory-practice gap?*
>
> The expression of perfect optimization used in Theorem 3.4 refers to the existence of the optimal parameter such that the $\alpha$-mixture assistant distribution becomes equal to the teacher (or) student distribution. This realizability assumption is widely used in the theoretical analyses for optimality and is not strong for LLMs, which have flexible function space due to huge parameter size. Nevertheless, we modify the statement of Theorem 3.4 in the revised manuscript to remove possible confusion.
>
> *(The response for Q3 is provided in global response 2. For your convenience, the same content is attached below.)*
>
> We have discussed this issue in Appendix D of the previous manuscript. In particular, for certain divergences, such as $D_{RKL}$, involve an expectation with respect to $r$ by the definition. When $\alpha = 1$, the support of $\alpha$-mixture assistant distribution equals to the intersection of the teacher and student distribution’s supports, which can be narrow in high-dimensional LLM outputs. Consequently, this small support might induce unstable training and insufficient knowledge transfer. Therefore, restricting $\alpha < 1$ is simple and guarantees the theoretical stability in most cases.
>
> Since the primary cause of this issue is the narrow support of the $\alpha$-mixture assistant distribution, we conjecture that applying distribution softening technique could alleviate the instability even for such problematic combinations. To verify this conjecture, we employ temperature $T > 1$, which is a widely used flattening technique in various area. The table below shows the performance when using various temperature values under $D_{RKL}(p || r_\theta^{(\alpha,\lambda)})$ with $\alpha = 1$. The results exhibit that introducing the temperature leads to stable training and can even yield high performance with an appropriately chosen value. However, large temperature causes an over-flattening effect, inducing large shift in the $\alpha$-mixture assistant distribution and consequently degrading performance. Overall, we demonstrate that employing temperature scaling can empirically mitigate the instability associated with problematic combinations under the appropriate temperature value.
>
> | | Val. (↑) | Dolly Eval (↑) | Self Inst (↑) | Vicuna (↑) | Super NI (↑) | UnNI (↑) | Average (↑) |
> |-|-|-|-|-|-|-|-|
> | AMiD ($\alpha = 1.0, T = 1.0$) | 0.16 | 4.27 | 2.81 | 9.12 | 1.64 | 1.84 | 3.94 |
> | AMiD ($\alpha = 1.0, T = 1.5$) | 27.40 | 24.61 | 11.55 | 17.11 | 21.41 | 23.39 | 19.61 |
> | AMiD ($\alpha = 1.0, T = 2.0$) | 28.64 | 26.83 | 12.74 | 17.62 | 23.56 | 27.01 | 21.55 |
> | AMiD ($\alpha = 1.0, T = 5.0$) | 28.78 | 26.78 | 12.43 | 17.30 | 25.87 | 27.74 | 22.02 |
> | AMiD ($\alpha = 1.0, T = 10.0$) | 27.33 | 24.73 | 12.11 | 16.95 | 23.65 | 26.80 | 20.85 |

---

### Official Review · Reviewer_yQkB · 2025-10-31

**Soundness:** 3
**Presentation:** 3
**Contribution:** 3
**Rating:** 6
**Confidence:** 3

**Summary:**

This paper presents  a unified framework for knowledge distillation with an assistant distribution, where the teacher and student distributions are mixed to bridge the capacity gap between them. Previous distillation mixing strategies (e.g., using arithmetic and geometric mean) become special cases under this framework.

**Strengths:**

1. The authors presents an interesting point of view on existing distillation methods.
2. The experiments are generally comprehensive, showing consistent improvements.

**Weaknesses:**

1. The motivation of adjusting alpha is still a bit unclear to me. The authors mention alpha adjusts the mode-seeking and mode-covering properties. However, this can also be addressed in the divergence function, as mentioned in 2.1. I am not too sure about the intuition behind doing this again in the mixing stage.

2. The paper mainly focuses on instruction following, evaluated by rouge. This is slightly concerning, because the approach may be hacking the ROUGE score rather than making the model better at instruction following. The paper showed very limited results on reasoning (GSM8K, omitting ABKD for some reason, and no standard deviations).

3. The paper is not very clear on its hyper-parameters. For example, the author mentions that their approach work with any divergence functions, but it's not clear which one is used for Table 1. Also, the values of alpha/lambda is not shown on the table. The author mentions D_AB being used for Table 2, but how about Table 1. Even in Table 2, it says alpha is not -1/+1, but it doesn't state what alpha is. My main worry is that the authors may have spent lots of efforts on tuning these individually, which gave the approach an unfair advantage.

**Questions:**

See weaknesses about hyper-parameters.

---

> ### Author Response · Authors · 2025-11-21
> **Response to Reviewer yQkB [1/3]**
>
> We appreciate the constructive reviews and valuable comments. We address the concerns below.
>
> > **W1. [Unclear motivation for adjusting $\alpha$]** *The motivation of adjusting alpha is still a bit unclear to me. The authors mention alpha adjusts the mode-seeking and mode-covering properties. However, this can also be addressed in the divergence function, as mentioned in 2.1. I am not too sure about the intuition behind doing this again in the mixing stage.*
>
> The proposed $\alpha$-mixture assistant distribution introduces a unified and generalized assistant distribution family that integrates the previous assistant distributions. This framework not only enables systematic comparison of their behaviors but also provides unexplored assistant distributions through the additional design variable $\alpha$.
>
> As the reviewers noted, adjusting the mode-covering and mode-seeking has traditionally relied on the choice of divergence. However, selecting a specific divergence, such as Forward KL or Reverse KL, inevitably induces a trade-off between quality and diversity. Through our gradient analysis under the $f$-divergences, we show that the $\alpha$-mixture assistant distribution enables relatively adjustment of mode-covering and mode-seeking via $\alpha$. We highlight that this property (1) holds regardless of the specific divergence choice within the $f$-divergence family and (2) enables the continuous adjustment for the trade-off. Overall, $\alpha$ provides a new orthogonal axis of quality-diversity control, which is distinct from divergence selection. Additionally, it is noted that $\alpha$ determines the optimization curriculum, while the divergence evaluates that curriculum, highlighting their distinct roles. Our toy experiments (Figure 3c in the revised manuscript) and real data experiments (Figures 5a and 5b in the revised manuscript) empirically demonstrate that even under the same divergence, adjusting $\alpha$ provides additional control over the quality and diversity.
>
> Furthermore, by tuning $\alpha$, we obtain diverse optimization dynamics. As discussed in Theorem 3.2, the $\alpha$-mixture assistant distribution corresponds to an interpolation point under $\alpha$-divergence geometry, which induces various optimization trajectories (Figure 3b in the revised manuscript). Adjusting $\alpha$ allows to change optimization behavior based on a straight path ($\alpha = 1$) or a curved path ($\alpha \neq \pm 1$), enabling a diverse of teacher–student alignment. This flexibility may help the model avoid local optima and reduce training instability, thereby improving the optimized solution. Our experimental results consistently show that $\alpha$ has a significant impact on performance, which reflects meaningful changes in the underlying optimization dynamics.

---

> ### Author Response · Authors · 2025-11-21
> **Response to Reviewer yQkB [2/3]**
>
> >**W2. [Effectiveness beyond the instruction following and ROUGE-L metric]** *The paper mainly focuses on instruction following, evaluated by rouge. This is slightly concerning, because the approach may be hacking the ROUGE score rather than making the model better at instruction following. The paper showed very limited results on reasoning (GSM8K, omitting ABKD for some reason, and no standard deviations).*
>
> We basically utilize ROUGE-L, a widely adopted evaluation metric in KD for LLMs, as our primary metric. To address the reviewer’s concern regarding evaluation diversity, we additionally conducted experiments using MT-bench score, another metric to assess instruction-following ability. For this experiment, we employ the code implementation of TAID and utilize Phi-3-mini (3.8B) as the teacher model and TinyLlama (1.1B) as the student model. For all baselines and AMiD, we use for 20 epochs and performed a learning-rate search over [0.0001, 0.00005, 0.00001, 0.000005]. As shown in the table below, the proposed AMiD achieves the highest performance on both ROUGE-L and MT-bench, confirming its robustness across evaluation metrics.
>
> | Model                   | Val ROUGE-L | MT-Bench Score (↑) |
> |-------------------------|-------------|----------------------|
> | DistiLLM (SKL)          | 24.74       | 4.29                 |
> | DistiLLM (SRKL)         | 23.56       | 4.23                 |
> | TAID w/o adaptive update | 32.04       | 4.24                 |
> | TAID                    | 32.48       | 4.39                 |
> | AMiD ($\alpha = 0.5, D_{AB}$)  | **34.24**   | **4.45**             |
>
> We further conducted additional experiments on mathematical reasoning, using the DistiLLM-2 code implementation. Following the configuration of DistiLLM-2, we employ the MetaMathQA dataset for the distillation and evaluate on GSM8K. We use the teacher and student models as Qwen2.5-Math-7B-Instruct and Qwen2.5-Math-1.5B-Instruct, respectively. We utilize the original DistiLLM-2 settings without any extra hyperparameter tuning, and set $\alpha = -5.0$, which showed consistently strong performance in our prior experiments. As shown in the table below, integrating our $\alpha$-mixture assistant distribution exhibits higher performance compared to the DistiLLM-2 approach. These additional results demonstrate the broad applicability and robustness of AMiD across the evaluation metric and tasks.
>
> | Model                                      | GSM8K (↑) |
> |---------------|-----------|
> | Teacher (Qwen2.5-Math-7B-Inst)             | 89.3      |
> | Student (Qwen2.5-Math-1.5B-Inst)           | 74.3      |
> | DistiLLM-2 w/ $m$-mixture assistant          | 76.9      |
> | DistiLLM-2 w/ $\alpha$-mixture assistant          | **77.4**  |

---

> ### Author Response · Authors · 2025-11-21
> **Response to Reviewer yQkB [3/3]**
>
> >**W3. [Details of experimental configuration]** *The paper is not very clear on its hyper-parameters. For example, the author mentions that their approach work with any divergence functions, but it's not clear which one is used for Table 1. Also, the values of alpha/lambda is not shown on the table. The author mentions D_AB being used for Table 2, but how about Table 1. Even in Table 2, it says alpha is not -1/+1, but it doesn't state what alpha is. My main worry is that the authors may have spent lots of efforts on tuning these individually, which gave the approach an unfair advantage.*
>
> We apologize for not clearly specifying the experimental configurations used in the paper. In below, we provide a detailed clarification
> * (Divergence $D$) Following ABKD, we use the $\alpha_{AB}$-$\beta_{AB}$-divergence with $\alpha_{AB} = 0.2$ and $\beta_{AB} = 0.7$ as our default divergence. We have been explicitly stated in Appendix B.3.
> * (Search range for $\alpha$) We search $\alpha$ over $[-5.0, -3.0, -1.0, -0.5, 0.0, 0.5, 1.0]$. This range is motivated by our theoretical analysis regarding support union and mode-seeking behavior, as discussed in the manuscript. For further details of theoretical insight based $\alpha$ tuning guidelines, please refer to the global response 1. Throughout the paper, we refer to the region, which is newly provided in our framework AMiD, $\alpha \neq \pm 1$ to distinguish it from $\alpha = \pm 1$ which are already used in prior work. The values of $\alpha$ used in each experiment newly added to Table 7 in Appendix of the revised manuscript.
> * ($\lambda$) Consistent with DistiLLM, we adopt $\lambda = 0.1$ as the default setting.
> * (Others) All additional hyperparameters and experimental details basically follow the prior works.
>
> We have updated Appendix B.3 to include the set of hyperparameters used, and we have added key configuration details directly in each experimental results table to avoid any potential ambiguity. Furthermore, we investigated the relationship between $\alpha$ and $\lambda$. Please refer to global response 3 for the corresponding details.

---

### Official Review · Reviewer_Gj8D · 2025-11-01

**Soundness:** 2
**Presentation:** 3
**Contribution:** 2
**Rating:** 6
**Confidence:** 5

**Summary:**

This paper addresses the capacity gap between teacher and student models and the training instability caused by high-dimensional output in LLM knowledge distillation (KD). It proposes α-mixture assistant distribution and a unified distillation framework AMiD (α-mixture distillation). By systematically expanding the auxiliary distribution and divergence selection, it achieves better performance and training stability.

**Strengths:**

1. Generalization and Unification of Fragmented Methods.

Proposes a novel α-mixture assistant distribution by extending generalized \(f_\alpha\)-mean to KD, introducing the tunable parameter α. This generalizes prior isolated assistant distributions (m-mixture for α=-1, e-mixture for α=1) into a continuous, flexible family, covering new distributions (e.g., harmonic mean for α=3) not explored in LLM KD before.


2. Rigorous Theory and Comprehensive Experiments. Provides formal proofs for key properties (continuity of α-mixture distribution, optimality of AMiD, gradient analysis of f-divergence), establishing a solid mathematical foundation for the framework. Validates AMiD across diverse settings, task-agnostic (5 instruction-following datasets) and task-specific (translation, summarization, reasoning) distillation; multiple model scales (GPT-2 series, OpenLLaMA); various divergences (KL, RKL, AB) and SGO strategies. Results are consistent and statistically robust (5 random seeds), confirming reliability.

3. Addressing Core Limitations and Advancing Practical KD. Directly mitigates the capacity gap between teacher and student models and training instability from near-zero probabilities, two fundamental limitations of prior KD methods. AMiD’s flexibility (compatible with arbitrary divergences and datasets) and robustness (stable performance across λ values) make it adaptable to real-world LLM compression scenarios. The α parameter offers a simple control knob to balance quality and diversity, addressing a longstanding trade-off in generation tasks.

**Weaknesses:**

1. The lack of a systematic strategy for α parameter tuning limits its practicality. The paper demonstrates that the α parameter can control the balance between "pattern coverage" and "pattern finding" in the student model, but it fails to provide the basis for α selection and efficient tuning methods. The experiments only demonstrate the effects of fixing α (e.g., α=-5, -3, etc.), without explaining the optimal range and tuning logic of α under different tasks (translation/summarization/inference), different model capacity differences (e.g., 10B→0.1B), and different divergences (D), leading to extensive trial and error for users in practical applications.

2. The paper theoretically proves the optimality of AMiD under "perfect optimization," but two unresolved contradictions exist in practice: a) Conflict between divergence and α: Experiments show that the performance of D_RKL combined with α=1 is extremely poor (Avg. only 3.94 in Table 3), attributed to the narrow intersection of support sets, but no specific criteria are given on how to avoid this conflict;

b) Adaptation of optimizer and hyperparameters: The impact of different optimizers (such as AdamW, Lion) and learning rate scheduling on AMiD is not discussed, while in practice, optimizer selection significantly changes the gradient propagation effect of α-mixture.

3. The ablation experiments do not fully decompose the core contributions of AMiD: The interaction between α and λ is not verified: α is only tested with λ=0.1, without analyzing whether the optimal value of α changes under different λ (such as 0.3/0.7), and the collaborative tuning strategy between the two; the contributions of α-mixture and divergence are not isolated: Is the performance improvement of AMiD due to the auxiliary distribution of α expansion or the flexibility of divergence? Verification needs to be conducted by comparing "fixed α = ±1 (i.e., baseline auxiliary distribution) + arbitrary divergence" with "fixed divergence + α ≠ ±1".

4. The paper's baseline does not include cutting-edge LLM distillation methods since 2025, resulting in insufficient proof of advancement. For example, it does not compare distillation methods based on contrastive learning or reinforcement learning-based distillation (RL-KD), which may have already surpassed traditional divergence-based baselines in some scenarios.

**Questions:**

1. Adaptive Selection of α Parameter.  The paper demonstrates that α controls the trade-off between mode-seeking and mode-covering, but it does not provide a systematic method for selecting α in different scenarios (e.g., different task types, model size gaps, or data characteristics). Is there an adaptive strategy to determine α (e.g., curriculum learning-based scheduling, data-driven tuning) instead of manual adjustment? For small student models (e.g., 0.1B) vs. large gaps (e.g., 7B→0.5B), does the optimal α range differ significantly?

2. Extensibility to Larger Model Scales and Complex Tasks. The experiments focus on GPT-2 (up to 1.5B teacher) and OpenLLaMA-7B→3B. Have the authors tested AMiD on larger teacher models (e.g., 10B+ LLMs like LLaMA 3 70B) or smaller student models (e.g., <0.1B)? Additionally, do the performance gains hold for complex tasks such as code generation, mathematical reasoning with multi-step logic, or cross-lingual understanding?

3. Mitigation of Instability in Specific Divergence-α Combinations. The paper notes that the combination of \(D_{RKL}\) and α=1 leads to extremely poor performance due to support intersection issues. Have the authors explored mitigation strategies (e.g., adjusting λ, adding regularization terms, or modifying the assistant distribution’s normalization) instead of simply avoiding this combination? Are there general principles to avoid such conflicting pairs?

---

> ### Author Response · Authors · 2025-11-21
> **Response to Reviewer Gj8D [1/7]**
>
> We appreciate the constructive reviews and valuable comments. We address the concerns below.
>
> >**W1. [Lack of a selection strategy for $\alpha$]** *The lack of a systematic strategy for α parameter tuning limits its practicality. The paper demonstrates that the α parameter can control the balance between "pattern coverage" and "pattern finding" in the student model, but it fails to provide the basis for α selection and efficient tuning methods. The experiments only demonstrate the effects of fixing α (e.g., α=-5, -3, etc.), without explaining the optimal range and tuning logic of α under different tasks (translation/summarization/inference), different model capacity differences (e.g., 10B→0.1B), and different divergences (D), leading to extensive trial and error for users in practical applications.*
>
> *(The response for W1 is provided in global responses 1 and 2. For your convenience, the same content is attached below. Also, we introduce the systematic strategy for $\alpha$ selection in the global responses 1,2 and the response for Q1.)*
>
> Our proposed $\alpha$-mixture assistant distribution employs a new distribution design variable $\alpha$, which unifies and generalizes existing assistant distributions into a coherent distribution family. While this additional variable provides substantial flexibility and opens up unexplored assistant distribution candidates, identifying appropriate $\alpha$ values that achieve consistently stable and strong performance is important.
>
> We address this concern by providing tuning guidelines grounded in the theoretical properties of $\alpha$ analyzed in the manuscript. First, in Section 3.2, we show that the support of the $\alpha$-mixture assistant distribution varies with $\alpha$: $supp(r_\theta^{(\alpha,\lambda)}) = supp(p) \cup supp(q_\theta)$ when $\alpha < 1$, $supp(r_\theta^{(\alpha,\lambda)}) = supp(p) \cap supp(q_\theta)$ when $\alpha \geq 1$. In KD for LLMs, the teacher and student models often exhibit a capacity gap and produce high-dimensional outputs with many near-zero probabilities. As a result, they do not share a sufficiently large common support in general. For this reason, we recommend using $\alpha < 1$ in most practical settings, as this choice improves training stability and enables more reliable knowledge transfer.
>
> Furthermore, our gradient analysis and experimental results demonstrate that $\alpha$ enables to control the trade-off between mode-covering and mode-seeking behavior of the optimized student distribution. Under the $\alpha \leq 1$, increasing $\alpha$ relatively encourages mode-covering, thereby improving the diversity of outputs. Conversely, smaller $\alpha$ emphasizes mode-seeking, which enhances fidelity to the teacher. Therefore, for enhancing the teacher-student alignment and performance, we suggest using small $\alpha$ values. However, since too small $\alpha$ can induce high curvature in the geometry of interpolation path, which may reduce optimization efficiency, so such choices should be used with caution.
>
> Based on these theoretical insights, we basically consider $\alpha$ over [−5, −3, −1, −0.5, 0, 0.5, 1.0]. We observe consistently stable and strong performance across most of these candidates.
>
> ---
> >**W2(a). [Lack of mitigation strategy for the conflict between divergence and $\alpha$]** *The paper theoretically proves the optimality of AMiD under "perfect optimization," but two unresolved contradictions exist in practice: a) Conflict between divergence and α: Experiments show that the performance of D_RKL combined with α=1 is extremely poor (Avg. only 3.94 in Table 3), attributed to the narrow intersection of support sets, but no specific criteria are given on how to avoid this conflict;*
>
> The notion of perfect optimization used in Theorem 3.4 refers to the existence of the optimal parameter such that the $\alpha$-mixture assistant distribution becomes the teacher (or) student distribution. This realizability assumption is widely used in the optimality analyses and is not strong for LLMs, which have flexible function space due to extremely many parameters. Nevertheless, we modify the statement of Theorem 3.4 in the revised manuscript to remove possible confusion.
>
> *(The remain response for W2(a) is provided in global response 2. For your convenience, the same content is attached below.)*
>
> We have discussed this issue in Appendix D of the previous manuscript. In particular, for certain divergences, such as $D_{RKL}$, involve an expectation with respect to $r$ by the definition. When $\alpha = 1$, the support of $\alpha$-mixture assistant distribution equals to the intersection of the teacher and student distribution’s supports, which can be narrow in high-dimensional LLM outputs. Consequently, this small support might induce unstable training and insufficient knowledge transfer. Therefore, restricting $\alpha < 1$ is simple and guarantees the theoretical stability in most cases.

---

> ### Author Response · Authors · 2025-11-21
> **Response to Reviewer Gj8D [2/7]**
>
> >**W2(b). [Robustness to the optimizer and learning rate scheduling]** *b) Adaptation of optimizer and hyperparameters: The impact of different optimizers (such as AdamW, Lion) and learning rate scheduling on AMiD is not discussed, while in practice, optimizer selection significantly changes the gradient propagation effect of α-mixture.*
>
> Following prior work [1, 2], we use the AdamW optimizer and cosine learning rate scheduling. We have confirmed that this configuration was not included and have added it to Appendix B.3. We agree with the reviewer that the choice of optimizer and scheduling affects optimization dynamics. To assess the robustness of AMiD to such configuration choices, we additionally conducted experiments using the Lion optimizer [3] and Noam learning rate scheduling. As shown in the table below, AMiD consistently achieves strong performance regardless of the optimizer or learning rate scheduling, demonstrating robustness across optimization settings.
>
> | | Val. (↑) | Dolly Eval (↑)       | Self Inst (↑)       | Vicuna (↑)        | Super NI (↑)      | UnNI (↑)         | Average (↑) |
> |---------------------|-------------|-----------------------|-----------------------|---------------------|---------------------|-------------------|-------------|
> | **Lion optimizer**            |             |                       |                       |                     |                     |                   |             |
> | No assistant        | 28.38       | 26.02          | 12.19          | 17.24        | 26.29       | 28.88      | 22.12       |
> | AMiD ($\alpha = -1$)       | 28.68       | 25.29           | 12.21          | 17.81       | 24.82        | 27.87      | 21.60       |
> | AMiD ($\alpha = 1$)       | 24.98       | 22.33         | 9.50        | 15.50     | 15.71        | 18.01       | 16.21       |
> | AMiD ($\alpha \neq \pm 1$)       | 27.85       | 26.14           | 12.55         | 17.25        | 28.28         | 29.69       | **22.78**   |
> | **Noam scheduling**            |             |                       |                       |                     |                     |                   |             |
> | No assistant        | 28.42       | 25.83          | 13.35          | 16.43         | 28.30         | 29.90       | 22.76       |
> | AMiD ($\alpha = -1$)       | 28.91       | 26.02          | 14.07         | 17.09         | 27.78        | 29.49       | 22.89       |
> | AMiD ($\alpha = 1$)       | 28.33       | 25.59          | 13.61         | 16.25        | 26.42       | 28.29       | 22.03       |
> | AMiD ($\alpha \neq \pm 1$)       | 29.39       | 26.12          | 13.07         | 16.53        | 29.36       | 30.86     | **23.19**   |
>
> ---
> >**W3(a). [Interaction between $\alpha$ and $\lambda$]** *The ablation experiments do not fully decompose the core contributions of AMiD: The interaction between α and λ is not verified: α is only tested with λ=0.1, without analyzing whether the optimal value of α changes under different λ (such as 0.3/0.7), and the collaborative tuning strategy between the two;
>
> *(The response for W3(a) is provided in global response 3. For your convenience, the same content is attached below.)*
>
> We employ $\lambda = 0.1$ as our default setting by following the prior work [1]. To further investigate the empirical relationship between $\alpha$ and $\lambda$, we conducted additional experiments on $\lambda = 0.5, 0.9$ with $D_{AB}(p||r_\theta^{(\alpha,\lambda)})$. The table below presents the average performance of five instruction-following datasets among the various $\alpha$ and $\lambda$ combinations. The analyses of experimental results are as follow: *(Continued below)*
>
> [1] DistiLLM: Towards Streamlined Distillation for Large Language Models
>
> [2] ABKD: Pursuing a Proper Allocation of the Probability Mass in Knowledge Distillation via α-β-Divergence
>
> [3] Symbolic Discovery of Optimization Algorithms

---

> ### Author Response · Authors · 2025-11-21
> **Response to Reviewer Gj8D [3/7]**
>
> *(Continued)*
>
> * Across all tested values of $\lambda$, using smaller $\alpha$ consistently achieves higher performance. This observation aligns with our theoretical analysis indicating that smaller $\alpha$ relatively induces a more mode-seeking behavior.
> * When $\lambda$ is too large ($\lambda = 0.9$), performance slightly degrades and exhibits a larger standard deviation. We attribute this to the $\alpha$-mixture assistant distribution being overly close to the teacher distribution, which (1) limits effective knowledge transfer and (2) makes the optimization more sensitive to curvature variations. In contrast, $\lambda = 0.5$ achieves both high and stable performance, demonstrating that choosing a midpoint provides robustness against curvature changes.
> * Compared to $\lambda$, the performance is generally more robust to changes in $\alpha$, as indicated by lower standard deviations. However, the $\alpha$ values close to 1 show higher standard deviations. We conjecture that this instability arises because the change of mixing coefficient ($\lambda$) between the teacher and student distributions makes hard mode-covering.
>
> | | $\alpha = -5.0$ | $\alpha = -3.0$ | $\alpha = -1.0$ | $\alpha = -0.5$ | $\alpha = 0.0$ | $\alpha = 0.5$ | $\alpha = 1.0$ | Avg. along $\alpha$ | std. along $\alpha$ |
> |-------------------------|----------|----------|----------|----------|---------|---------|---------|---------------|----------------|
> | $\lambda = 0.1$ | 23.40    | 23.10    | 22.45    | 22.51    | 22.25   | 21.87   | 15.92   | 21.64         | 2.58           |
> | $\lambda = 0.5$ | 23.29    | 23.02    | 23.13    | 22.80    | 21.87   | 20.02   | 17.46   | 21.66         | 2.17           |
> | $\lambda = 0.9$ | 23.12    | 23.25    | 22.89    | 22.75    | 21.99   | 18.58   | 15.04   | 21.09         | 3.12           |
> | Avg. along $\lambda$ | 23.27    | 23.12    | 22.82    | 22.69    | 22.04   | 20.16   | 16.14   | —             | —              |
> | std. along $\lambda$ | 0.14     | 0.12     | 0.34     | 0.16     | 0.19    | 1.65    | 1.22    | —             | — |
>
> We present these summarized results as Figure 6 and entire results as Table 14 in the revised manuscript.
>
> ---
> >**W3(b). [Combined contributions of $\alpha$-mixture and divergence]** * the contributions of α-mixture and divergence are not isolated: Is the performance improvement of AMiD due to the auxiliary distribution of α expansion or the flexibility of divergence? Verification needs to be conducted by comparing "fixed α = ±1 (i.e., baseline auxiliary distribution) + arbitrary divergence" with "fixed divergence + α ≠ ±1".*
>
> We have presented the experimental results of the combinations between the divergence and $\alpha$ in Tables 3 and 9 in the revised manuscript (Tables 3 and 7 in the previous manuscript). When we fix the divergence and varying $\alpha$, we observe that $\alpha \neq \pm 1$ consistently achieves the highest performance, which is the newly provided and explored in our AMiD framework. It demonstrates that extending the assistant distribution beyond the previous assistant distributions provides effective performance gains.
>
> Moreover, based on Tables 3 and 9 in the revised manuscript, we summarize the reviewer’s suggested comparison between arbitrary divergences with $\alpha = \pm 1$ and fixed divergences with $\alpha \neq \pm 1$ in the table below:
>
> | $\alpha$ | Divergence | Avg. (↑)|
> |----|----|---|
> | $\alpha = -1$ (best) | $D_{AB}(p, r_\theta^{(\alpha,\lambda)})$ | 22.45 |
> | $\alpha = 1$ (best) | $D_{KL}(q_\theta, r_\theta^{(\alpha,\lambda)})$ | 21.37 |
> | $\alpha \neq \pm 1$ (lowest among the best) | $D_{RKL}(p, r_\theta^{(\alpha,\lambda)})$ | 22.30 |
> | $\alpha \neq \pm 1$ (best among the best) | $D_{AB} (p, r_\theta^{(\alpha,\lambda)})$ | 23.40 |
>
> We highlight that the best performance of $\alpha = \pm 1$ are comparable to the lowest among the best performances of $\alpha \neq \pm 1$, and in most best cases $\alpha \neq \pm 1$ provides substantially higher performance. These results further validate the effectiveness of introducing the $\alpha$-mixture assistant distribution.

---

> ### Author Response · Authors · 2025-11-21
> **Response to Reviewer Gj8D [4/7]**
>
> > **W4. [Comparison with recent methods]** *The paper's baseline does not include cutting-edge LLM distillation methods since 2025, resulting in insufficient proof of advancement. For example, it does not compare distillation methods based on contrastive learning or reinforcement learning-based distillation (RL-KD), which may have already surpassed traditional divergence-based baselines in some scenarios*
>
> First, the divergence-based distillation baselines used in our manuscript, TAID and ABKD, are published in 2025. Therefore, our experiments already include comparisons against the most recent divergence-based approaches.
>
> Following the reviewer’s concern, we additionally compare our method against contrastive approaches applied to KD, specifically DPKD [4] and DistiLLM-2 [5]. We refer to Table 1 of the DPKD paper and Table 12 of the DistiLLM-2 paper, and for fairness, we compare these results to AMiD using on-policy SGO variants.
>
> As shown in the table below, AMiD achieves competitive performances in databricks-dolly-15k and self-instruct datasets, and higher performance in super-natural instruction dataset with significant gap. This comparison demonstrates that AMiD remains effective even when compared to the latest contrastive-based distillation methods.
>
> | Model                                  | Dolly Eval (↑) | Self Inst (↑) | Super NI (↑) |
> |-----|------|-----|----|
> | DPKD (Official report)                 | 24.60         | 13.80          | 25.40         |
> | DPKD (Reported in DistiLLM-2)          | 6.85          | -              | -             |
> | DistiLLM-2 (Reported in DistiLLM-2)    | 26.37         | -              | -             |
> | AMiD   | 26.46         | 13.62          | 28.13         |
>
> Moreover, the proposed $\alpha$-mixture assistant distribution can be incorporated into contrastive-based distillation methods since it is a theoretically valid probability distribution. To validate this compatibility, we replace the assistant distribution of DistiLLM-2 ($m$-mixture) with our $\alpha$-mixture assistant distribution, and conduct the instruction-following experiment under the DistiLLM-2 setting. We set $\alpha = -5.0$, which consistently performed well in our experimental results, and faithfully follow the DistiLLM-2’s hyperparameters.
>
> The table below presents the performance of evaluating three instruction-following benchmark datasets. DistiLLM-2 w/ $\alpha$-mixture assistant distribution exhibits the higher performance among all tested benchmark datasets. These results support the applicability of the $\alpha$-mixture assistant distribution.
>
> | Model | AlpacaEval WR (%) (↑) | Evol-Inst WR (%) (↑) | UltraFeed WR (%) (↑) | Average (↑) |
> |---------|-------|-----|-------|-------------|
> | Teacher (Qwen2.5-7B-Instruct)                 | 93.7                   | 89.6                  | 80.8                   | 88.0        |
> | Student (Qwen2.5-1.5B-Instruct)               | 64.2                   | 46.2                  | 40.0                   | 50.1        |
> | DistiLLM-2 w/ $m$-mixture assistant     | 79.5                   | 69.0                  | 62.6                   | 70.4        |
> | DistiLLM-2 w/ $\alpha$-mixture assistant     | 80.7                   | 71.0                  | 63.3                   | **71.7**    |
>
> [4] Direct Preference Knowledge Distillation for Large Language Models
>
> [5] DISTILLM-2: A Contrastive Approach Boosts the Distillation of LLMs

---

> ### Author Response · Authors · 2025-11-21
> **Response to Reviewer Gj8D [5/7]**
>
> >**Q1. [Adaptive strategy for $\alpha$ selection]** *Adaptive Selection of α Parameter. The paper demonstrates that α controls the trade-off between mode-seeking and mode-covering, but it does not provide a systematic method for selecting α in different scenarios (e.g., different task types, model size gaps, or data characteristics). Is there an adaptive strategy to determine α (e.g., curriculum learning-based scheduling, data-driven tuning) instead of manual adjustment? For small student models (e.g., 0.1B) vs. large gaps (e.g., 7B→0.5B), does the optimal α range differ significantly?*
>
> *(The response for Q1 is provided in global responses 1 and 2. For your convenience, the same content is attached below.)*
>
> The theoretical insights based $\alpha$ tuning guidelines efficiently exclude low potential candidates. However, applying a single fixed global $\alpha$ value can still be sub-optimal in certain cases.
>
> To address this concern, we introduce a curriculum-based adaptive $\alpha$ scheduling based on the degree of overlap between token-level teacher distribution $p(y_l | y_{<l}, x)$ and student distribution $q_\theta(y_l | y_{<l}, x)$. The intuition is that when the teacher and student distributions are highly overlapped, we encourage mode-covering to align further, whereas when the overlap is low, we enhance mode-seeking to find the mode first.
>
> We define token-level overlap as $ovl_{i,l} := \sum_{y_l} \min(p(y_l | y_{<l}, x), q_\theta(y_l | y_{<l}, x))$. Obviously, the overlap value $ovl_{i,l}$ is bounded into $[0, 1]$. Also, $ovl_{i,l}$ approaches $0$ when $p(y_l | y_{<l}, x)$ and $q_\theta(y_l | y_{<l}, x)$ significantly differ, and approaches $1$ when they are well-aligned. Furthermore, $ovl_{i,l}$ can be expressed in terms of total variation distance $1-TVD(p(y_l | y_{<l}, x), q_\theta(y_l | y_{<l}, x))$, which provides same interpretation.
>
> Given the predefined $\alpha_{min}$ and $\alpha_{max}$, we set the token-level $\alpha_{i,l}$ as the linearly increasing value along the line passing through $(0, \alpha_{min})$ and $(1, \alpha_{max})$ i.e. $\alpha_{i,l} \leftarrow (\alpha_{max} - \alpha_{min}) * ovl_{i,l} + \alpha_{min}$. Under this idea, when the teacher and student distributions differ substantially, the $ovl_{i,l}$ becomes small, leading to a smaller assigned $\alpha$, which strengthens mode-seeking. Conversely, when the teacher and student distributions are similar, both $ovl_{i,l}$ and $\alpha$ have larger values, thereby reinforcing mode-covering. This mechanism systematically determines $\alpha$ by combining the degree of alignment between the teacher and student distribution, which continuously changes through training, with the theoretical characteristics of $\alpha$. Below pseudocode presents the batch-wise training scheme of AMiD with proposed overlap-based adaptive $\alpha$ scheduling.
> ```
> Given: Dataset $\mathcal{D}$, Divergence $D$, $\alpha_min$, $\alpha_max$, $\lambda$
> 1. For each iteration
> 2.     Sample mini-batch $B$ from $\mathcal{D}$
> 3.     Obtain batch-level distribution of teacher $p$ and student $q$    // [B, L, V] shape
> 4.     Calculate batch-level overlap value $ovl \leftarrow \min(p, q).sum(-1)$    // [B, L] shape
> 5.     Calculate batch-level $\alpha \leftarrow (\alpha_{max} - \alpha_{min}) * ovl + \alpha_{min}$    // [B, L] shape
> 6.     Reshape $\alpha$    // [B, L, V] shape
> 7.     Calculate $r_\theta^{(\alpha, \lambda)}$    // [B, L, V] shape
> 8.   Update $\theta$ by minimizing $D(\cdot, r_\theta^{(\alpha, \lambda)})$    // either $p$ or $q_\theta$ is possible.
> ```
>
> The table below presents the performance of fixed $\alpha$ and overlap-based adaptive $\alpha$ scheduling under the $D_{AB}$ with $\alpha_{AB}=0.2, \beta_{AB}=0.7$ and $\lambda = 0.1$. For fixed approach, we set $\alpha = -5.0$ which is best performer among the fixed methods. For adaptive approach, we set $\alpha_{min} = -5.0, \alpha_{max} = -1.0$. As shown in the table below, the overlap-based adaptive $\alpha$ scheduling achieves higher performance than the best fixed global $\alpha$ strategy, demonstrating its effectiveness.
>
> | | Val. (↑) | Dolly Eval (↑) | Self Inst (↑) | Vicuna (↑) | Super NI (↑) | UnNI (↑) | Avg. (↑) |
> |-|-|-|-|-|-|-|-|
> | AMiD (Fixed $\alpha$) | 29.24 | 26.44 | 13.74 | 16.76 | 29.71 | 30.35 | 23.40 |
> | AMiD (Overlap-based adaptive $\alpha$) | 29.31 | 26.50 | 14.02 | 16.59 | 29.87 | 30.60 | **23.52** |
>
> We also compare the wall-clock time for each training step. As shown in the table below, the proposed overlap-based adaptive $\alpha$ scheduling shows a negligible time complexity increase compared to no-assistant baseline (ABKD) and fixed $\alpha$ approach due to the simple structure while theory-based approach.
>
> | | Avg. of wall-clock time of single training step (sec) |
> |-|-|
> | ABKD | 1.17 |
> | AMiD (fixed $\alpha$) | 1.26 |
> | AMiD (overlap-based adaptive $\alpha$) | 1.29 |
> *(Continued below)*

---

> ### Author Response · Authors · 2025-11-21
> **Response to Reviewer Gj8D [6/7]**
>
> *(Continued)* We believe that more sophisticated adaptive $\alpha$ scheduling, such as bi-level optimization $\min_{(\alpha, \lambda)} D(p, q_{\theta’})$ where $\theta’ = argmin_{\theta} D(p, r_{\theta}^{(\alpha, \lambda)})$, could further enhance performance. Investigating such adaptive $\alpha$ and/or $\lambda$ scheduling could be interesting research direction.
>
> Referring to Table 7 in the revised manuscript, which summarizes the $\alpha$ values used in the experiments, the optimal range for small students and large gaps does not differ significantly within the range of $\alpha$ we tested.
>
> ---
> > **Q2. [Extensibility to teacher model and complex tasks]** *Extensibility to Larger Model Scales and Complex Tasks. The experiments focus on GPT-2 (up to 1.5B teacher) and OpenLLaMA-7B→3B. Have the authors tested AMiD on larger teacher models (e.g., 10B+ LLMs like LLaMA 3 70B) or smaller student models (e.g., <0.1B)? Additionally, do the performance gains hold for complex tasks such as code generation, mathematical reasoning with multi-step logic, or cross-lingual understanding?*
>
> We conjecture that $\alpha$-mixture assistant distribution remains effective even when distillation form large teacher models since it provides many possible assistant distribution, which mitigate the capacity gap. To verify it, we conduct an additional experiment by distilling Qwen2.5-14B-Instruct into Qwen2.5-1.5B-Instruct under the instruction-following task. We follow the DistiLLM-2 implementation details as mentioned in our response to Weakness 4. As shown in the table below, employing the $\alpha$-mixture assistant distribution enhances the performance even in the large size teacher KD.
>
> | Model  | AlpacaEval WR (%) (↑) | Evol-Inst WR (%) (↑) | UltraFeed WR (%) (↑) | Average (↑) |
> |----------------------------------------------|------------------------|-----------------------|------------------------|-------------|
> | Teacher (Qwen2.5-14B-Instruct)               | 95.7                   | 92.2                  | 84.9                   | 90.9        |
> | Student (Qwen2.5-1.5B-Instruct)              | 64.3                   | 47.2                  | 40.4                   | 50.6        |
> | DistiLLM-2 w/ $m$-mixture assistant            | 80.2                   | 70.4                  | 61.7                   | 70.8        |
> | DistiLLM-2 w/ $\alpha$-mixture assistant            | 81.3                   | 71.1                  | 63.6                   | **72.0**    |
>
> To further validate the generality of $\alpha$-mixture assistant distribution across tasks, we conducted the experiments on translation, summarization, and reasoning tasks, with reported results in Table 2 of the manuscript. Furthermore, we conduct the additional experiment on mathematical reasoning and code generation. For the reasoning task, we train on 50k randomly selected instances from MetaMathQA and evaluate on GSM8K, using Qwen2.5-Math-7B-Instruct as the teacher and Qwen2.5-Math-1.5B-Instruct as the student. For the code generation task, we train with WizardCoder while utilize HumanEval and MBPP for the evaluation. We distill Qwen2.5-Coder-7B-Instruct into Qwen2.5-Coder-1.5B-Instruct. Similar with the instruction-following task, we use the DistiLLM-2’s configuration without applying any modifications. As shown in the table below, incorporating our $\alpha$-mixture assistant distribution exhibits competitive performances or further performance improvements over the DistiLLM-2.
>
> These experimental results demonstrate both the scalability of AMiD to larger teacher models and the generality across diverse tasks, including instruction following, translation, summarization, mathematical reasoning, and code generation.
>
> | Model       | GSM8K (↑) |
> |------------|-----------|
> | Teacher (Qwen2.5-Math-7B-Inst)             | 89.3      |
> | Student (Qwen2.5-Math-1.5B-Inst)           | 74.3      |
> | DistiLLM-2 w/ $m$-mixture assistant          | 76.9      |
> | DistiLLM-2 w/ $\alpha$-mixture assistant          | **77.4**  |
>
>
> | Model    | HumanEval pass@1 (↑) | MBPP pass@1 (↑) | Average pass@1 (↑) |
> |---------------|----------|---|---------|
> | Teacher (Qwen2.5-Coder-7B-Inst)            | 90.9                   | 83.1             | 87.0                 |
> | Student (Qwen2.5-Coder-1.5B-Inst)          | 70.7                   | 69.3             | 70.0                 |
> | DistiLLM-2 w/ $m$-mixture assistant          | 72.0                   | 74.6             | 73.3                 |
> | DistiLLM-2 w/ $\alpha$-mixture assistant          | 73.2                   | 73.5             | **73.4**             |

---

> ### Author Response · Authors · 2025-11-21
> **Response to Reviewer Gj8D [7/7]**
>
> >**Q3. [Alleviation the confict of divergence and $\alpha$]** *Mitigation of Instability in Specific Divergence-α Combinations. The paper notes that the combination of (D_{RKL}) and α=1 leads to extremely poor performance due to support intersection issues. Have the authors explored mitigation strategies (e.g., adjusting λ, adding regularization terms, or modifying the assistant distribution’s normalization) instead of simply avoiding this combination? Are there general principles to avoid such conflicting pairs?*
>
> *(The response for Q3 is provided in global response 2. For your convenience, the same content is attached below.)*
>
> Since the primary cause of this issue is the narrow support of the $\alpha$-mixture assistant distribution, we conjecture that applying distribution softening technique could alleviate the instability even for such problematic combinations. To verify this conjecture, we employ temperature $T > 1$, which is a widely used flattening technique in various area. The table below shows the performance when using various temperature values under $D_{RKL}(p || r_\theta^{(\alpha,\lambda)})$ with $\alpha = 1$. The results exhibit that introducing the temperature leads to stable training and can even yield high performance with an appropriately chosen value. However, large temperature causes an over-flattening effect, inducing large shift in the $\alpha$-mixture assistant distribution and consequently degrading performance. Overall, we demonstrate that employing temperature scaling can empirically mitigate the instability associated with problematic combinations under the appropriate temperature value.
>
> | | Val. (↑) | Dolly Eval (↑) | Self Inst (↑) | Vicuna (↑) | Super NI (↑) | UnNI (↑) | Average (↑) |
> |-|-|-|-|-|-|-|-|
> | AMiD ($\alpha = 1.0, T = 1.0$) | 0.16 | 4.27 | 2.81 | 9.12 | 1.64 | 1.84 | 3.94 |
> | AMiD ($\alpha = 1.0, T = 1.5$) | 27.40 | 24.61 | 11.55 | 17.11 | 21.41 | 23.39 | 19.61 |
> | AMiD ($\alpha = 1.0, T = 2.0$) | 28.64 | 26.83 | 12.74 | 17.62 | 23.56 | 27.01 | 21.55 |
> | AMiD ($\alpha = 1.0, T = 5.0$) | 28.78 | 26.78 | 12.43 | 17.30 | 25.87 | 27.74 | 22.02 |
> | AMiD ($\alpha = 1.0, T = 10.0$) | 27.33 | 24.73 | 12.11 | 16.95 | 23.65 | 26.80 | 20.85 |

---

### Author Response · Authors · 2025-11-21
**Global Response [1/3]**

>**GQ1. Is there a range or tuning strategy that can be used to effectively identify generally good values of $\alpha$? Moreover, are there any strategies for selecting $\alpha$ in an adaptive manner?**

**[Theoretical insight based tuning guidelines for $\alpha$]** Our proposed $\alpha$-mixture assistant distribution employs a new distribution design variable $\alpha$, which unifies and generalizes existing assistant distributions into a coherent distribution family. While this additional variable provides substantial flexibility and opens up unexplored assistant distribution candidates, identifying appropriate $\alpha$ values that achieve consistently stable and strong performance is important.

We address this concern by providing tuning guidelines grounded in the theoretical properties of $\alpha$ analyzed in the manuscript. First, in Section 3.2, we show that the support of the $\alpha$-mixture assistant distribution varies with $\alpha$: $supp(r_\theta^{(\alpha,\lambda)}) = supp(p) \cup supp(q_\theta)$ when $\alpha < 1$, $supp(r_\theta^{(\alpha,\lambda)}) = supp(p) \cap supp(q_\theta)$ when $\alpha \geq 1$. In KD for LLMs, the teacher and student models often exhibit a capacity gap and produce high-dimensional outputs with many near-zero probabilities. As a result, they do not share a sufficiently large common support in general. For this reason, we recommend using $\alpha < 1$ in most practical settings, as this choice improves training stability and enables more reliable knowledge transfer.

Furthermore, our gradient analysis and experimental results demonstrate that $\alpha$ enables to control the trade-off between mode-covering and mode-seeking behavior of the optimized student distribution. Under the $\alpha \leq 1$, increasing $\alpha$ relatively encourages mode-covering, thereby improving the diversity of outputs. Conversely, smaller $\alpha$ emphasizes mode-seeking, which enhances fidelity to the teacher. Therefore, for enhancing the teacher-student alignment and performance, we suggest using small $\alpha$ values. However, since too small $\alpha$ can induce high curvature in the geometry of interpolation path, which may reduce optimization efficiency, so such choices should be used with caution.

Based on these theoretical insights, we basically consider $\alpha$ over [−5, −3, −1, −0.5, 0, 0.5, 1.0]. We observe consistently stable and strong performance across most of these candidates.

---
**[Overlap-based adaptive $\alpha$ scheduling]** The theoretical insights based $\alpha$ tuning guidelines efficiently exclude low potential candidates. However, applying a single fixed global $\alpha$ value can still be sub-optimal in certain cases.

To address this concern, we introduce a curriculum-based adaptive $\alpha$ scheduling based on the degree of overlap between token-level teacher distribution $p(y_l | y_{<l}, x)$ and student distribution $q_\theta(y_l | y_{<l}, x)$. The intuition is that when the teacher and student distributions are highly overlapped, we encourage mode-covering to align further, whereas when the overlap is low, we enhance mode-seeking to find the mode first.

We define token-level overlap as $ovl_{i,l} := \sum_{y_l} \min(p(y_l | y_{<l}, x), q_\theta(y_l | y_{<l}, x))$. Obviously, the overlap value $ovl_{i,l}$ is bounded into $[0, 1]$. Also, $ovl_{i,l}$ approaches $0$ when $p(y_l | y_{<l}, x)$ and $q_\theta(y_l | y_{<l}, x)$ significantly differ, and approaches $1$ when they are well-aligned. Furthermore, $ovl_{i,l}$ can be expressed in terms of total variation distance $1-TVD(p(y_l | y_{<l}, x), q_\theta(y_l | y_{<l}, x))$, which provides same interpretation.

Given the predefined $\alpha_{min}$ and $\alpha_{max}$, we set the token-level $\alpha_{i,l}$ as the linearly increasing value along the line passing through $(0, \alpha_{min})$ and $(1, \alpha_{max})$ i.e. $\alpha_{i,l} \leftarrow (\alpha_{max} - \alpha_{min}) * ovl_{i,l} + \alpha_{min}$. Under this idea, when the teacher and student distributions differ substantially, the $ovl_{i,l}$ becomes small, leading to a smaller assigned $\alpha$, which strengthens mode-seeking. Conversely, when the teacher and student distributions are similar, both $ovl_{i,l}$ and $\alpha$ have larger values, thereby reinforcing mode-covering. This mechanism systematically determines $\alpha$ by combining the degree of alignment between the teacher and student distribution, which continuously changes through training, with the theoretical characteristics of $\alpha$. *(Continued below)*

---

### Author Response · Authors · 2025-11-21
**Global Response [2/3]**

*(Continued)* Below pseudocode presents the batch-wise training scheme of AMiD with proposed overlap-based adaptive $\alpha$ scheduling.
```
Given: Dataset $\mathcal{D}$, Divergence $D$, $\alpha_min$, $\alpha_max$, $\lambda$
1. For each iteration
2.     Sample mini-batch $B$ from $\mathcal{D}$
3.     Obtain batch-level distribution of teacher $p$ and student $q$    // [B, L, V] shape
4.     Calculate batch-level overlap value $ovl \leftarrow \min(p, q).sum(-1)$    // [B, L] shape
5.     Calculate batch-level $\alpha \leftarrow (\alpha_{max} - \alpha_{min}) * ovl + \alpha_{min}$    // [B, L] shape
6.     Reshape $\alpha$    // [B, L, V] shape
7.     Calculate $r_\theta^{(\alpha, \lambda)}$    // [B, L, V] shape
8.   Update $\theta$ by minimizing $D(\cdot, r_\theta^{(\alpha, \lambda)})$    // either $p$ or $q_\theta$ is possible.
```

The table below presents the performance of fixed $\alpha$ and overlap-based adaptive $\alpha$ scheduling under the $D_{AB}$ with $\alpha_{AB}=0.2, \beta_{AB}=0.7$ and $\lambda = 0.1$. For fixed approach, we set $\alpha = -5.0$ which is best performer among the fixed methods. For adaptive approach, we set $\alpha_{min} = -5.0, \alpha_{max} = -1.0$. As shown in the table below, the overlap-based adaptive $\alpha$ scheduling achieves higher performance than the best fixed global $\alpha$ strategy, demonstrating its effectiveness.

| | Val. (↑) | Dolly Eval (↑) | Self Inst (↑) | Vicuna (↑) | Super NI (↑) | UnNI (↑) | Avg. (↑) |
|-|-|-|-|-|-|-|-|
| AMiD (Fixed $\alpha$) | 29.24 | 26.44 | 13.74 | 16.76 | 29.71 | 30.35 | 23.40 |
| AMiD (Overlap-based adaptive $\alpha$) | 29.31 | 26.50 | 14.02 | 16.59 | 29.87 | 30.60 | **23.52** |

We also compare the wall-clock time for each training step. As shown in the table below, the proposed overlap-based adaptive $\alpha$ scheduling shows a negligible time complexity increase compared to no-assistant baseline (ABKD) and fixed $\alpha$ approach due to the simple structure while theory-based approach.

| | Avg. of wall-clock time of single training step (sec) |
|-|-|
| ABKD | 1.17 |
| AMiD (fixed $\alpha$) | 1.26 |
| AMiD (overlap-based adaptive $\alpha$) | 1.29 |

We believe that more sophisticated adaptive $\alpha$ scheduling, such as bi-level optimization $\min_{(\alpha, \lambda)} D(p, q_{\theta’})$ where $\theta’ = argmin_{\theta} D(p, r_{\theta}^{(\alpha, \lambda)})$, could further enhance performance. Investigating such adaptive $\alpha$ and/or $\lambda$ scheduling could be interesting research direction.

---
>**GQ2. Why does specific combination of divergence and $\alpha$ e.g., $D_{RKL}(p || r_\theta^{(\alpha,\lambda)})$ with $\alpha = 1$ exhibit low performance? Is there a mitigation strategy for this combination?**

**[Narrow support is the primary cause]** We have discussed this issue in Appendix D of the previous manuscript. In particular, for certain divergences, such as $D_{RKL}$, involve an expectation with respect to $r$ by the definition. When $\alpha = 1$, the support of $\alpha$-mixture assistant distribution equals to the intersection of the teacher and student distribution’s supports, which can be narrow in high-dimensional LLM outputs. Consequently, this small support might induce unstable training and insufficient knowledge transfer. Therefore, restricting $\alpha < 1$ is simple and guarantees the theoretical stability in most cases.

**[Mitigation via distribution softening]** Since the primary cause of this issue is the narrow support of the $\alpha$-mixture assistant distribution, we conjecture that applying distribution softening technique could alleviate the instability even for such problematic combinations. To verify this conjecture, we employ temperature $T > 1$, which is a widely used flattening technique in various area. The table below shows the performance when using various temperature values under $D_{RKL}(p || r_\theta^{(\alpha,\lambda)})$ with $\alpha = 1$. The results exhibit that introducing the temperature leads to stable training and can even yield high performance with an appropriately chosen value. However, large temperature causes an over-flattening effect, inducing large shift in the $\alpha$-mixture assistant distribution and consequently degrading performance. Overall, we demonstrate that employing temperature scaling can empirically mitigate the instability associated with problematic combinations under the appropriate temperature value.

| | Val. (↑) | Dolly Eval (↑) | Self Inst (↑) | Vicuna (↑) | Super NI (↑) | UnNI (↑) | Average (↑) |
|-|-|-|-|-|-|-|-|
| AMiD ($\alpha = 1.0, T = 1.0$) | 0.16 | 4.27 | 2.81 | 9.12 | 1.64 | 1.84 | 3.94 |
| AMiD ($\alpha = 1.0, T = 1.5$) | 27.40 | 24.61 | 11.55 | 17.11 | 21.41 | 23.39 | 19.61 |
| AMiD ($\alpha = 1.0, T = 2.0$) | 28.64 | 26.83 | 12.74 | 17.62 | 23.56 | 27.01 | 21.55 |
| AMiD ($\alpha = 1.0, T = 5.0$) | 28.78 | 26.78 | 12.43 | 17.30 | 25.87 | 27.74 | 22.02 |
| AMiD ($\alpha = 1.0, T = 10.0$) | 27.33 | 24.73 | 12.11 | 16.95 | 23.65 | 26.80 | 20.85 |

---

### Author Response · Authors · 2025-11-21
**Global Response [3/3]**

>**GQ3. What is the relationship between $\alpha$ and $\lambda$?**

**[Theoretical relationship between $\alpha$ and $\lambda$]** The proposed $\alpha$-mixture assistant distribution is the mixture of the teacher distribution $p$ and student distribution $q_\theta$ via the generalized $f_\alpha$-mean. From the information geometry perspective, $\alpha$ controls the geometry of interpolation path and $\lambda$ determines the portion of interpolation as depicted in Figure 3a of the revised manuscript (Figure 2a of the revised manuscript). Since the generalized $f_\alpha$-mean only depends on $\alpha$, when $\alpha$ is fixed, $\lambda$ only enables to adjust the ratio between $p$ and $q_\theta$ along the determined path.

---
**[Empirical relationship between $\alpha$ and $\lambda$]** We employ $\lambda = 0.1$ as our default setting by following the prior work [1]. To further investigate the empirical relationship between $\alpha$ and $\lambda$, we conducted additional experiments on $\lambda = 0.5, 0.9$ with $D_{AB}(p||r_\theta^{(\alpha,\lambda)})$. The table below presents the average performance of five instruction-following datasets among the various $\alpha$ and $\lambda$ combinations. The analyses of experimental results are as follow:

* Across all tested values of $\lambda$, using smaller $\alpha$ consistently achieves higher performance. This observation aligns with our theoretical analysis indicating that smaller $\alpha$ relatively induces a more mode-seeking behavior.
* When $\lambda$ is too large ($\lambda = 0.9$), performance slightly degrades and exhibits a larger standard deviation. We attribute this to the $\alpha$-mixture assistant distribution being overly close to the teacher distribution, which (1) limits effective knowledge transfer and (2) makes the optimization more sensitive to curvature variations. In contrast, $\lambda = 0.5$ achieves both high and stable performance, demonstrating that choosing a midpoint provides robustness against curvature changes.
* Compared to $\lambda$, the performance is generally more robust to changes in $\alpha$, as indicated by lower standard deviations. However, the $\alpha$ values close to 1 show higher standard deviations. We conjecture that this instability arises because the change of mixing coefficient ($\lambda$) between the teacher and student distributions makes hard mode-covering.

We present these summarized results as Figure 6 and entire results as Table 14 in the revised manuscript.

| | $\alpha = -5.0$ | $\alpha = -3.0$ | $\alpha = -1.0$ | $\alpha = -0.5$ | $\alpha = 0.0$ | $\alpha = 0.5$ | $\alpha = 1.0$ | Avg. along $\alpha$ | std. along $\alpha$ |
|-------------------------|----------|----------|----------|----------|---------|---------|---------|---------------|----------------|
| $\lambda = 0.1$ | 23.40    | 23.10    | 22.45    | 22.51    | 22.25   | 21.87   | 15.92   | 21.64         | 2.58           |
| $\lambda = 0.5$ | 23.29    | 23.02    | 23.13    | 22.80    | 21.87   | 20.02   | 17.46   | 21.66         | 2.17           |
| $\lambda = 0.9$ | 23.12    | 23.25    | 22.89    | 22.75    | 21.99   | 18.58   | 15.04   | 21.09         | 3.12           |
| Avg. along $\lambda$ | 23.27    | 23.12    | 22.82    | 22.69    | 22.04   | 20.16   | 16.14   | —             | —              |
| std. along $\lambda$ | 0.14     | 0.12     | 0.34     | 0.16     | 0.19    | 1.65    | 1.22    | —             | — |

[1] DistiLLM: Towards Streamlined Distillation for Large Language Models

---

### Author Response · Authors · 2025-11-21
**Dear Reviewers**

Dear Reviewers

We sincerely appreciate the constructive feedback on our work and are grateful for the time and effort you have dedicated to reviewing it. In the following, we address the concerns commonly raised across reviewers, followed by individual responses to each reviewer’s specific concerns. We have revised the manuscript accordingly, with highlighted in purple. Please check our responses and feel free to provide any additional comments. We would be happy to offer further clarification on any points that remain unclear.

Best regards

---

### Author Response · Authors · 2025-12-03
**Summary of Discussion Period [1/2]**

Dear AC and Reviewers,

We sincerely appreciate the time, effort, and dedication to review our manuscript. The comprehensive feedback has been invaluable in improving both the clarity and the overall quality of our work. We are grateful for the acknowledgement from the reviewers for the **addressing core limitations of KD for LLMs** (`Gj8D`), **novelty** (`Gj8D`, `yQkB`), **principled, generalized and unified framework** (`Gj8D`, `RkqC`), **solid theoretical foundation** (`Gj8D`, `RkqC`), and **comprehensive experiments with consistent improvements** (`Gj8D`, `yQkB`, `RkqC`) in our work.

The reviewers also raised several concerns, and we have attempted to address these concerns as thoroughly as possible. Below, we summarize the raised key concerns and our responses.

* **About $\alpha$ selection**
    * [*Tuning guidelines for $\alpha$*] The reviewers pointed out that there is a lack of a range or tuning strategy that can effectively search $\alpha$ which achieve generally good performance (`Gj8D`, `RkqC`). We provided the tuning guidelines grounded on our theoretical analysis of $\alpha$ which is included in the previous manuscript. Based on these theoretical insights based tuning guidelines, we utilized the range for $\alpha$ and experimentally demonstrated the effectiveness of this range.
    * [*Adaptive $\alpha$ scheduling via overlap*] The reviewers inquired whether there is any systematic and adaptive strategy for the $\alpha$ selection (`Gj8D`, `RkqC`). We introduced a curriculum-based adaptive $\alpha$ scheduling based on the degree of overlap between token-level teacher distribution and student distribution. We also provided the pseudocode and the connection with total variation distance, and further demonstrated through additional experiments that the adaptive strategy achieves higher performance than the fixed strategy despite having a comparable time cost.
* **Conflict between $\alpha$ and divergence**
    * [*Root cause analysis*] The reviewers requested a discussion on why certain combinations of $\alpha$ and divergence (e.g., $D_{RKL}(p || r_\theta^{(\alpha,\lambda)})$ with $\alpha = 1$) lead to poor performance (`Gj8D`, `RkqC`). We explained that when $\alpha = 1$, the support of the $\alpha$-mixture assistant distribution becomes narrow, and as a result, divergences that compute expectations using this distribution may suffer from unstable training and insufficient knowledge transfer. We have already discussed this issue in Appendix D of the previous manuscript.
    * [*Mitigation via distribution flattening*] The reviewers inquired whether there is a way to mitigate such vulnerable combinations rather than merely avoiding them (`Gj8D`, `RkqC`). Noting that the main cause is the narrow support of the $\alpha$-mixture assistant distribution, we proposed employing temperature scaling, a simple flattening technique that is widely used in practice. We demonstrated the effectiveness of temperature scaling through additional experiments.
* **Relationship between $\alpha$ and $\lambda$**
    * The reviewer asked about the relationship between $\lambda$, which is previously studied, and the newly proposed $\alpha$ in this work (`Gj8D`). We clarified that $\alpha$ controls the geometry of interpolation path and $\lambda$ determines the portion of interpolation. Through the additional experiments, we revealed several interactions which is aligned with our theoretical analysis.
* **Extensibility to model size, complex task, and diverse metric**
    * [*Model size*] We reported the results for OpenLLaMA2-7B $\rightarrow$ OpenLLaMA2-3B in the previous manuscript. Furthermore, to verify whether our method remains effective even with teacher models larger than 10B (`Gj8D`), we additionally conducted experiments on Qwen2.5-14B-Instruct $\rightarrow$ Qwen2.5-1.5B-Instruct, and the proposed AMiD exhibit the higher performance than the baseline.
    * [*Complex task*] We already included experimental results on translation, summarization, and reasoning tasks in the previous manuscript. Moreover, to validate the generalization across diverse complex tasks (`Gj8D`, `yQkB`), we additionally conducted experiments on reasoning and code generation tasks using new datasets and models; and the results demonstrate the effectiveness of AMiD.
    * [*Diverse metric*] To verify the superior performance of AMiD beyond ROUGE-L (`yQkB`), we additionally conducted the evaluations with other metrics such as MT-Bench and LLM-as-a-Judge’s win-rate. As a result, AMiD consistently shows the strong performance.

*(Continued below)*

---

> ### Author Response · Authors · 2025-12-03
> **Summary of Discussion Period [2/2]**
>
> *(Continued)*
>
> * **Robustness to the various experimental configurations**
>     * [*Divergence*] To confirm the compatibility of $\alpha$-mixture assistant distribution with divergences (`Gj8D`), we already presented the experimental results of the extensive combinations between the divergence and $\alpha$ in the previous manuscript. Furthermore, through additional experiments, we demonstrated that AMiD achieves strong performance across a wide range of $\alpha_{AB}$ and $\beta_{AB}$ combinations of the $\alpha_{AB}$-$\beta_{AB}$ divergence, which is used for our default setting (`RkqC`).
>     * [*Optimizer*] We repeatedly confirmed that AMiD achieves the best performance even when using different optimizer through additional experiments (`Gj8D`).
>     * [*Learning rate scheduling*] We also showed that AMiD consistently exhibits the superior performance under the different learning rate scheduling through additional experiments (`Gj8D`).
> * **Comparison with recent contrastive-based methods**
>     * The reviewer pointed out the lack of comparisons with recently proposed contrastive-based distillation methods (`Gj8D`). We demonstrated that AMiD remains effective even when compared to the recent contrastive-based distillation methods. Furthermore, we claimed that since our $\alpha$-mixture assistant distribution is a theoretically valid distribution, it can be combined with contrastive-based methods. The additional experiments exhibit that incorporating $\alpha$-mixture assistant distribution contributes to further performance improvements.
> * **Further explanation**
>     * We provided further clarification regarding the meaning of adjusting $\alpha$ (`yQkB`), the reason for better performance of $\alpha \neq \pm 1$ (`RkqC`), the detailed captions (`RkqC`), and the experimental details (`yQkB`).
> * **Improvements in paper writing**
>     * We revised the manuscript to improve the readability (`RkqC`) and remove the ambiguity from the inconsistent notation (`RkqC`).
>
> We have incorporated the points raised by the reviewers into the revised manuscript, which has been submitted. We hope that this summary provides a deeper understanding of the proposed method and addresses the reviewers’ concerns.
>
> Once again, we sincerely express our gratitude to the AC and the all reviewers.
>
> Best Regards,

---

### Meta-Review · Area_Chair_nrpn · 2026-01-06

**Summary:**

Reviewers generally acknowledge the novelty and theoretical grounding of the α-mixture assistant distribution and the AMiD framework, as well as the comprehensive experimental validation across tasks, model scales, and divergences. Their main concerns focus on practical guidance for selecting the α parameter and its interaction with divergence choices, the gap between theoretical optimality and observed instability in certain settings, clarity and readability of the exposition (particularly Section 3 and figure captions). These concerns informed the cautious but generally positive recommendations and the need for clarifications and additional experiments. The authors have responded to these concerns and have largely addressed them. Therefore, I recommend accepting this paper.

**Reviewer Concerns:**

The authors have effectively addressed several key concerns raised by the reviewers. Specifically:

α Parameter Selection and Practical Guidance: The authors provided an overlap-based adaptive α scheduling strategy, clarifying how α can be set and adjusted across tasks, model scales, and divergences.

Theory-Practice Gap and Stability Issues: They explained the instability observed with certain divergence–α combinations (e.g., DRKL with α=1) and provided guidance on avoiding these configurations, supported by gradient analysis and additional experiments.

Experimental Transparency: The authors clarified all hyperparameters, divergence choices, α/λ values, and added additional reasoning and reasoning dataset results.

Comparison with Recent Methods: Additional comparisons to contrastive-based distillation approaches were included, demonstrating AMiD’s effectiveness relative to newer baselines.

Remaining concerns are relatively minor and mainly pertain to future work rather than fundamental flaws:

Extension to Very Large Models and Complex Tasks: The method’s performance on extremely large-scale LLMs (10B+) or highly complex tasks (e.g., multi-step reasoning, code generation) has not yet been fully explored.

Potential Further α/Divergence Optimization: More sophisticated adaptive strategies could further enhance performance, which remains an opportunity for future work.

Overall, the rebuttal successfully resolves the reviewers’ primary concerns, leaving only minor points related to readability and extended evaluation for future work.

**Reviewer Scores:**

The authors have addressed major concerns raised by the reviewers through detailed responses and additional experiments. Therefore, it is likely that the reviewers would maintain their initial positive evaluations, and minor outstanding points would not substantially change their scores.

---

### Decision · Program_Chairs · 2026-01-26

Accept (Poster)